# Biodegradable Polymers and Polymer Composites with Antibacterial Properties

**DOI:** 10.3390/ijms24087473

**Published:** 2023-04-18

**Authors:** Anna Smola-Dmochowska, Kamila Lewicka, Alicja Macyk, Piotr Rychter, Elżbieta Pamuła, Piotr Dobrzyński

**Affiliations:** 1Centre of Polymer and Carbon Materials, Polish Academy of Sciences, 34 Marii Curie-Skłodowskiej Str., 41-819 Zabrze, Poland; asmola@cmpw-pan.pl; 2Faculty of Science and Technology, Jan Dlugosz University in Czestochowa, 13/15 Armii Krajowej Av., 42-200 Czestochowa, Poland; k.lewicka@ujd.edu.pl (K.L.); p.rychter@ujd.edu.pl (P.R.); 3Department of Biomaterials and Composites, Faculty of Materials Science and Ceramics, AGH University of Science and Technology, 30 Mickiewicza Av., 30-059 Kraków, Poland; amacyk@student.agh.edu.pl

**Keywords:** biodegradable polymers, antibacterial polymers, polysaccharides, peptides, antimicrobial peptides, polycarbonates, polyesters, biomaterials, antibacterial composites, food packaging

## Abstract

Antibiotic resistance is one of the greatest threats to global health and food security today. It becomes increasingly difficult to treat infectious disorders because antibiotics, even the newest ones, are becoming less and less effective. One of the ways taken in the Global Plan of Action announced at the World Health Assembly in May 2015 is to ensure the prevention and treatment of infectious diseases. In order to do so, attempts are made to develop new antimicrobial therapeutics, including biomaterials with antibacterial activity, such as polycationic polymers, polypeptides, and polymeric systems, to provide non-antibiotic therapeutic agents, such as selected biologically active nanoparticles and chemical compounds. Another key issue is preventing food from contamination by developing antibacterial packaging materials, particularly based on degradable polymers and biocomposites. This review, in a cross-sectional way, describes the most significant research activities conducted in recent years in the field of the development of polymeric materials and polymer composites with antibacterial properties. We particularly focus on natural polymers, i.e., polysaccharides and polypeptides, which present a mechanism for combating many highly pathogenic microorganisms. We also attempt to use this knowledge to obtain synthetic polymers with similar antibacterial activity.

## 1. Introduction

Microbial contamination is a hard problem not only in medicine and health care but also in many other branches of human activity, such as water purification systems, hospitals, dental equipment, food packaging, food storage, household sanitation, etc. Everywhere, the growing resistance of many pathogenic microorganisms to antibiotics, a phenomenon that has been occurring more and more recently, has become a serious problem. In simple terms, the main reason for antimicrobial resistance is the widespread and huge increase in the use of antibiotics. Especially during the COVID-19 pandemic, treatment with antibiotics practically without medical supervision was used on a massive scale [1].

Furthermore, in the food industry, low antibiotic doses are routinely used for food processing and storage, allowing the survival of some microorganisms and in turn, permitting the development of strains with antibiotic resistance. This phenomenon causes an increase in the number of bacteria with developed drug resistance to common antibiotics. In addition, organic antibacterial agents used in agriculture to promote crop growth and prevent microorganism-induced diseases, such as quaternary ammonium salts, quaternary phosphonium salts, organotin compounds, and halogenated amine, also cause the formation of bacteria resistant to these compounds [2].

On the other hand, fewer and fewer new antibiotics are being introduced on the medical market. In 2019, the World Health Organisation (WHO) identified only 32 antibiotics in the clinical development stage that address the WHO list of priority pathogens, of which only six were classified as innovative. Antibiotic shortages affect countries at all levels of development, especially healthcare systems [3]. Additionally, the lack of access to high-quality antimicrobials remains a major problem. Therefore, it becomes of key importance to look for new materials that do not induce drug resistance development and whose use should reduce the risk of an epidemic outbreak or the appearance of mass infections during treatment, especially surgical treatment.

Bacterial contamination is a serious problem, especially in the medical disciplines where medical devices and biomaterials are used. Antimicrobial biomaterials, such as polycationic and other bactericidal polymers and polymeric composites, as well as polymer carrier-assisted delivery of nonantibiotic therapeutic agents, such as antimicrobial peptides and plant-derived polyphenols, have improved our ability to treat antibiotic-resistant and recurrent infections. Biomaterials not only can target the delivery of multiple agents but also can provide prolonged antibacterial effects at the site of infection, thereby reducing potential systemic side effects. Another interesting application of antibacterial polymers is their use in the production of food wrapping, thus improving food safety and extending its shelf life. In such applications, biodegradable polymeric materials, non-toxic in contact with food, with antibacterial properties, and cheap to produce and process are in demand.

In this review, we present recent advances in the development of biodegradable polymers and composites with antibacterial properties to be applied in many areas: particularly in medicine and food packaging.

## 2. Mechanisms of Antibacterial Action Used in the Development of Polymers and Polymer Composites with Antibacterial Properties

To fully understand the antimicrobial mechanisms of polymers, we need to know the structural characteristics of bacteria [4]. The bacterium is mainly composed of a cell wall, a membrane, and intracellular components. Both Gram-positive and Gram-negative bacteria possess cytoplasmic membranes, which are phospholipid bilayers containing functional proteins. The phospholipid bilayers include phosphatidylglycerol, cardiolipin anionic lipids, and phosphatidylserine. Compared to the membranes of mammalian cells, the bacterial membrane is much more abundant in zwitterionic lipids such as phosphatidylcholine, cholesterol, and phosphatidylethanolamine. For this reason, the membranes of bacteria are more negatively charged compared to those of mammals [5]. This difference between mammalian and bacteria cells is crucial for the development of a selective antibacterial compound capable of killing bacteria and being simultaneously non-toxic against human cells.

Bacteria can be roughly divided into two categories: Gram-positive bacteria and Gram-negative bacteria. The mode of antibacterial activity is a complicated process that differs between both categories due to differences in the composition of the cell wall. In Gram-positive bacteria, the cell wall is composed of peptidoglycan (PG), teichoic acids (TA) covalently linked to PG, and lipoteichoic acids (LTA) tied to the microorganism cell membrane. Teichoic acids are responsible for the structural stability of the cell wall, and they are crucial for the function of various membrane-bound enzymes.

Gram-negative bacteria possess an outer membrane (OM) containing lipopolysaccharide (LPS), which provides the bacterium with a hydrophilic surface. The lipid components of the LPS molecules contain anionic groups (phosphate, carboxyl), which contribute to the stability of the LPS layer through electrostatic interactions with divalent cations. The OM serves as a penetration barrier against macromolecules and hydrophobic compounds, so Gram-negative bacteria are relatively resistant to hydrophobic antibiotics and drugs.

The cytoplasmic membrane is a very sensitive point in the fight against bacteria. Most cationic polymers combat bacteria via electrostatic attraction to the cell membrane (polymers with positively charged functional groups such as amine or guanidine molecules vs. negatively charged bacterial membranes) and by hydrophobic insertion into lipid tails cause lysis of the cell membrane. Because of these opposite electrostatic forces, polymers with embedded cationic groups interact with the anionic bacterial membrane, and, as a consequence, their penetration through the membrane to the cytoplasm may have a place. Bacterial cell membranes can also be broken through insertion. It is known that antimicrobial polymers are harmful to negatively charged bacteria membranes when their cationic concentration reaches a certain amount to achieve the multivalence effect. Amphiphilic polymers, as a representative of macromolecules consisting of cationic and hydrophobic residues, cause the death of bacteria cells through the membrane lysis mechanism [6]. Such a structure facilitates antimicrobial activity because the hydrophobic region of the polymer molecules parallels the membrane lipids, while the hydrophilic ends form pores in the cytoplasmic membrane. There are several membrane lysis mechanisms (Wimley, 2010), including (1) the impact of antimicrobial polymers (AP) on the phospholipid structure of bacterial membranes that cause deterioration of the total volume in membranes, (2) spiral insertion of AP into the bacterial membrane that forms circular holes in the membrane surface, and (3) APs assemble into bundle-like spiral molecules in the membrane [7]. According to Silhavy et al. (2010) and Cox & Wright (2013), Gram-negative bacteria that possess additional outer membranes hinder macromolecule permeations and therefore are more protected against antibacterial polymers when compared to Gram-positive bacteria [4,8].

Cationic polymers are representative of the most recognizable antimicrobial polymers among all macromolecules. However, one of their drawbacks is low selectivity (Gordon et al., 2005) because the amphiphilic structure of cationic and hydrophobic polymers may interact not only with bacteria but also with mammalian cells, resulting in cytotoxicity. For this reason, cationic polymers may not be suitable for clinical applications [9]. Antimicrobial cationic polymers should contain two functional components, namely cationic groups and hydrophobic groups in monomers [10]. Ammonium, sulfonium, and phosphonium ions are one of the most representative cationic centers. Cationic groups are those that facilitate the adsorption of antimicrobial cationic polymers to the surface of microbial membranes. Ammonium groups (primary, secondary, tertiary, and quaternary), iminium (pyridinium, imidazolium, guanidinium salts), sulfonium, and phosphonium ions are one of the most representative cationic centers [10,11,12].

Cationic polymers with primary, secondary, or tertiary ammonium groups usually demonstrate relatively high antimicrobial activity compared to polymers containing quaternary ammonium groups. Palermo and Kuroda (2009) found that the composition of cationic polymers (hereafter peptides) determines the hemolytic activity that corresponds to the bactericidal activity [13]. This phenomenon may be very useful in medical applications. Ng et al. (2014) compared the antimicrobial activity of polymers containing the pyridinium and imidazolium groups as a representative of the iminium groups with the quaternary ammonium group [14]. They found that iminium-containing cationic polymers have a relatively low minimum inhibitory concentration (MIC) against various bacteria and fungi compared with the analogs of quaternary ammonium groups. An excellent example of the known mechanism of polymers containing alkyl ammonium cation is the antibacterial effect of chitosan. Chitosan has antimicrobial activity against Gram-positive bacteria (such as *B. cereus*, *B. megaterium*, *L. plantarum*, *L. brevis*, *L. bulgaricus*, *L.monocytogenes*, and *S. aureus*) and Gram-negative bacteria (such as *E. coli*, *E. aerogenes*, *P. aeruginosa*, *P. fluorescens*, *S. typhimurium*, *V. parahemolyticus*, and *V. cholera*). Some researchers have shown that chitosan has a stronger bactericidal effect against Gram-negative bacteria [15], while others claim that the opposite is true [16]. The mode of antifungal activity of chitosan depends on the structure of the fungal cell wall that is composed of chitin adjacent to the cell membrane, β-D-glucans and mannoproteins or mannan as the outer layer of the cell wall (Figure 1) [4,17,18,19,20].

The study of Fernandez-Saiz et al. (2009) tested the antimicrobial activity of chitosan acetate films against *S. aureus* and *Salmonella* spp. They suggested that Gram-positive microorganisms should be more susceptible than Gram-negative ones, which is related to the presence of a thick layer of peptidoglycan and teichoic acids in the cell wall of Gram-negative organisms. The teichoic acid backbone is highly charged by negatively charged phosphate groups that can cause electrostatic interactions with cationic antimicrobial compounds such as chitosan [21].

Hydrophobic groups are one of the factors that affect the antimicrobial activity of polymers. They usually penetrate the membrane; therefore, the composition of the monomer containing such a group has to be considered before synthesis. Both chain length and type are crucial from an antibacterial effect point of view. In the case of antimicrobial cationic polymers, the length of the hydrophobic alkyl chains determines their effectiveness. Engler et al. (2013) examined the antimicrobial activity of several polycarbonates with different lengths of alkyl chains (between the quaternary ammonium moiety and the polymer backbone) [22]. They found that the longer the alkyl chains were, the lower the MICs were noticed. This is probably due to the fact that the hydrophobic groups of antimicrobial cationic polymers can interact with the lipid bilayer of microbial membranes and, consequently, can cause cytoplasm leakage leading to microbe cell death [23]. Although longer alkyl chains exhibited stronger antimicrobial activities when compared with shorter ones, their length should be in the optimal range because if the chain length is too high, hydrophobic structures may aggregate in the polymer and weaken their biocidal activity, or they may result in an increase in hemolytic activity. In addition to the length of the hydrophobic group, its type also plays an important role in the antimicrobial activity of polymers. In addition to linear alkyl chains, cyclic structures determine antimicrobial activity [24]. In their study, the authors compared two types of random nylon-3 copolymers with cyclohexane groups or analog acyclic groups and found that polymers with cyclohexane groups demonstrated higher antimicrobial activity and weaker hemolytic effect than their counterparts with acyclic groups. In the other study, Wang et al. (2012) examined the antimicrobial impact of poly(ε-caprolactone) with pending derivatives of resin acids as hydrophobic groups on a broad spectrum of Gram-positive and Gram-negative microbes [25]. The authors demonstrated high selectivity between these two types of bacteria, and the MICs for Gram-positive bacteria ranged between 0.7–10.1 μM; meanwhile, for Gram-negative bacteria ranged between 3–40 μM.

Another factor that determines the antimicrobial activity of a polymer is its topology. Each polymer molecule possesses unique spatial features, and monomer structures with functional groups may form various hydrophobic or hydrophilic regions in the polymer matrix, determining the same antimicrobial properties. In the case of cationic polymers, a few types of cationic centers occur, namely main chain (linear polymers with cationic centers along the macromolecular chains), side chain (spatial distributions of cationic centers deployed on the side groups), and dendric and hyperbranched structures.

Main chain cationic polymers, due to the multiple cationic centers deployed in the polymeric backbone, facilitate the adsorption of the polymer to the surface of microbial membranes [26]. Liu et al. (2012) demonstrated the antimicrobial properties of biocompatible imidazolium-containing polymer against most pathogenic bacteria maintaining simultaneously non-hemolytic effect towards red blood cells. Pascual et al. (2015) examined the antimicrobial activities of biodegradable polycarbonate hydrogels synthesized by ring-opening polymerization (ROP) [27]. Although this hydrogel can biodegrade in 4 to 6 days, it demonstrates a very high ability to kill Gram-positive and Gram-negative bacteria and even fungi almost 100% after 18 h of incubation at 37 °C. Moreover, the hydrogel did not exhibit hemolytic activities in red blood cells after 1 h of incubation; therefore, these polycarbonate hydrogels may be successful as antimicrobial hydrogels in implantable and wound-healing biomaterials.

The main advantage of side-chain cationic polymers over main-chain cationic polymers is the fact that they can be synthesized by many methods of living polymerization and, therefore, their molecular weight, molecular weight distribution, and the same spatial distributions of cationic centers and hydrophobic groups can be relatively easily designed [28].

Dendritic and hyperbranched cationic polymers also demonstrated antimicrobial activities; however, these polymers possess structures that are very often very complicated and are not biodegradable. Dendrimers are well-defined macromolecules with narrow polydispersity and definite chemical structures that possess high charge density on surfaces, which determines their antimicrobial activities. For comparison, the topological structures of hyperbranched polymers cannot be controlled precisely; therefore, their study on a structure-activity relationship is considered difficult [29,30].

Another type of antimicrobial polymer is the host-guest system. In such a complex polymer, a host and a guest molecule are incorporated into the cavity of a macrocyclic host. The host-guest complexation (inclusion in the case of cyclodextrins) is driven by noncovalent interactions. It is a great advantage over covalent bonds because non-covalent interactions can impart host-guest systems with the ability to be reversible, degradable, and adaptive. Using a water-soluble macrocyclic host, such as cyclodextrins, the hydrophobic antimicrobial active agent can be incorporated into the host cavity, creating an inclusion complex [31].

Antimicrobial cationic polymers can be successfully used as an antimicrobial surface modifier to protect against infections, for example, in medical devices or clinical treatments. Antimicrobial cationic polymers may be incorporated on the surface in two ways, namely, covalent modification and noncovalent self-assembly. For instance, Li et al. (2011) prepared an antimicrobial surface-based porous polymeric hydrogel that has many positive charges capable of adsorption of anionic phospholipids from microbial membranes (the so-called anion sponge). Due to easy adsorption, the microbes were easily killed (herein, *E. coli*, *F. solani*, *S. aureus*, and *P. aeruginosa*) [32].

Inhibition of cell wall synthesis is one of the mechanisms determined by antibacterial peptides (AP). The bacteria of cell wall structure is mainly composed of peptidoglycans (N-acetylglucosamine and N-acetylmuramic acid bonded with beta 1,4-glycosidic bonds) which provide them mechanical support to maintain. The occurrence of cross-linking among glycans via peptide chains forms stable wall networks. Since lipid II (synthesized in the bacterial cytosol) is the essential precursor for the biosynthesis of these peptidoglycan layers, it could be the target of, for example, cationic polymers to inhibit cell wall synthesis [33]. After the synthesis of lipid II in the cytosol, it is transported by lipid shutters, undecaprenol phosphate, and lipase. In this regard, any molecules that can target lipid II shutters may inhibit cell wall synthesis [34]. Therefore, Gram-positive bacteria are much more sensitive to cationic AP because they do not have an additional outer membrane when compared to Gram-negative ones. Cationic peptides such as teixobactin, plectasin, or nisin are AP representants that have been successfully used to target lipids II of bacteria cells [35,36,37].

Cationic polymers may also target an intercellular molecule because they can permeate through the outer membrane and cytoplasmic membrane barriers and accumulate in the cytosol, resulting in the interruption of intracellular cell metabolism. For example, cationic polymer polyhexamethylene biguanide can cross the membrane and enter bacteria to inhibit cell division via direct interaction with DNA, which is a molecular basis for bacterial replication [38]. Chin et al. (2018) demonstrated similar activity of biodegradable guanidinium-functionalized polycarbonates that suppressed bacteria without membrane morphological changes. In both cases, guanidine was the functional group responsible for the cell penetration properties of both polymers [39].

Zhou et al. (2020) reported on the antimicrobial activity of cationic poly(2-oxazoline), which selectively killed *S. aureus* through a strong interaction with DNA at low concentrations that leads to the production of reactive oxygen species (ROS) capable of damaging the membrane and consequently killing bacteria [40]. Ribosomes are the organelles where protein synthesis occurs and, as a result, play a crucial role in the proper functioning of the bacteria cell. Gagnon et al. (2016) examined ribosomes as the potential target for bovine peptide bactenecin to inhibit translation initiation, resulting in the inhibition of protein synthesis [41].

In addition to the biochemical mechanism of antimicrobial polymers, physical stimuli such as mechanical, magnetic, and electrical affect bacteria cells [42]. One of the alternative antibacterial treatments is piezostimulation, which implies the application of a charge that accumulates on the surface of mechanically deformed microbially contaminated materials and thus results in the disassembling and destruction of the bacterial cells. Piezoelectric materials can affect bacteria through variable processes such as transmembrane current-induced electroporation [43], disruption of the metabolic system/membrane, positive surface charge [44], and ROS generation [45]. On the contrary, piezostimulation using biodegradable polymers is often studied to stimulate mammalian cell growth and its contribution to faster wound healing [46]. To this end, piezostimulation can be used as an alternative method to protect against acute and chronic wound infections during postoperative and posttraumatic recovery. By optimizing the piezostimulation effect, it would be possible to reduce the number of antibiotics after treatment. Piezostimulation has self-powering activation by simultaneously generating charge upon mechanical deformation and is not dependent on external electrical sources and the application of electrodes. Due to biodegradability and biocompatibility, organic piezoelectrics, such as polylactide (which has a left-handed helix orientation similar to natural piezoelectrics present in the human body, such as collagen or elastin), have gained great interest, especially in medicine [47].

Ando et al. (2017) reported that the antibacterial activity of poly-L-lactide (PLLA) reflected in the formation of pores through which bacterial contents leak was dependent on three mechanisms, including electric current, electroporation due to high voltage output and less likely ROS generation [48].

Gazvoda et al. (2022) examined the antimicrobial effect of fully organic piezoelectric biodegradable PLLA films with nanotextured or smooth surfaces on *S. epidermidis* and *E. coli* as representatives of Gram-positive and Gram-negative bacteria, respectively [49]. The differences in morphology of the surface determined piezoelectric properties and the same bactericidal impact. Piezostimulation induced bacterial death in both strains of tested bacteria. The damage in the bacterial cell was caused by the disruption of the transmembrane potential and was the main mechanism determining the death of the bacterial cells. Less important factors were pH changes and ROS generation. It is worth highlighting that the same samples did not have a harmful influence on red blood cells, demonstrating a high selectivity of the mechanism, which is very important from a human toxicology point of view and makes these materials potential for therapeutic use. Nevertheless, the antimicrobial activity of tested films was not as fast due to the poor adhesion of the bacterial cells to the surface of the films, and therefore, a more detailed study on improvement in their efficacy (using adhesion molecules that allow easier bacterial attachment to the surface of the films) is required.

Timofeeva and Kleshcheva (2011), in their review, focused on the antimicrobial activity of several nondegradable polymers such as (1) cationic polyelectrolyte salt poly(hexamethylene biguanide chloride) (PHMB) against *E. coli*, (2) quaternary ammonium/phosphonium polymers such as for example, polymethacrylate containing pendant biguanide groups and polyvinylbenzyl ammonium chloride (high biocidal activity against *S. aureus* and *E. coli*) or copolymers of 2-chloroethylvinyl ether and vinylbenzylchloride with immobilized ammonium or phosphonium salts (disruption of *S. aureus* cells) [50]. The authors turned attention to the fact that the outer envelope of cells is a net negative charge which is often stabilized by a cytoplasmic membrane composed of a phospholipid bilayer with embedded essential functional proteins, such as enzymes, the teichoic acid of the cell wall (or lipoteichoic acid) molecules of Gram-positive bacteria in the presence of divalent cations such as Mg^2+^ and Ca^2+^ and by lipopolysaccharides and phospholipids in the outer membrane of Gram-negative bacteria. The cytoplasmic membrane is semipermeable and possesses selective permeability, allowing for the transfer of solutes and metabolites in both directions (in and out of the cell body).

## 3. Natural Polymers with Antibacterial and Bacteriostatic Properties

Natural polysaccharides are polymeric carbohydrate molecules composed of simple sugar molecules connected by glycosidic links. They can be made of one type of monosaccharides (homo-polysaccharides) or of different simple sugars (hetero-polysaccharides). Polysaccharides occur in the form of linear chains or are branched and have different functional groups and different physicochemical properties [51,52]. Their functionalities divide into structural, storage, and gel-forming polysaccharides [53]. Polysaccharides are obtained from various natural resources such as animals, fungi, yeast, plants, and seaweed (Figure 2) [5,54]. Many methods are used for the extraction and production of polysaccharides, e.g., hot water extraction, alkaline extraction, fermentation, and ultrasonic technology, including microjetting and microstreaming [55,56].

This large group of biopolymers has been characterized by strong interest, especially in the last 20 years. Both publications and citations in scientific journals related to natural polysaccharides are constantly increasing. The data on the Web of Science website (http://apps.webofscience.com, accessed on 8 March 2023) shows that in 2004, 28 publications on “bioactive polysaccharides” were published, while in 2022, as many as 930 scientific articles on this subject were published. Natural polymers are owed their popularity to valuable physicochemical properties such as biodegradability, biocompatibility, nontoxicity, and hydrophilicity [57,58]. Numerous studies have proven that polysaccharides exhibit various types of biological effects, e.g., immunogenic, antioxidant, anti-inflammatory, anticancer, antithrombotic, antimutation, and antibacterial activity [51,59]. Due to their excellent properties, natural polysaccharides are used in medicine, especially in cosmetology and pharmacy, as well as in the food industry, agriculture, environmental protection, and wastewater management [58]. Examples of natural applications of antibacterial polysaccharide applications are presented in Figure 3.

### 3.1. Chitosan

Chitosan is a natural polysaccharide obtained from partial deacetylation of chitin (β-(1–4)-poly-N-acetyl-D-glucosamine), which is present in nature as ordered microfibrils and is the main structural component in the exoskeleton of arthropods and the cell walls of fungi and yeast. The main sources of chitin are crustaceans such as shrimp, crabs, and lobster. The chemical structure of chitin is similar to that of cellulose, having a hydroxyl group in each monomer substituted with an acetylamine group [17,60,61].

Today, it is still a widely used method for the chemical extraction of chitin that involves the removal of calcium carbonate (demineralization), generally by hot reaction with acid, followed by deproteinization, usually performed by alkaline treatments. The treated shell obtained after this process is chitin, which is dried and packed. The extracted raw form of chitin is characterized by a highly ordered crystal structure that makes it transparent, elastic, and quite hard. However, it has poor solubility and low reactivity [62].

The structure of chitin can be modified by removing the acetyl groups, which are bonded to amine radicals at position C2 of the glucan ring, by chemical hydrolysis in a concentrated alkaline solution at elevated temperature to produce a deacetylated form (Figure 4). When the fraction of acetylated amine groups is reduced to 40–35%, the resulting copolymer, (1 → 4)-2-amine-2deoxy-β-D-glucan and (1 → 4)-2-acetamide-2-deoxy-β-D-glucan, is then called chitosan. The degree of acetylation is very important due to its effects on the physical properties of chitin/chitosan [63,64].

Chitosan is the second most abundant natural biopolymer in nature, after cellulose. It is a non-toxic, biodegradable polymer of high molecular weight. Depending on the source and preparation procedure, its molecular weight can range from less than 100 to more than 1000 kDa with a degree of deacetylation (DA) of 30% to 95%, though by convention, only polymers with DA greater than 50% DA are referred to as chitosan. Both the molecular weight and DA affect the potential applications of chitosan, so it is important to know a correlation between the chitosan type and any further application before chitosan modification. The most important property of chitosan is its antibacterial effect. Chitosan is well known to be a bactericidal/bacteriostatic agent that acts on a wide range of common bacteria, both Gram-positive, Gram-negative, and fungi [65,66,67].

#### 3.1.1. Factors Influencing the Antimicrobial Activity of Chitosan

The antimicrobial properties of chitosan depend on essential factors, including the type of microorganism, the source of the chitosan, the concentration of chitosan, structural properties (namely the degree of deacetylation, the molecular weight), the pH of the chitosan medium, the temperature, the composite with certain materials, the derivatives of chitosan (Figure 5). Therefore, to achieve the highest antimicrobial activity, the optimal conditions for the application of chitosan must be investigated and tested [61,68].

#### 3.1.2. Sources and Concentration of Chitosan

Chitosan oligomers and polymers from different sources have been shown to have different antibacterial activities. The research by Hosseinnejad et al. (2016) showed that the antimicrobial activity of fungal chitosan was lower than that of chitosan obtained from crustacean shells. In addition, fungal chitosan, similar to crustacean chitosan, showed a better inhibitory effect on Gram-positive bacteria compared to Gram-negative bacteria [69].

The mechanism of action of the chitosan molecule in the bacterial cell depends on its concentration. On the one hand, when the concentration of chitosan is relatively low, the molecules bind to the surface of the bacterial wall, which causes a rupture of the bacterial membrane and induces leakage of bacterial cell components, ultimately leading to bacterial death. On the other hand, when the concentration of chitosan is higher, protonated chitosan can wrap on the surface of bacterial cells, which can prevent cell components from leaking out [70].

#### 3.1.3. Structural Properties—Molecular Weight and Degree of Deacetylation

Almost all studies reported a correlation between the antimicrobial effect of chitosan and its molecular weight (M_w_). Chitosans can be divided according to their molecular weight: high molecular weight (HM_w_) chitosan > 250 kDa; medium molecular weight (MM_w_) chitosan between 50 kDa and 250 kDa; and low molecular weight chitosan (LM_w_) named oligochitosan (short chain chitosan) < 50 kDa [71].

HM_w_ chitosan cannot penetrate cell membranes, accumulates on the cell surface, blocks the transport of nutrients into cells, and contributes to cell lysis. Dissociated HM_w_ chitosan molecules in solution can bind to the cell membrane, modifying its permeability, while dissociated solutions of LM_w_ chitosan molecules can bind to DNA during cell nucleus penetration and inhibit mRNA synthesis [72].

Verlee et al. (2017) reported that the lower the molecular weight of chitosan, the stronger the antimicrobial activity against tested bacteria such as *S. aureus*, *B. cereus*, *K. pneumoniae*, and *E. coli*. It is suggested that the size and conformation of its molecule play an important role in the bactericidal effectiveness of LM_w_ chitosan. The short chains of chitosan are more mobile and are more easily attracted, and therefore their ionic interaction with the bacterium is easier [73].

The degree of deacetylation (DD) is one of the basic physicochemical parameters that directly affect the structure of the chitosan chain and determines the possibilities of its application. Takahashi et al. (2008) studied the dependence of antimicrobial activity against S. aureus on chitosan DD. Conductivity test and incubation with mannitol salt agar medium showed that the higher the DD of chitosan, the higher the rate of inhibition of S. aureus growth. Younes et al. (2014) also obtained similar results. When DD was 99%, chitosan inhibited almost all types of bacteria tested at the minimum inhibitory concentration (MIC) [74,75].

#### 3.1.4. Effect of pH

One of the most important factors affecting the antimicrobial activity of chitosan and its derivatives is the pH of the environment. Native chitosan is insoluble in organic solvents and aqueous solutions above pH 7, while at pH < 6, it has polycationic behavior and dissolves well. Its solubility in dilute aqueous solutions is related to the conversion of glucosamine units to the soluble form of R-NH^3+^ [76].

Meng et al. (2012) showed that at pH lower than pKa, chitosan molecules are protonated due to the high density of amino groups (-NH^3+^) that convert to the quaternary form, giving a positive charge to the polymer, increasing the intermolecular electric repulsion, resulting in a polycationic macromolecule [71]. At a pH higher than pKa, chitosan shows the ability to lose its positive charge and precipitates as a result of the deprotonation of amino groups, becoming insoluble but remaining reactive, with the possibility of forming gels or protective layers [77].

As a result of the studies conducted, it was shown that the adsorption of chitosan on the bacterial surface increases with the decrease in pH, which is associated with the interaction of positively charged chitosan molecules with negatively charged components of the bacterial cell, such as proteins, fatty acids, and phospholipids. In this way, the permeability of the cells is disturbed, which ultimately leads to the death of the bacteria. It is suggested that the antibacterial mechanism of chitosan, based on the interaction between protonated chitosan and negatively charged cell membranes, is the most common mechanism that explains cell death [69,71,77].

The difference between the physiological, neutral pH of most bacterial cells and the pH at which chitosan is soluble causes also precipitation of chitosan molecules on the surface of the bacterial cell, resulting in the formation of a layer that blocks ion exchange channels, destabilization of the morphology and function of the cell wall, and consequently the death of the microorganism.

Research by Kulikov et al. (2015) showed that the critical factor expressing antimicrobial activity is the presence of a positive charge in the polymer structure, and not its solubility depending on the pH range [78]. In addition, when the bactericidal effect of chitosan derivatives was examined, it was shown that the presence of a positive charge was not enough, while the location of the cationic charge that determines the structure of the polymer backbone plays a decisive role. The antimicrobial effect appears to be greatest when the cationic fragments are closer to the polymer backbone and thus decreases as the functional groups are present further and further away from the polymer chain. This effect is observed when the same functional group is bound to the polymer with chains of different lengths [79,80].

#### 3.1.5. Antibacterial Activity of Chitosan and Its Derivatives

There are many studies on the minimum inhibitory concentration (MIC) of chitosan and its derivatives, with different results for different microorganisms. It depends on many factors, and the nonstandardized procedures make it difficult to compare MIC. The sample values are presented in Table 1.

The mechanisms of action of chitosan against a wide range of target organisms, such as bacteria and fungi, have been investigated and reported in many articles. Since the antimicrobial properties of chitosan are highly associated with its physicochemical characteristics and environmental conditions, the mode of action of chitosan against microbes can be classified as extracellular, intracellular, or both, depending on the target site of antimicrobial effects [93].

Because the antimicrobial activity of chitosan is due to interactions with the surface of the microorganism, four possible models have been proposed to explain this mechanism: (1) formation of a dense polymer film on the cell surface; (2) chelation of nutrients by chitosan; (3) disruption of the cell membrane or cell wall; and (4) interaction with microbial DNA [94,95,96].(a)HM_w_ chitosan forms a dense polymeric membrane on the surface of the cell, which blocks the exchange of nutrients, leading to metabolic disorders and, consequently, to the death of microbial cells. The deposition of chitosan on the cell surface can be confirmed by SEM observation. This study was conducted by Helander et al. (2001), who observed a thicker appearance of the cell walls of chitosan-treated *E. coli* and *Salmonella typhimurium*.(b)The presence of cationic charges in chitosan is a key requirement for displaying antimicrobial activity. The cationic groups in chitosan are attracted by electrostatic interactions with negatively charged components present on the surface of the bacteria. Quaternary ammonium groups (R-NH^3+^) in an acidic environment (pH < 6) compete with divalent metal ions such as Ca^2+^ and Mg^2+^ present in the bacterial cell wall for bonding to polyanions, leading to an imbalance of surface potential and mutual repulsion of negatively charged particles, and finally rupture of the cell membrane.(c)As a result of electrostatic interactions between chitosan and the anionic surface of Gram-positive and Gram-negative bacteria, the cell membrane is disrupted. In Gram-positive bacteria, positively charged chitosan can electrostatically interact with negatively charged teichoic acid in peptidoglycan, destroying the cell membrane and resulting in leakage of intracellular components and simultaneous entry of chitosan into the microbial cells. In Gram-negative bacteria, the high negative charge from LPS can be neutralized by the positive charges from chitosan. As a result of this phenomenon, the OM is torn apart, and the chitosan is absorbed into the cell, resulting in the death of the bacterial cell. Several studies have indicated that chitosan can also bind to the phosphorylated mannosyl side in fungi, leading to disruption of the plasma membrane and leakage of intracellular materials.(d)LM_w_ chitosan and its hydrolysis products can perforate the microbial cell and interact with DNA. By binding to DNA, chitosan prevents DNA transcription and interrupts the synthesis of proteins and mRNA. Studies by Fei Liu et al. (2001) used a confocal laser scanning microscope to determine the antibacterial activity of chitosan oligomers in *E. coli* cells. Studies have shown the presence of chitosan oligomers within microbial cells, confirming that the likely cause of the antibacterial effect was the prevention of DNA transcription. Xing et al. (2009) analyzed the effect of the concentration of oleyl chitosan (a derivative of the fatty acid chitosan) nanoparticles (OCNP) on changes in the bacterial genome. Studies have shown that negatively charged phosphate groups in DNA/RNA can react with positively charged amino groups in OCNP, thus inhibiting the activity of microorganisms. In their research, Márquez et al. (2013) analyzed the chemical-genetic interactions of low-molecular-weight chitosan with genes from *S. cerevisiae*. The disruption of protein synthesis by chitosan was supported by an in vivo β-galactosidase expression assay of -galactosidase, suggesting that this is a primary mode of antifungal action. Furthermore, indicated that chitosan has a minor membrane disruption effect [17,18,94,95,97,98,99].

### 3.2. Chitosan Derivatives

In order to improve the aspect of its solubility, chitosan is subjected to various modification methods. The chemical nature of chitosan provides many possibilities for covalent and ionic modifications, which allow for extensive adjustment of the properties of chitosan-based products. Chitosan is highly susceptible to chemical modification through one of its three reactive functional groups: the amino group -NH_2_, as well as the primary and secondary hydroxyl groups -OH in C(2), C(3) and C(6), respectively, which provide a basis for interaction with other polymers and biological molecules. Chitosan possesses positive ionic charges, giving it the ability to chemically bind to negatively charged fats, lipids, cholesterol, proteins, macromolecules, and metal ions [79].

Among the chitosan derivatives, the following can be distinguished e.g., quaternary ammonium chitosan, carboxymethyl chitosan, sulfonated chitosan, phosphorylated chitosan, chitosan containing alkyl/aromatic groups, and chitosan hydrogels obtained by cross-linking/hydrophobic interactions.

#### 3.2.1. Chitosan Containing Quaternary Ammonium Groups

A common method for quaternizing the amino group in chitosan is treatment with methyl iodide in the presence of NaOH to form N,N,N-trimethylchitosan (TMC). The introduction of quaternary ammonium groups can occur in any of the three reactive centers (two OH groups and an NH_2_ group).

The introduction of quaternary alkyl groups gives the polymer a permanent positive charge, which improves the solubility of chitosan in the aqueous environment. Long-chain derivatives of N-alkylated chitosan have amphiphilic properties, which play a very important role in the adhesion of these compounds to the bacterial cell membrane.

The quaternary chitosan salt makes the chitosan molecule slightly flattened, which facilitates its entry into the bacterial cell and easier unpacking of the DNA. It is suggested that the higher efficiency of alkylated chitosan in genetic transcription is due to its increased cell entry, and as the alkyl side chain in alkylated chitosan increased, the transcription efficiency increased [70,100].

Zhou et al. (2016) studied the antibacterial activity of N,N,N-trimethylchitosan (TMC) fibers against Gram-negative *E. coli* and Gram-positive *S. aureus*. The developed TMC fibers showed a more effective antibacterial effect compared to chitosan-based fibers, as demonstrated by in vitro and in vivo studies. In addition, TMC fibers in an animal wound healing test significantly improved re-epithelialization and wound contraction compared to chitosan fibers, indicating their potential for use as dressing materials [101].

The study by Mohamed et al. (2015) investigated the antimicrobial activity of TMC-based and poly(vinyl alcohol) (PVA) hydrogels. The antimicrobial activity of hydrogels was evaluated against Gram-positive bacteria *S. aureus* and *B. subtilis*; Gram-negative bacteria *E. coli* and *K. pneumoniae*; fungi *Aspergillus fumigatus* and *Geotricum candidum*. All the prepared samples under investigation showed good antimicrobial activity against the tested microorganisms [102].

#### 3.2.2. Carboxymethyl Chitosan

Carboxymethylation occurs by dispersing chitosan in 2-propanol in an alkaline medium. A 2-propanol/monochloroacetic acid mixture is then added to the resulting suspension. O- and N-carboxymethylation can occur simultaneously, although, by selecting the process parameters, it is possible to make the reaction proceed through one of them. The resulting chitosan-carboxymethyl is an amphoteric polymer whose solubility depends on pH [100,103].

Yin et al. (2018) tested the antibacterial activity of films based on the quaternary carboxymethyl chitosan derivative (QCMC) and PVA loaded with Cu^2+^. The tests showed that this material inactivated 98.3% of *S. aureus* and 99.9% of *E. coli* [104].

Similarly, Huang et al. (2016) synthesized silver nanoparticles in an aqueous solution of QCMC. These nanoparticles enhanced the antimicrobial activity of the QCMC solution against *S. aureus*. QCMC-Ag was characterized by favorable thermal stability, strong antibacterial activity, and low toxicity to humans, which means that this material may be a promising antibacterial candidate for use in the medical, food, and textile industries [105].

#### 3.2.3. Sulfonated Chitosan

Chitosan sulfonated derivatives are soluble in water and have anionic behavior in nature; they exert antibacterial, antiviral, and also anticoagulant activity similar to that reported for heparin. In addition, sulfated chitosans are potent scavengers of free radical ions, including hydroxyl and superoxide ions. During the reaction of chitosan with sulfating agents, the main substitution takes place in the C(6)-OH group, and the mechanism for obtaining sulfur-derived chitosan is the electrophilic substitution reaction, in which the proton is substituted with the -SO_3_H group.

The sulfur heteroatom can be introduced into the chitosan backbone by many methods: (1) direct reaction of chitosan with carbon-sulfur; (2) direct reaction with mercaptoacetic acid; (3) grafting of thiourea; and (4) grafting of sulfonic groups onto chitosan. As a result of the direct attachment of the sulfonate group to the amino group of chitosan, chitosan sulfate (-NH-SO_3_-) can be formed, or as a result of the introduction of compounds containing sulfonate groups (R-SO_3_-), sulfonated derivatives (-NH-R-SO_3_-) can be formed [89,106,107,108,109].

In the research carried out by Huang et al. (2019), the antimicrobial activities of sulfonated chitosan (SCS) were evaluated compared to those of unmodified chitosan hydrochloride (WCS) against *E. coli* and *S. aureus*. Based on the conducted studies, a two-fold increase in the minimum bactericidal concentrations (MBC) for SCS compared to those for WCS was demonstrated. Furthermore, the study showed that SCS could be used as an alternative to antibiotics and chemical protective agents in the medical and food industries [110].

#### 3.2.4. Phosphorylated Chitosan

Phosphorylated chitosan at pH > 6.5 has deprotonated phosphate groups that increase its solubility in water, which affects its use. Generally, chitosan phosphorylation occurs at positions C(3) and C(6). Typically, chitosan is phosphorylated by reaction with solutions of phosphoric acid, triethyl phosphate, and phosphorus pentoxide, resulting in low-phosphorylation derivatives. Phosphorus oxychloride (POCl_3_) is used to avoid the formation of intra- and intermolecular bonds that promote monophosphorylation. Therefore, to obtain selective monophosphorylated chitosan while avoiding the formation of polyphosphates, the polymer reacts with phosphorus oxychloride in N,N-dimethylformamide (DMF). The use of POCl_3_ leads to negligible degradation of the product by breaking O-glycosidic bonds.

As a result of simultaneous treatment with phosphorous acid and formaldehyde in an aqueous acidic medium of chitosan, water-soluble N-mono, and diphosphonic methylene chitosans can be obtained. In this environment, a hydrophobic alkyl chain is introduced into the free -NH_2_ groups of N-methylenephosphonic chitosan, and an amphiphilic chitosan derivative is formed.

Because chitosan phosphate derivatives have antibacterial and osteoproductive properties, they have been used in medicine (especially orthopedics) and cosmetology. Due to the cation-exchange properties of phosphate groups, they are places of specific binding of biologically active forms (e.g., cell growth factors), playing an important role in the process of tissue regeneration [100,111,112].

There are relatively few studies on the antibacterial activity of phosphorylated chitosan; however, the results of Shanmugam et al. (2015) suggest that the concentration of chitosan and phosphorylated chitosan has dependent antibacterial activity with variation against several bacterial strains [90].

#### 3.2.5. Chitosan Containing Alkyl and Aromatic Groups

Alkyl groups can be introduced in the -OH or -NH_2_ positions of chitosan in the presence of alkyl halides (i.e., ethyl, butyl, dodecyl halides) and a strong base. If chitosan is modified with halogenated alkanes, the alkylation process occurs at -NH_2_ of the C(2) position. When selective alkylation is required at the -NH_2_ position of chitosan to obtain N-alkyl chitosan, reductive amination (formation of the Schiff base) occurs first, followed by reduction to the final product. The method based on the production of Schiff’s base consists of protecting the amino groups in chitosan, enabling the modification of the hydroxyl groups, and is used in several reactions of chitosan with substituted aldehydes. The bactericidal activity was observed to be proportional to the chain length of the alkyl substituent due to the contribution of the hydrophobic properties of the derivates [113,114].

Chitosan can also be acylated in both -NH_2_ and -OH groups, forming either an amide or an ester. Acylation is promoted in the amino group because it is more nucleophilic than the hydroxyl group. A condensation reaction occurs between the primary amine and the carbonyl group, leading to the formation of the N-acylated derivative of chitosan, the Schiff base (-RC=N). By modifying the reaction conditions to protect the amino groups, O-acyl chitosan can be obtained. Since the carbonyl groups in the aldehydes or ketones with which chitosan reacts can effectively combine with the -NH_2_ groups of chitosan to form the corresponding Schiff’s base with a characteristic imine group (-RC=N-), derivatives of chitosan Schiff’s bases have been shown to significantly improve the antimicrobial activity of chitosan. Compared to unmodified chitosan, the derivatives are characterized by better hydrophilic properties, as well as charging of the molecule with positive ions [114,115].

To improve the antibacterial properties of chitosan, Tamer et al. (2016) prepared two aromatic CS Schiff bases with 4-chlorobenzaldehyde or benzophenone as a reactant. An evaluation of antimicrobial activities was performed against Gram-negative bacteria (*E. coli*, *P. aeruginosa*, and *Salmonella* sp.), Gram-positive bacteria (*S. aureus* and *B. cereus*), and strain of *Candida albicans*. The results suggested that Schiff’s bases could be applied as antimicrobial wound dressing agents for wound healing [116].

Haj et al. (2020) synthesized three new types of derivatives of CS Schiff’s base. To obtain them, they used 2-chloroquinoline-3-carbaldehyde, quinazoline-6-carbaldehyde, and oxazole-4-carbaldehyde as modifying reagents. The inhibition zone method was used to investigate the antibacterial activity of synthetic compounds against Gram-negative and Gram-positive bacteria. Antibacterial activity results revealed that compared to CS, the CS Schiff’s base derivatives achieved a significantly increased antibacterial activity against fungi and bacteria [91].

Hassan et al. (2023) synthesized two novel chitosan Schiff base derivatives via coupling with cyclohexanone and 2-N-methyl pyrrolidone [117]. The two Schiff bases demonstrated higher antimicrobial performances against Gram-negative and Gram-positive bacteria alongside *C. albicans* than the original chitosan. Simultaneously chitosan base derivatives showed no toxicity concerning fibroblast cells. Since examined microorganisms are prevalent in wound infections, formulated Schiff bases have potential in wound healing applications instead of the native chitosan.

In another study, Tamer et al. (2023) examined antimicrobial properties of chitosan via functionalization with an aromatic side chain using Schiff base bond via coupling with 2,2′,4,4′-tetrahydroxybenzophenone against pathogenic antibiotic-resistant bacteria [118]. The authors revealed much better antioxidant and antimicrobial properties of obtained chitosan derivative compared to unmodified chitosan (CS-THB derivative improved the wound healing activity of chitosan 2.1 folds). The cytotoxicity study of obtained derivative using normal human dermal fibroblast cell line (HFB4) exhibited a safety profile while retaining the anticancer effect against the human skin cancer cell line (A375). Additionally, authors made quantum chemical calculations and revealed that combining polyphenol with chitosan makes it more effective as an antioxidant than either chitosan or polyphenol alone. With this respect, such a chitosan Schiff base derivative could be utilized for tissue regeneration applications.

#### 3.2.6. Chitosan Hydrogels

Chitosan is a biopolymer that can form hydrogels with slight modifications to the pH or ionic strength. Because the amino acids of chitosan are protonated in an acidic environment, electrostatic repulsion occurs, which promotes swelling of the chitosan structure. The formation of hydrogels is favored by electrostatic interactions of the hydroxyl groups in the C(3) and C(6) positions and the amino group in the C(2) positions of the monomers. It has been shown that the degree of cross-linking of chitosan is related to the properties of hydrogels, such as the degree of swelling, mechanical strength, and pore size. Chitosan can form cross-linked three-dimensional structures with dialdehydes (e.g., glutaraldehyde or glyoxal). In addition, biocompatible chitosan hydrogels were synthesized with glutaric and adipic acids. Since chitosan is a biocompatible polymer, these hydrogels have been used as controlled drug release systems, as well as in the production of biodegradable sutures or dressings for wounds and burns [100,119,120].

The use of chitosan hydrogels was limited because of their poor flexibility. Therefore, it was decided to develop various composite hydrogels. In the research of Wu et al. (2018) developed chitosan/sodium alginate (CS-ALG) hydrogels that were used to deliver lysozyme to eliminate food-borne microorganisms. CS/ALG was more effective in inhibiting *S. aureus* than *E. coli*, while CS/ALG loaded with lysozyme showed increased activity against both *S. aureus* and *E. coli* [121].

Huang et al. (2019) designed a series of hydrogels based on chitosan or thiolated chitosan with poly(ethylene glycol) diacrylate (PEGDA). The developed materials showed good antibacterial activity, and their antibacterial effectiveness against *E. coli* and *S. aureus* was greater than 80% after incubation for 6, 12, 24, and 36 h [122].

However, in the study by Hu et al. (2019), double cross-linking hydrogels based on maleylated chitosan (CS-MA) and hyaluronan were found to show significantly higher antibacterial efficacy against *S. aureus* than chemically cross-linking hydrogels, which was associated with the synergistic effect of CS-MA and chitosan [123].

Silver nanoparticles (AgNPs) are also used as an additive in the development of antimicrobial hydrogels [85]. Nesović et al. (2019) developed biocompatible chitosan and poly (vinyl alcohol) (PVA) hydrogels with incorporated AgNPs for use as antibacterial wound dressings. After 1 h of incubation with AgNP-containing hydrogels, the number of *S. aureus* and *E. coli* bacteria was significantly reduced. Importantly, the Ag release profile was characterized by an initial burst to prevent bacterial infection, followed by a slow release until day 28, maintaining a sterile environment around the wound [124].

Jiang et al. (2023) examined the antimicrobial properties of sericin-AgNPs/Curcumin antimicrobial agent encapsulated in a biobased physically double cross-linking 3D structure network of sodium alginate-chitosan [125]. This novel composite sponge demonstrated good mechanical properties, high porosity, an interconnected porous structure, moisture retention ability, and swelling properties. In vitro antibacterial study of obtained composite revealed excellent antibacterial and biocompatibility properties. In vivo animal experiments confirmed that SC/Se-Ag/Cur sponges promoted angiogenesis and repair of *P. aeruginosa* or *S.aureus*-infected wounds, thus, this material may have the potential for regenerating damaged skin tissue.

#### 3.2.7. Chitosan-Metal Composites

Currently, the preparation of chitosan composites with metals has become the subject of research in the field of antibacterial properties. Many studies have shown that the addition of metals to chitosan can improve its antibacterial effect.

Raghavendra et al. (2016) have shown that the presence of AgNPs enhances the antibacterial effect of chitosan. AgNPs accumulate on the bacterial membrane and create cavities in it, leading to the leakage of biologically necessary lipopolysaccharide molecules and proteins from the bacterial membrane, ultimately leading to the death of the bacterial cell [126].

The study by An et al. (2014) tested the antibacterial effect of chitosan/silver microspheres against *E. coli*, *S. aureus, Rhizopus*, and *Mucor* was tested. Research carried out showed that the addition of silver microspheres to chitosan significantly improved the antibacterial effect of chitosan. The antibacterial effect of chitosan-Ag microspheres was also greater than that of silver microspheres [127].

Lu et al. (2018) studied the antibacterial properties of Au-chitosan with N-heterocyclic molecules against *E. coli* and *S. aureus*. The researchers demonstrated that the introduction of Au and 2-mercapto-1-methylimidazole improved the antimicrobial properties of chitosan [128].

Furthermore, Khan et al. (2013) showed that zinc, which forms a complex with chitosan, increases antimicrobial activity. The antibacterial response of chitosan and chitosan-Zn complexes showed a broad antibacterial spectrum against both Gram-positive (*P. aeruginosa*, *S. aureus*) and Gram-negative (*E. coli*) bacteria. Furthermore, the complexes showed excellent antifungal activity with no growth of *A. fumigatus* and *F. solani* even after two weeks [129].

In addition, copper was a commonly used antibacterial agent. It was shown that the antibacterial activity of the chitosan-Cu nanoparticle composite against *E. coli* was higher than that of chitosan. The Cu nanoparticles in the chitosan-Cu composite can adhere to the cell wall or penetrate it, causing the leakage of proteins and other intracellular components, leading to the death of the bacterial cell [130].

Furthermore, to improve the antibacterial effect of chitosan, it can be combined with more than two metals to form composites. Mallick et al. (2015) tested the antibacterial activity of a composite of chitosan-Cu-Ag nanoparticles against *E. coli* and *Bacillus cereus*. Composite chitosan-Cu-Ag nanoparticles showed a very effective bactericidal effect, which was associated with the synergistic effect of the Cu-Ag nanoparticles. In addition, it was shown that chitosan was electrostatically connected to the bacterial cell wall, and Cu-Ag nanoparticles interacted with membrane proteins, causing perforation of the cell membrane and release of intracellular material [131].

The study by Li et al. (2017) loaded Au-Ag nanoparticles into chitosan. The chitosan-Au-Ag complex was used as a wound dressing. The inhibition zone method was carried out to evaluate antimicrobial activity and *S. aureus* and *E. coli*. CS-Au-Ag released silver ions faster, in a greater amount than chitosan dressing loaded with silver nanoparticles with the same silver content, thus showing increased antibacterial activity [132].

#### 3.2.8. Chitosan-Metal Oxide Composites

Chitosan can also form composites with metal oxides that improve its antibacterial properties [68,133,134,135]. Studies by Khan et al. (2016) analyzed the antibacterial activity of the chitosan-CuO composite against *P. aeruginosa*. The antibacterial activity of the composite was higher than that of chitosan, as evidenced by the strong zone of bacterial inhibition around the disc with the composite, while the lack of an inhibition zone around the disc with chitosan [133]. In other studies, the cellulose/chitosan/copper oxide nanoparticle composite showed good antibacterial activity against bacteria and fungi such as *S. aureus*, *S, agalactiae*, *E. coli*, *P. aeruginosa*, *S. maltophilia*, and *C. albicans*. This activity was related to the content of copper oxide nanoparticles in the composite [136].

Chitosan can also be combined with another antibacterial agent, i.e., ZnO. Several mechanisms of antimicrobial activity have been suggested: (1) the release of antibacterial ions; (2) the formation of ROS by the effect of light radiation; and (3) the interaction of zinc particles with microorganisms [137,138]. The antibacterial properties of the chitosan and ZnO–chitosan-coated textiles were tested against Gram-positive (*E. faecalis*) and Gram-negative (*E. coli*) bacteria species. More than a twofold increase in antibacterial activity was detected compared to fabrics treated with chitosan nanoparticles alone [136]. Chitosan-ZnO films also showed great antibacterial activity against bacteria that cause food spoilage *Campylobacter jejuni.* In chitosan films where the addition of zinc oxide was in the ratio of chitosan: ZnO (5:1 and 2:1), the inhibition of bacterial growth was 90% [139]. In addition, Rhaman et al. (2018) showed that the chitosan-ZnO composite films exhibited an increased antimicrobial efficacy compared to pure chitosan film and are linearly related to the amount of ZnO particles [140].

Another attractive bactericide is titanium dioxide. Siripatrawas and Kaewklin (2018) developed the active packaging of the nanocomposite of chitosan and TiO_2_ to be used as an antimicrobial and ethylene-scavenging film. Chitosan films containing different concentrations of TiO_2_ (0, 0.25, 0.5, 1, and 2% *w*/*w*) were characterized by structural, mechanical, optical, and antimicrobial properties. The chitosan film containing 1% TiO_2_ exhibited antimicrobial activity against Gram-positive (*S. aureus*) and Gram-negative (*E. coli*, *S. typhimurium*, and *P. aeruginosa*) bacteria and fungi (*Aspergillus* and *Penicillium*). It is suggested that TiO_2_ in composites can generate ROS, including O_2_ and OH•, through a photocatalytic reaction. ROS changed the permeability of the bacterial membrane, caused intracellular component leakage, and finally destroyed the cell [141]. Xing et al. (2020) also showed that the chitosan-TiO_2_ nanocomposite exhibited an inhibitory effect on the growth of *E. coli* and *S. aureus*. These results indicate that chitosan-based coating films incorporated with TiO_2_ NP could become a potential packaging system to prolong the shelf life of fruits and vegetables [142].

Studies conducted for all tested chitosan-clustered metal oxides have shown that biological activity increased as the number of metal components in the chitosan film.

### 3.3. Carrageenan

Polysaccharides obtained from algae are gaining more and more popularity due to their availability, sustainability, and specific chemical composition [143]. This group of biobased polysaccharides includes carrageenan obtained from edible red seaweed of the *Rhodophyceae* family, which in its structure contains numerous sulfur groups with an average molecular weight greater than 100 kDa [144,145]. The carrageenan structure consists of repeating 3-linked disaccharide b-D-galactopyranose (G units) and 4-linked a-D-galactopyranose (D units) or 4-linked 3,6-anhydrous-a-D-galactopyranose (DA-units) [146]. There are several types of carrageenan depending on its solubility in potassium chloride: (λ)—lambda, (κ)—kappa, (ι)—iota, (μ)—Mu, (ν)—Nu, (θ)—theta (Figure 6). The different types differ in the number and position of ester sulfate groups (22–35%), as well as the content of 3.6-DA [144]. The most important commercial carrageenans are kappa (one sulfate ester group), iota (two sulfate ester groups), and lambda (three sulfate ester groups) [146].

Carrageenans are characterized by physical properties such as thickening, gelling, emulsifying, and stabilizing abilities, which is why they are widely used in the food industry to improve the quality of, e.g., cheeses, puddings, and milk sweets [146]. They are used as stabilizers and binders in the production of low-calorie meat products [145]. In addition, they can serve as an oxygen barrier to prevent lipid oxidation in meat products [147]. Carrageenan-based films have great potential in food packaging due to their low cost, biodegradability, compatibility, and film-forming properties [148]. Carrageenan is also used in non-food items, for example, firefighting foams, air freshener gels, and shoe polish. In cosmetology, carrageenans act as a “binder” giving the desired rheological properties and providing the cosmetic quality of a “gloss” in shampoos, toothpaste, and cosmetic creams [144]. Carrageenans are also widely used in the pharmaceutical industry as an antitumor [149,150,151], immunomodulatory [150,152] and anticoagulant properties [153].

#### 3.3.1. Carrageenan Nanocomposites

Antibacterial activity can be imparted to carrageenans through various methods of their modification by incorporating nanoparticles, nanoclay, essential oils, or mixing with chitosan [1]. Silver nanoparticles are one of the most commonly used nanomaterials due to their strong antibacterial properties [154]. Recent studies describe the mechanisms underlying the antibacterial activity of AgNPs, including attachment of AgNPs to the bacterial membrane, damage to intracellular biomolecules and structures, and oxidative stress induced by AgNPs and silver ions [154,155].

J.-W. Rhim et al. (2014) received antimicrobial bio-nano composite films from k-carrageenan and silver nanoparticles (AgNP) and organically modified clay mineral for potential use as food packaging [156]. Antibacterial activity was evaluated by the agar disk diffusion method in two food-borne pathogenic bacteria: Gram-positive bacterium (*Listeria monocytogenes*) and Gram-negative bacterium (*E. coli*). The nanocomposite films containing AgNPs show pronounced antimicrobial activity against Gram-negative bacteria. In contrast, the clay-included nanocomposite film showed distinctive antimicrobial activity against Gram-positive bacteria. However, the combined use of AgNP and clay minerals exhibited more pronounced antimicrobial activity against both Gram-positive and Gram-negative bacteria.

J.-W. Rhim et al. (2019) also received antimicrobial nanocomposite films from carrageenan and silver nanoparticles using melanin as a reducing and capping agent [157]. The addition of AgNP to carrageenan improved its UV shielding and reduced the contact angle and water vapor barrier properties while increasing the thermal stability, mechanical strength, and resilience of the nanoparticle films. Carr-AgNP films showed antibacterial activity against Gram-positive bacteria (*L. monocytogenes*) and Gram-negative bacteria (*E. coli*) in the broth microdilution method. Carr-AgNP showed the highest antimicrobial activity against the Gram-negative bacteria probably due to the shape and the size of AgNP, as well as due to *L. monocytogenes* with a thicker peptidoglycan layer, making it difficult to absorb the AgNP into their cytoplasm. These nanocomposite films have the potential to be used as an active food packaging material.

In another publication J-W. Rhim et al. (2020) described carrageenan-based AgNP obtained by reducing silver nitrate using ascorbic acid and adsorbed to halloysite nanotube (HNT) and stabilized with sodium dodecyl sulfate (SDS) [158]. Antibacterial activity was evaluated using the colony count method. Carr/HNT-AgNP(SDS) showed strong antibacterial activity against Gram-negative (*E. coli*) and Gram-positive (*L. monocytogenes*) bacteria.

J-W. Rhim et al. in the latest article on modified carrageenan present studies of carrageenan with AgNP obtained from pine needles [159]. The antibacterial properties of the pine needle extract films were determined by a time-kill assay against the foodborne pathogenic bacteria *S. aureus* (Gram-positive) and *E. coli* (Gram-negative). The studies showed that the addition of AgNP prepared using a pine needle extract-mediated synthesis method gives antibacterial properties to the films obtained, while the antibacterial activity was not as strong as in the case of classical silver nanoparticles. The cell population decreased with the incubation time, with the effect being slightly higher against the Gram-positive bacteria (3.5 Log CFU/mL reduction) than against Gram-negative bacteria (2 Log CFU/mL reduction).

J. Liu et al. (2022) described antibacterial nanocomposite films obtained from soluble soybean polysaccharide, carrageenan, and silver nanoparticles (SSPS/AgNPs/Carr) [154]. The antibacterial activities of the nanocomposite films were tested against Gram-positive (*S. aureus*) and Gram-negative (*E. coli*) foodborne pathogens by the agar disk diffusion method. The mean values of the zone of inhibition (ZOI) against *E. coli* were 1.92 ± 0.25 mm and against *S. aureus* were 1.83 ± 0.13 mm. It was found that incorporation of carrageenan significantly improved the structural integrity of the films and delayed the migration of silver nanoparticles from the film to agar plates. Slow and sustained release of AgNPs from packaging films affects long-term antibacterial activity and high stability.

Gün Gök et al. (2021) reported the synthesis of carrageenan-coated silver nanoparticles (CA-AgNP) using carrageenan as a reducing and stabilizing agent and their antibacterial evaluation with the diffusion of agar wells and a liquid test [160]. Zones of inhibition were observed for carrageenan with concentrations of 5 mM and 10 mM AgNPs. CA–AgNPs showed higher inhibitory activity against the Gram-positive *S. aureus* (ZOI = 15.67 ± 0.44 mm, 17.67 ± 1.15 mm, respectively) compared to the Gram-negative *E. coli* (ZOI = 11.33 ± 0.41 mm, 12.67 ± 0.41 mm, respectively). Synthesized CA-AgNPs can be used as wound dressings or as topical agents for burns and wounds.

Pramesti et al. (2022) received and described antibacterial nanocomposite carrageenan with AgNPs using a chemical reduction method assisted by microwave irradiation using NaOH as an accelerator [161]. The antibacterial activity of nanocomposites against Gram-positive *S. aureus* was evaluated by calculating the inhibition zones (ZOI). The inhibitory power of Carr/AgNPs nanocomposites against *S. aureus* bacteria is quite large, so this material can be used as a wound dressing material. Carrageenan with 0.6% AgNPs showed ZOI 7.9. The ZOI value increased by 9.12 with increasing AgNPs content to 1.2%.

In the study of A. Bahadoran et al. (2021) the synthesis and antibacterial evaluation of silver/iron on graphitic carbon nitride nanoparticles with carrageenan (Ag/Fe/g-C_3_N_4_-Carr) were described [162]. Compounds were tested against *Klebsiella pneumoniae* and *Enterococcus faecalis* bacteria by the shake flask method. Ag/Fe/g-C_3_N_4_-Carr nanocomposites exhibit good bactericidal efficiency against both strains of bacteria.

J-W. Rhim et al. (2021) also conducted research on carrageenan with iron and silver nanoparticles for food packaging applications. Iron oxide nanoparticle (Fe_3_O_4_) was modified using 3-aminopropyl trimethoxysilane (APTMS) to increase silver ion adsorption capacity and antibacterial activity [163]. The neat carrageenan and composite films Carr/Fe_3_O_4_, Carr/Fe_3_O_4_-Ag, and Fe_3_O_4_@NH_2_-Ag were tested as antimicrobial against Gram-positive (*Listeria monocytogenes*) and Gram-negative bacteria (*E. coli*) using the broth microdilution method. Against both test bacteria, Carr/Fe_3_O_4_ showed a slight decrease in bacterial growth, while Carr/Fe_3_O_4_-Ag was able to stop bacterial growth after 12 h. The Fe_3_O_4_@NH_2_-Ag composite carrageenan films exhibited strong antimicrobial activity against *E. coli* (8.82 vs. 5.02 log reduction) and *Listeria monocytogenes* (10.09 vs. 3.93 log reduction).

Recently J-W. Rhim et al. (2021) reported another paper on carrageenan-based antimicrobial films containing iron (Fe_3_O_4_) and sulfur nanoparticles (SNPs) [164]. Three different nanoparticles (SNP, Fe_3_O_4_, and Fe_3_O_4_@SNP) were used to fabricate carrageenan nanocomposite films, which were evaluated for their antibacterial activity against Gram-negative *E. coli* and Gram-positive *L. monocytogenes*. These experiments were performed using a viable total colony count method (CFU). The SNP-added film was more active against the Gram-positive bacteria than the Gram-negative bacteria; however, the Fe_3_O_4_-modified film was more active against the Gram-negative bacteria than against the Gram-positive bacteria. However, the carrageenan film received with the addition of Fe_3_O_4_ and SNP nanoparticles showed distinctive bactericidal activity against both *E. coli* and *L. monocytogenes*.

J-W. Rhim et al. (2020) described a modified carrageenan, namely pullulan/carrageenan-based functional film by integrating copper sulfide nanoparticles (CuSNP) and D-limonene as fillers for active food packaging applications [165]. Antibacterial activity was evaluated using a broth microdilution method. The Pul/Carr/DL/CuS film showed a superficial bactericidal effect. This material showed a significant reduction in *E. coli*, *while*, in the case of *Listeria monocytogenes*, a slight decrease in growth was observed.

Li Wang et al. (2020) reported on the synthesis and antimicrobial evaluation of carrageenan and copper sulfide nanoparticles (Carr/CuS) films for food packaging [166]. CuS NPs were prepared with photothermal conversion performance via a one-step hydrothermal method. Antibacterial tests induced by photothermal exposure confirmed that the Carr/CuS film has antibacterial activity against *E. coli* and *S. aureus*. Carr/CuS nanoparticle films compared to unpackaged beef can effectively reduce *E. coli* and *S. aureus* by 52.6% and 69.8%, respectively. An antibacterial test was also performed on beef. Near-infrared (NIR) irradiation significantly improved the antibacterial activity of the Carr/CuS against bacteria in the food sample. The authors indicate that NIR irradiation for 10 min in the initial packaging process can significantly improve the antimicrobial efficiency of the carrageenan-based film.

J-W. Rhim et al. (2021) also synthesized carrageenan-based films integrated with CuO-doped titanium nanotubes (TNT-CuO) for active food packaging applications [167]. TNT nanotubes were prepared using a hydrothermal method and modified with CuO.

M.P. Sudhakar et al. (2022) obtained a bio-nanocomposite film using κ-Carrageenan and the entire *Kappaphycus alvarezii* incorporated with different nanofillers: ZnONPs, CuONPs and SiO_2_NPs [168]. The antimicrobial activity of the bio-nanocomposite films against Gram-positive (*S. aureus*) and Gram-negative (*E. coli*) was evaluated using the standard colony count method. The obtained carrageenan-based bio-nanocomposite films (CBF) and Kappaphycus-based bio-nanocomposite films (KBF) loaded with nanoparticles exhibited strong antibacterial activity against both studied strains. The addition to the carrageenan matrix did not provide antibacterial activity against *S. aureus*. The CBF films incorporated with nanoparticles exhibited stronger antibacterial properties against Gram-negative bacteria. KBF films incorporated with nanoparticles showed stronger antibacterial activity against both Gram-negative and Gram-positive bacteria compared to CBF films.

#### 3.3.2. Carrageenan Hydrogels

Carrageenan hydrogels can be attractive materials for biotechnology and biomedical applications, e.g., in wound healing, drug delivery, and tissue engineering [169].

J-W. Rhim et al. (2020) reported functional hydrogel wound dressing [170]. The hydrogel was prepared with AgNPs using lignin as a reducing and capping agent in the ĸ-carrageenan matrix cross-linked with CaCl_2_, CuCl_2_, and MgCl_2_ (Carr/Lig/AgNPs/CaCl_2_ Carr/Lig/AgNPs/CuCl_2_, and Carr/Lig/AgNPs/MgCl_2_). The antibacterial activity of the hydrogels against *S. aureus* and *E. coli* was assessed using the total viable colony count method. Antibacterial activity was more pronounced for *E. coli* than for *S. aureus*. All prepared hydrogels prevented both complete bacterial growth activity after 3–6 h of incubation.

A.M. Abdelgawad, et al. (2020) described the ability of an antimicrobial cryogel system based on carrageenan/cellulose nanocrystals (Carr/CNC) with AgNPs for wound dressing applications [171]. Antibacterial activity against *S. aureus* and *E. coli* was evaluated using the colony formation unit (CFU) technique by agar surface dilution and by calculating the growth reduction (R) of the treated samples with respect to the control sample. The cryogels obtained from CAR/CNC loaded AgNPs exhibited a 100% reduction for *S. aureus* and *E. coli*.

L. Muthulakshmi et al. (2021) reported an antibacterial Ag/carrageenan–gelatin hybrid hydrogel nanocomposite for wound healing [172]. Ag/carrageenan was mixed with a gelatin hydrogel using glutaraldehyde as a crosslinking agent to prepare the biohybrid hydrogel nanocomposites. Ag/Carr/Gelatin showed antibacterial activity against human pathogens: *Streptococcus agalactiae*, *Streptococcus pyogenes*, and *E. coli* using a disc diffusion method.

J-W. Rhim et al. (2017) obtained and characterized hydrogels and dry films by blending with metallic nanoparticles of zinc oxide (ZnO), copper oxide (CuO), and their combination using KCl as a crosslinking agent KCl [173]. Zinc oxide nanoparticles (ZnONPs) and copper oxide nanoparticles (CuONPs), because of their antimicrobial activity and UV-light barrier properties, are often used in biomedicine as wound dressing, in cosmetics in the form of facial masks, sunscreen, and toothpaste and as active food packaging. The authors studied the antibacterial activity of the obtained films against foodborne pathogenic bacteria, *Listeria monocytogenes*, and *E. coli*, using a viable cell colony count method. The films exhibited stronger antibacterial activity against the Gram-negative bacteria (*E. coli*) than the Gram-positive bacteria (*L. monocytogenes*).

F.A. Ngwabebhoh et al. (2021) described the preparation of gelatin/carrageenan/bacterial cellulose hydrogel scaffolds with self-antibacterial properties for wound healing and tissue regeneration applications [174]. Antibacterial activity was tested against three different bacteria from *S. aureus*, *E. coli*, and *K. pneumonia* using the agar disc diffusion test. The conducted research showed that the tested hydrogel scaffolds possess self-antibacterial properties with growth inhibition potency against Gram-positive and Gram-negative bacteria.

S. Tavakoli et al. (2020) studied the sprayable methacrylate kappa-carrageenan hydrogel (KaMA) by incorporation of polydopamine-modified ZnO (ZnO/PD) to cover diabetic wounds [175]. Antibacterial activity against *S. aureus* and *E. coli* was tested using a good diffusion agar assay. The incorporation of ZnO/PD nanoparticles in the carrageenan hydrogel matrix dramatically increased the level of antibacterial activity.

T. Khodaei et al. (2023) obtained and characterized the self-healing aldehyde-carrageenan hydrogel for use in wound healing applications [176]. Carrageenan was oxidized with sodium periodate (NaIO_4_) and then hydrogels using dopamine (PDA) and zinc ions (Zn^2+^) (H-OCA-Dop-Zn) were synthesized. The antibacterial activity of the hydrogels was evaluated against Gram-negative (*E. coli*) and Gram-positive (*S. aureus*) bacteria by counting the number of colonies. The antibacterial activity against *E. coli* was higher than that against *S. aureus*, as a result of the difference in their peptidoglycan layers.

Johnson et al. (2020) obtained antibacterial hydrogels composed of κ-carrageenan oligosaccharides (CO) and cellulose nanofibers (CNF) nanoparticles and subsequent drug loading, for the treatment of periodontitis [177]. Two antimicrobials named herbmedotcin and surfactin were selected to obtain the drug delivery system. The prepared hydrogel has good antimicrobial activity against pathogens. *S. mutans*, *P. gingivalis*, *F. nucleatum*, and *P. aeruginosa* cause direct or indirect periodontitis.

Özbaş et al. (2021) synthesized hydrogels based on ĸ-carrageenan (ĸ-Carr), poly(vinyl alcohol) (PVA), and hyaluronic acid (HA) as antibiotic-releasing wound dressing [178]. HA was grafted into the ĸ-Carr backbone via esterification reaction by 4-dimethylaminopyridine/1-(3-dimethylaminopropyl)3-ethyl-carbodiimide hydrochloride as the catalyst system and then PVA/HA-ĸ-Carr was prepared by the freeze–thawing method repeating four cycles. Antibacterial activity was determined by agar disk using *S. aureus* and *E. coli* bacteria. The nonloaded hydrogels did not show antibacterial activity. PVA/HA-ĸ-Carr drug-loaded (ampicillin sodium salt) showed a zone inhibition of 13 mm for *E. coli* and 12 mm for *S. aureus*.

A. Akbari et al. (2021) synthesized hydrogels based on ĸ-carrageenan (ĸ-Carr), poly(vinyl alcohol) (PVA), and calcium phosphate (HA) with potential application as a drug delivery system [179]. HA was first synthesized in situ with the use of κ-carrageenan and then the obtained HA/κ-carrageenan was dispersed in the solution of PVA. Finally, freezing-thawing cycles were used to reach a physically cross-linked PVA/Carr/HA hydrogel nanocomposite. Antibacterial activity was evaluated for the PVA/Carr/HA hydrogel and ciprofloxacin loaded PVA/Carr/HA hydrogel in the agar media with *S. aureus* and *E. coli* bacteria. The PVA/Carr/HA hydrogels did not exhibit any antibacterial activity against the *S. aureus* and *E. coli* bacteria. In contrast, the ciprofloxacin-loaded hydrogels showed considerable growth-inhibition zones.

W. Shao reported a study on Berberine (BB) loaded ĸ-carrageenan/konjac glucomannan (KGM) dried hydrogels (BB/Carr/KGM) [180]. Evaluation of antibacterial activity against *S. aureus* and *C. albicans* by the disk diffusion method. The dried BB/Carr/KGM hydrogel showed good antibacterial activity and could represent great potential for their future application as novel antibiotic materials.

J-W. Rhim et al. (2019) described κ-carrageenan-based (Carr) functional wound healing hydrogel films obtained by incorporating chitosan capped sulfur nanoparticles (SNP) and grapefruit seed extract (GSE) [181]. Antibacterial activity was evaluated against two pathogenic bacteria: *S. epidermis* and *E. coli* using a total colony count method. The hydrogel film Carr/GSE/SNP showed a synergistic effect of destroying the bacteria within 3 h of incubation.

M.U. Khan et al. (2021) reported an antibacterial and hemocompatible pH-responsive hydrogel for wound care and treatment. Hydrogels obtained from carrageenan (Carr), arabinoxylan (ARX), and reduced graphene oxide (rGO) were cross-linked via tetraethylorthosilicate (TEOS) [182]. For this hydrogel, the bacterial zone of the inhibitions was evaluated against Gram-negative: *P. aeruginosa*, *E. coli*, and Gram-positive: *S. aureus* bacteria using the disc diffusion method. Hydrogels were more active against *P. argenosa* (ZOI ≈ 31 mm) and *S. aureus* (ZOI ≈ 33 mm) bacteria and less active against *E. coli* (ZOI ≈ 29 mm).

Tunç et al. (2020) described antibacterial hydrogels obtained by combining carrageenan with agar and montmorillonite (MMT) as wound dressing materials [183]. The antibacterial activity was tested against *E. coli* and *S. aureus* bacteria by disc diffusion method for Carr/Agar/MMT hydrogel and Carr/Agar/MMT—hydrogels loaded with antibiotics: LDC (lidocaine hydrochloride) and CLP (chloramphenicol). The hydrogel-containing antibiotic drug CLP exhibited good antibacterial activity against *E. coli* (ZOI = 24.5–33.1 mm) and *S. aureus* (ZOI = 17.3–24.2 mm). The results of the research on the antibacterial activity of composites containing carrageenans are presented in Table 2.

### 3.4. Antimicrobial Peptides (AMPs)

One of the many components of defence systems against pathogenic microorganisms are peptides with antibacterial or antifungal activity. Antimicrobial peptides (AMPs) are crucial effectors of innate immunity. They exhibit antibacterial, antifungal, antiprotozoal, and often antiviral and anticancer properties. Antimicrobial peptides are low molecular weight proteins (3–10 kDa) and constitute a primitive immune defence mechanism against viruses, bacteria, parasites, and fungi. Many of them are involved in the neutralisation of pathogen endotoxins and have immunomodulatory properties, and therefore are also known as host defence peptides (HDP) [184,185,186]. Natural AMPs can be isolated from unicellular prokaryotes, as well as unicellular and multicellular eukaryotes (Figure 7) [184,185].

Most of the AMPs reported to date are of eukaryotic origin. Bacterial AMPs known as bacteriocins are thought to be produced by many bacteria and are extremely potent compared to most of their eukaryotic counterparts [187]. Bacteriocins are classified into two broad categories: lantibiotics and nonlantibiotics. Nisin was one of the first lantibiotics isolated and characterized. Nisin, produced by *L. actococcus lactis*, has been widely used for more than 50 years as a food preservative without significant resistance development. Nisin is active against a variety of Gram-positive bacteria in MICs in the low nanomolar range [187].

Antimicrobial peptides can be classified in many different ways depending on their structure, sequence, or mechanism of action [186]. AMPs can be divided according to their length, secondary and tertiary structure, and the presence or absence of disulfide bridges [185]. Most AMPs are cationic due to the presence of lysine and arginine and sometimes histidine residues [186]. Cationic antimicrobial peptides are generally classified into five structural groups: α-helical AMPs, cytosine-rich AMPs, β- sheet AMPs, AMPs rich in regular amino acids, and AMPs with rare modified amino acids [185,188].

#### 3.4.1. α-Helical AMPs

Linear α-helical peptides are among the most abundant and widespread α-helical AMPs are linear peptides without intramolecular disulfide bridges in aqueous solutions and usually have a random conformation, but when in contact with certain organic solvents such as trifluoroethanol they adopt an amphipathic α-helical structure [186]. AMPs with the α-helical structure are predominantly found in the extracellular matrix of insects (Cecropin) and frogs (Magainins). Cecropins were the first antibacterial peptides to be isolated from the pupae of silk moths (*Hyalophora cecropia*) [189]. The polypeptide chain of cecropins, consisting of 31–39 amino acids, in a hydrophobic environment, forms a structure consisting of two α-helical segments (C- and N-terminal), connected by a short hinge region. The N-terminal helix is amphipathic and more hydrophilic (positively charged), while the C-terminal domain is hydrophobic [190]. In mammals, cecropin P1 found in the porcine small intestine (*Ascaris suum*) has a similarity in amino acids to insect cecropins [191]. Magainins, consisting of 23 amino acids, were isolated from the skin of African *Xenopus laevis* frogs [189]. Magainins and their synthetic analogs exhibit antibiotic activity in a wide range of organisms and activity directed against cancer cells [192]. Dermaseptins are linear peptides with a chain length of 28 amino acids, isolated from skin frogs that belong to the Phyllomedusinae subfamily. Dermaseptin exhibits activity against *Trypanosoma cruzi* in both cell culture and blood medium, preventing infections during blood transfusions [185]. Dermaseptin not only has antibacterial properties, but also acts against some fungi and protozoa [193,194].

Another example of α-helical AMP is bombinins isolated from the skin secretion of *Bombina variegata* and *Bombina orientalis* frogs. Bombinins showed antimicrobial activity against Gram-positive and Gram-negative bacteria and fungi but were virtually inactive in hemolysis assays [195].

Other peptides capable of adopting helical structures are the aurein family. Aurein peptides are isolated from the granular dorsal glands of the Australian frog [186]. Most aurein peptides showed antibacterial activity against Gram-positive bacteria such as *S. aureus* and *S. epidermidis*. Among these peptides, aurein 2.2 is the best antibiotic agent, and aureins 1.2, 3.2, and 3.3 display the strongest activity against 30 to 50 different types of cancer [196]. Among the α-helical peptides isolated from humans, the LL-37 peptide is intensively studied. LL-37 belongs to the cathelicidin group, one of the most diverse AMPs of vertebrates, found mainly in mammals. Cathelicidin AMPs range from 12 to 80 amino acids and can adopt a variety of other structures. LL-37 exhibits a broad spectrum of antimicrobial activity against bacteria, fungi, and viral pathogens and plays an important role in immunomodulatory and inflammation responses [186,197].

Most helical peptides require C-terminus amidation for greater antimicrobial activity. The amidation of the C-terminus increases the electrostatic interaction between the positively charged peptide and the negatively charged bacterial membrane [186].

#### 3.4.2. Cysteine-Rich AMPs

Defensins are cysteine-rich antimicrobial peptides first discovered in rabbit and guinea pig granulocytes [185]. Vertebrate defensins are a whole family of peptides composed of 29–42 amino acids, with molecules that have the conformation of a three-stranded antiparallel β-sheet stabilized by three intramolecular disulfide linkages. Defensins are found in cells and tissues that are involved in the defence of the host against microbial infections. In many animals, the highest concentrations of defensins are found in granules, the storage organelles of leukocytes [198]. The range of action of defensins against bacteria, fungi, protozoa, and viruses is very wide [199]. The three classes of defensins can be found: α-defensins, β-defensins and θ-defensins, differ in the length of the peptide segments between the six cysteines and the pairing of cysteines that are connected by disulfide bonds [200].

In humans, four α-defensins have been mainly isolated from azurophilic granules of neutrophil granulocytes (HNP-1, HNP-2, HNP-3, HNP-4). In neutrophils, the α-defensins conduce to oxygen-independent killing of phagocytosed microorganisms [200]. β-defensins have been isolated from human plasma (HβD-1) and psoriatic scales (HβD-2, HβD-3). HβD-4 was detected in the testes, gastric antrum, and epithelia of the thyroid gland, lung, uterus, and kidney [201]. θ-defensins are 18-amino acid cyclic peptides found in old-world monkeys [202]. Humans also have genes that code for θ-defensins (retrocyclins) but are inactivated due to mutations that encode premature stop codons [203]. θ-defensins have antimicrobial activity in the presence of physiological concentrations of salt, divalent cations, and serum. On the contrary, the antimicrobial activities of α and β-defensins are significantly reduced in the presence of salt and divalent cations [204].

The group of cysteine-rich peptides includes the spider peptide Gomesin, an 18-residue-long peptide originally isolated from the hemocytes of the Brazilian tarantula *Acanthoscurria gomesiana*. Gomesin shows cytotoxic activity against a wide range of Gram-positive and Gram-negative bacteria, fungi, and yeast, as well as demonstrating anti-malarial, anti-cryptococcal, and anti-*Leishmania* activity. Gomesin also shows in vivo anticancer activity in a mouse model of melanoma and in vitro activity against human cancers [205]. Peneidines are the best AMPs described in crustaceans. Shrimp peneidines *Penaeus vannamei* are alkaline peptides with M_W_ 5–7 kDa, containing two domains: N-terminal proline-rich domain and C-terminal cysteine-rich. Penaeidins are active against Gram-positive and Gram-negative bacteria, and also against filamentous fungi [206].

Another example of a peptide in a cysteine-rich group is the lysozyme, the first antibacterial polypeptide (14 kDa) isolated from insects [206]. The lysozyme exhibits antimicrobial activity against microorganisms, especially Gram-positive bacteria, by hydrolyzing 1,4-beta-linkages between N-acetylmuramic acid and N-acetylglucosamine in the cell wall. The lysozyme is a natural preservative in mammalian milk and can be used as a biopreservative in dairy products [207].

#### 3.4.3. β-Sheet AMPs

The peptides of the β-sheet group are stabilized by one, two, three, or four disulfide linkages. The simplest of these peptides adopt a β-hairpin structure, while others contain a β-sheet and an α-helix structure [185]. Tachyplesins and polyphemusin peptides from the horseshoe crab, both share a β-hairpin motif stabilized by two disulfide bonds. They are molecules composed of 16–18 amino acids and an amidated C-terminus carboxyl group. They exhibit a wide range of activity against bacteria and fungi and certain viruses, including extracellular HIV-1. They also can bind to lipopolysaccharide, which is probably a factor enhancing their activity against Gram-negative bacteria [189,208].

Structurally similar to tachyplesins and protegrins is thanatin isolated from the hemipteran insect *Podisus maculiventris* [185,209]. Thanatin is an insect-derived peptide of 21 amino acids and exhibits broad spectrum activity against both Gram-negative and Gram-positive bacteria as well as against various species of fungi. Thanatin is a strongly cationic peptide and contains a distinct short eight residue basic loop [209].

Lactoferrin is an iron-binding glycoprotein with a molecular weight of 80 kDa, found in human milk and other secretions of the glandular epithelium. Lactoferricin B is a 25 amino acid proteolytic derivative of lactoferrin, which in solution has a β-sheet structure stabilized by a single disulfide bond [185]. The antimicrobial activity of lactoferrin was originally thought to be exerted by the strong chelation of iron required for microbial growth. Lactoferrin is released by neutrophils in response to inflammatory stimuli and its contribution to many aspects of host defence includes antiviral properties [210]. Lactoferrin administered orally to animals has anti-infective properties, including *Helicobacter pylori*, *Candida albicans*, and *Toxoplasma gondii*, and reduces the number of *S. aureus* and *E. coli* bacteria in bladder infections in mice [211].

#### 3.4.4. AMPs Rich in Regular Amino Acids

AMPs rich in regular amino acids have a characteristic structure stabilized by two or more disulfide bonds, and extended structures with a high content of one of the amino acids, usually proline, glycine, histidine, or tryptophan [212].

An example belonging to this group is the histatins family which are small, cationic, histidine-rich peptides of 3–4 kDa found in human saliva. The histatin family consists of several members; among them, histatin 1, 3, and 5 are the most important. These three peptides showed linear structures containing 38, 32, and 24 amino acid residues, respectively, and each of them contains seven histidine residues [200]. Of all histatins, histatin 5 has the strongest antimicrobial activity and could inhibit *Candida* species at a certain concentration (15–30 μM) [188].

AMPs rich in regular amino acid groups include the large family of cathelicidins, which are proline-rich peptides and have irregular structures [185]. Cathelicidins can be found in a variety of mammalian species, such as humans and farm animals (bovine, porcine, caprine, and chicken) [213]. Cathelicidins appear to be expressed in response to cytokines produced early during infection. Endogenous cathelicidins share strongly conserved N-terminal pre-pro regions but have highly varied C-terminal domains that give them antimicrobial properties [214].

Indolicidin is a natural peptide belonging to cathelicidins isolated from bovine neutrophils. Indolicidin is rich in tryptophan, consists of 13 amino acid residues, and has a unique “membrane-bound peptide structure” [186,215]. Indolicidin shows activity against the emergence of multidrug-resistant microorganisms, being active against bacteria, fungi, parasites, and certain viruses [215].

Another example of a peptide from the arginine-rich cathelicidin family is tritrpticin—C-amidated 13 amino acid tryptophan-rich peptide amidated C derived from a porcine cathelicidin [214]. Tritrpticin adopted an amphipathic turn structure in sodium dodecylsulfate micelles, where the three tryptophan residues are clustered and terminal cationic residues occupy a larger conformational space [216]. Tritrpticin has a wide spectrum of antimicrobial activities against Gram-positive and Gram-negative bacteria as well as some fungi [217].

Crotalicidin, a cathelicidin-related vipericidin, is a recently discovered 34 residue AMP from the venom of South American pit viper snakes. Crotalicidin demonstrated powerful antimicrobial, as well as antifungal and antiproliferative activities against several cancer cell lines but was moderately hemolytic and unstable in serum [218,219].

The group of AMP peptides with an irregular structure rich in regular amino acids, bactenecins Bac-5 [220] and Bac-7 [221] rich in proline and PR-39^50,^ peptide [210] rich in arginine residues, should also be mentioned.

#### 3.4.5. AMPs with Rare Modified Amino Acids

The least numerous groups of antimicrobial peptides are that with rare modified amino acids. Nisin is a notable example of AMPs from this group. Nisin is a polypeptide with 34 amino acid residues and consists of rare amino acids such as lanthionine, 3-methyllanthionine, dehydroalanine, and dehydrobutyrine [185]. Nisin is a lantibiotic isolated from *Lactococcus lactis* and is permitted as a safe food additive. This AMP exhibits antibacterial activities against Gram-positive bacteria and also against oral pathogenic bacteria [222].

Leucocin A is another 37 amino acids peptides isolated from *Leuconostoc gelidum*. Leucocin A forms an amphiphilic conformation and could interact with cell membranes. Such peptides undergo post-translational modification, which is a conformational effect that is not seen in other groups of antimicrobial peptides [185].

#### 3.4.6. Mechanism of Antimicrobial Peptide Action

The mechanism of antibacterial action of peptides has been extensively studied since their discovery. Membrane targeting was originally thought to be the only mechanism, but there is now growing evidence that AMPs have other modes of action. The mechanism of antibacterial action can be divided into two main classes: direct killing and immune modulation (Figure 8) [186].

Most membrane-permeabilizing AMPs are amphipathic, having cationic and hydrophobic parts. This causes an electrostatic interaction with the negatively charged cell membrane to increase the permeability of the membrane and lead to cell membrane lysis and cell content release. The hydrophobic part of an AMP helps to insert the AMP molecule into the cell membrane. After the initial electrostatic and hydrophobic interactions, the AMPs accumulate on the surface and, after reaching a certain concentration, self-assemble on the bacterial membrane. There are two activation models: transmembrane pore and non-pore models. Transmembrane pore models can be divided into barrel-stave pore and toroidal pore models. In the barrel-stave model, staves are formed first parallel to the cell membrane but then inserted perpendicularly to the plane of the membrane bilayer. This promotes lateral peptide-peptide interactions, such as ion channels of membrane proteins. Only a few peptides form barrel-stave channels (lamethicin, pardaxin, protegrins). In the toroidal pore model, AMPs align perpendicularly into the lipid bilayer, but specific peptide-peptide interactions are not present. In this model, the pores are created by both the peptide and lipid groups. The mechanism of the toroidal pore model is similar to that of the barrel-stave model, the difference being that in the toroidal-pore model, peptide helixes insert into the membrane and bind with lipids to form toroidal pore complexes. Several AMPs form toroidal-pore models such as magainin, lacticin, and aurein 2.2. Ultimately, the toroidal pore model and the barrel-stave model led to membrane depolarisation and eventually cell death [184,186,223].

AMPs can also act without forming specific pores in the membrane. In the carpet model, peptides accumulate on and orient parallels to the membrane surface and reach a threshold concentration to cover the surface of the membrane, thereby forming a “carpet” [186,223]. High concentrations of AMP are required to form micelles and destroy the microbial membrane [224]. The final collapse of the membrane bilayer structure into micelles causes the membrane to rupture in a surfactant-like manner (detergent-like). This model did not require specific peptide-peptide interactions of membrane-bound peptide monomers; the peptides are not inserted into the hydrophobic core of the membrane, nor do they assemble with their hydrophilic surfaces facing each other. This model was proposed for cecropin, indolicidin, aurein 1.2, and LL-37 [186].

The nonmembrane-targeting AMPs can be divided into two groups: those that target the bacterial cell wall and those that have intracellular targets. AMPs often interact with various precursor molecules that are required for cell wall synthesis, especially highly conserved lipid II. For example, human β defensin and α defensin 1 are based on selective binding to lipid II to confer bactericidal activity. Several AMPs have intracellular targets because AMPs do not cause membrane permeability at minimum effective concentrations but do cause bacterial death. In simplification, AMPs interact with the cytoplasmic membrane and accumulate inside the cell, where they can block critical cellular processes. Many new mechanisms are currently being discovered, e.g., inhibition of protein/nucleic acid synthesis and disruption of enzyme/protein activity [186].

AMPs can also recruit and activate immune cells, resulting in increased microbial killing and/or inflammation control. In the case of infection, it is important to produce an immune response to attract other immune cells, as to well as control the destructive effects of inflammatory diseases. Interestingly, some AMPs can induce a variety of immune responses. Human AMPS such as LL-37 and β defences hβD2—hβD4 can activate and degranulate mast cells and dendritic cells. Additionally, AMPs such as LL-37 can up-regulate the expression of chemokine receptors (IL-8RB, CXCR4, and CCR2) in macrophages. Furthermore, peptides such as LL-37 and PR-39 can stimulate angiogenesis and arteriogenesis [186,225].

The effect of some HDPs on enhancing innate immune responses to control infection, while at the same time enhancing their ability to control inflammation, makes these peptides attractive candidates for anti-infective and anti-inflammatory drugs [188]. Table 3 presents a brief overview of polypeptides tested and described in the literature with good antibacterial properties.

Among the antimicrobial peptides with interesting antibacterial properties, proteins derived from the silkworm cocoon should also be mentioned, namely silk sericin and fibroin, which are characterized by incredible biological features and wound healing capacity. Cocoon components play a role in protecting pupae from bacterial and fungal infections, and the presence of effector proteins in silkworm hemolymph can target and kill bacteria and fungi. In addition, the silk proteins sericin and fibroin have antioxidant and anticoagulant effects [226].

Silk sericin is a natural polypeptide containing 18 amino acids, enclosing strong polar side groups (serine, aspartic acid, and glutamic acid), which gives the sericin protein a hydrophilic character. This composition of amino acids allows for scavenging free radicals, which contributes to the antioxidant properties of sericin. In addition, the sericin silk layer contains organic pigments known for their antioxidant and antityrosinase properties. Silk sericin extracts exhibit both amorphous and α-helical structures, with small crystalline regions [227]. Sericin exhibits antimicrobial activity against several microorganisms such as E. coli and S. aureus and fungi such as C. albicans and A. flavus [228]. The effective antibacterial properties of sericin have been demonstrated against Gram-positive bacteria (*B. subtilis, S. aureus*, and *S. epidermidis*) [227]. Sericin has shown great potential in wound dressing and skin repair due to its beneficial effects on keratinocytes and fibroblasts [229]. Recently, numerous studies have been conducted on sericin-based hydrogels and sericin nanoparticles for wound healing applications [229,230,231].

Silk fibroin is composed of a heavy and a light polypeptide chain (connected with a disulfide bond) along with the P25 glycoprotein in a molar ratio of 6:6:1. The heavy chain consists of hydrophobic domains (glycine, alanine, and serine amino acid sequences) that can adopt a β-sheet conformation, and random hydrophilic domains with α-helical conformation containing acidic or charged amino acids (glutamic acid, aspartic acid, arginine, and lysine) [232]. This multidomain natural protein has demonstrated superior stretchability, biocompatibility, biodegradability, processability, and thermal stability [226]. The main limitation of fibroin is the lack of natural antibacterial activity. Over the last decade, many studies have been carried out on the prevention of bacterial colonization of the surface of fibroin. Different methods were implemented, such as combination, modification, and functionalisation with various antibacterial agents or alteration of the surface architecture [232]. Silk fibroin in the form of a hydrogel, sponge, film, and electrospun nanofiber, has demonstrated excellent properties as a wound dressing biomaterial, improving cell growth, proliferation, and migration of different cells involved in the wound healing process [226].

**Table 3 ijms-24-07473-t003:** Summary of antibacterial peptides, known and described in the available literature.

Peptide	Source	Active Agent	Mode of Action	References
**α-Helical AMPs**				
Cecropins	insects (e.g., *Hyalophora cecropia*, *Musca domestica*, bacteria (*Helicobacter pylori*), tunicates, ascarid nematodes, mammals (*Ascaris suum*)	strong antimicrobial activity against G^+^ and G^−^ bacteria	Membrane permeabilization; Carpet model	[233,234,235]
Magainins	*Xenopus laevis*	active against G^+^ and G^−^ bacteria, MIC for *E. coli*, *S. aureus*, *C. albicans*, *P. aeruginosa* and *Trichomonas vaginalis* is in the range of 200–300 μg/10^6^ cfu/mL, strong activity against drug-resistant *A. baumannii*	Membrane permeabilization; Toroidal pore model	[185,191,236,237]
Dermaseptins	skin secretions of frogs (*Phyllomedusa sauvagii*, *Phyllomedusa oreades*, *Phyllomedusa hypochondrialis*)	active against G^+^ and G^−^ bacteria, inhibition of C. albicans biofilm formation at 50 μg mL^−1^, MIC against *E. coli* 16 μM, *P. aeruginosa* 64 μM, *S. aureus* 32 μM	Membrane permeabilization; Carpet model	[238,239,240]
Bombinins	skin secretion of frogs (e.g., *Bombina variegata*, *Bombina orientalis*, *Bombina maxima*)	active against G^+^ (*Bacillus megaterium, S. aureus)* and G^−^ (*E. coli*, *Yersinia pseudotuberculosis*, *Pseudomonas aeruginosa*), against *Candida albicans*; no appreciable hemolytic capacity	Formation of ion channels or pores	[241,242,243,244]
Aurein	granular dorsal glands of the Australian frog (*Litoria aurea*, *L. raniformis*)	more activity against G^+^ than G^−^, MIC for *S. aureus* for aurein 1.2 = 25 μg/mL, for aurein M.3 = 3.12 μg/mL, MIC for *E. coli* for aurein 1.2 = 200 μg/mL, for aurein M.3 = 6.25 μg/mL	Membrane permeabilization; aurein 2.2 barrel-stave model, aurein 1.2 carpet model	[186,196,245,246]
LL-37	variety of cells (neutrophils, leukocytes and epithelial cells, Myelocytes, metamyelocytes), tissues, body fluids, and professional phagocytes	strong active against G^+^ and G^−^ bacteria e.g., against *L. monocytogenes* MIC = 1.5 μgmL^−1^, *S. aureus* MIC = 3.6 μgmL^−1^, *Bacillus subtilis* MIC = 2.7 μgmL^−1^, *E. coli* MIC = 0.1 μgmL^−1^, *Salmonella typhimurium* MIC = 0.4 μgmL^−1^	Membrane permeabilization; barrel-stave model and carpet model; Immune modulation	[188,200,247,248,249]
**Cysteine rich AMPs**				
α-defensins (HNP1—HNP4)	neutrophils, late promyelocytes, bone marrow, some surface epithelial cells, intestinal epithelial cells, female reproductive tract	active against G^+^ and G^−^ bacteria, e.g., against *L. monocytogenes* MIC = 39.7 μgmL^−1^, *S. aureus* MIC = 2.2 μgmL^−1^, *Bacillus subtilis* MIC = 6.4 μgmL^−1^, *E. coli* MIC = 1.8 μgmL^−1^, *Salmonella typhimurium* MIC = 0.4 μgmL^−1^	Mainly membrane permeabilization and disrupt bacterial membranes	[188,200,250]
β-defensins (HβD-1—HβD-4)	human (skin, respiratory tract, gastrointestinal tracts, human genomic sequences)	HβD-1—HβD-3 showed antibacterial activities against *E. coli* and *S. aureus* in a dose-dependent manner, HβD-3 showed a broad spectrum of antimicrobial activities against *S. aureus*, *E. faecium*, *P. aeruginosa*, *K. pneumonia*, *S. pneumoniae*, *B. cepacia*. HβD-4 showed antibacterial activity against on *F.nucleatum*, *P. gingivalis*, *P. aeruginosa*, *E. faecalis*	Immune modulation	[247,251,252,253,254]
θ-defensins	neutrophils and monocytes of the rhesus monkey	antimicrobial activities against *Staphylococcus aureus*, *Candida albicans*, *Cryptococcus neoformans*, *E. coli*	Membrane permeabilization	[203,204,255]
Gomesin	hemocytes of the spider *Acanthoscurria gomesiana*	antimicrobial activities against *E. coli*, *P. aeruginosa*, *S. aureus*, *K. pneumoniae*, *B. megaterium*	Membrane permeabilization; carpet/detergent mechanisms	[205,256,257]
Peneidines	shrimp (*Penaeus vannamei*, *Litopenaeus vannamei*)	better antimicrobial activity against G^−^ than G^+^ bacteria	Binding to superficial membrane and DNA, thereby destroying the bacterial structure and/or interfering with the bacterial proliferation	[258,259]
Lysozyme	Insects, plants, chicken egg white, body fluids, and tissues in living organisms	most effective against G^+^ bacteria	Hydrolysis in microbial cell walls, which results in the rupture of the β(1,4) linkages in their peptidoglycan	[260,261,262]
**β-sheet AMPs**				
Tachyplesins	hemocytes of horseshoe crabs (*Tachypleus tridentatus*)	antimicrobial activity against G^+^ bacteria (MIC = 0.2–0.9 μM) G^−^ bacteria (MIC = 0.3–1 μM)	Membrane permeabilization	[208,263,264]
Polyphemusin	hemocyte debris of horseshoe crabs (*Limulus polyphemus*)	antimicrobial activity against G^+^ bacteria: *S. aureus* MIC = 2 μg/mL, *S. epidermidis* MIC = 1 μg/mL, *E. faecalis* MIC = 1 μg/mL G-bacteria: *E. coli* MIC = 0.125–1 μg/mL, *S. typhimurium* MIC = 0.25–1 μg/mL, *P. aeruginosa* MIC = 0.25–1 μg/mL	Translocates into cells	[265,266]
Thanatin	insect (*Podisus maculiventris*)	activity against G^−^ bacteria: MIC < 1.2 μM for *E. coli*, *S. typhimurium*, *K. pneumoniae*, *E. cloacae*; against G^+^ bacteria: MIC < 5 μM for *A. viridans*, *M. luteus*, *B. megaterium*, *B. megate*; No activity against *S. aureus*	Membrane permeabilization	[209,267]
Lactoferricin B	bovine, human	antimicrobial activity against G^+^ bacteria (*S. mutans*, *S. epidermidis*) and G^−^ bacteria (*E. coli*, *S. typhimurium*, *P. aeruginosa*, *B. cepacia*, *B. cenocepacia*)	Prevents biofilm formation	[268,269]
**AMPs rich in regular amino acids**				
Histatins	salivary glands	Histatin 5: *S. aureus* MIC 12.5 μgmL^−1^, *P. aeruginosa* MIC 3.1 μgmL^−1^; MIC > 100 μgmL^−1^ for *B. cepacia*, *A. xylosoxidans*, *S. maltophilia*	Disruption of the plasma membrane	[188,200]
Indolicidin	cytoplasmic granules of bovine neutrophils	*S. aureus* MIC 2–30 μg/mL, *S. hemolyticus* MIC 2 μg/mL, *S. epidermidis* MIC 4–20 μg/mL, *B. cereus* MIC 12.5 μg/mL *L. monocytogenes* MIC 3–60 μg/mL, *E. coli* MIC 5–30 μg/mL, *S. enterica* MIC 100 μg/mL, *P. aeruginosa* MIC 100 μg/mL,	Membrane permeabilization; carpet model	[215,217,270,271]
Tritrpticin	Human, porcine	MIC 20–30 μg/mL for *E. coli* MIC 32 μg/mL for *S. typhimurim* and *P. aeruginosa*; MIC 8 μg/mL for *B. subtilis* and *S. epidermidis*, MIC 10–20 μg/mL for *S. aureus*	Inhibition of intracellular synthesis of protein, DNA, or RNA	[217,272]
Crotalicidin	pit viper	antibacterial activity particularly against G^−^; *P. aeruginosa* MIC 0.24–3.8 μM, *K. pneumoniae* MIC 1.9 μM, *E. coli* MIC 0.06–3.8 μM, *A. baumannii* MIC 3.8 μM	Membrane permeabilization	[218,273]
Bactenecins (Bac-5, Bac-7)	bovine neutrophils	Bac-5 is active against G^−^ bacteria; Bac-7 has antibacterial activity against *E. coli* MIC 1–2 μM, *S. enterica* MIC 1 μM, *S. marcescens* MIC 1 μM, *S. aureus* MIC > 128 μM	Older research suggests a permeabilizing mode of action; a recent study showed no strong effect on bacterial membrane integrity	[220,221,274,275]
PR-39	porcine neutrophils	*E. coli* MIC 1–4 μg/mL, *P. aeruginosa* MIC > 32 μg/mL, *S. typhimurium* MIC 4 μg/mL, *S. enterica* MIC 0.5 μM, *A. pleuropneumoniae* MIC 4–8 μM, *S. aureus* MIC > 32 μg/mL, *B. cereus* MIC > 32 μg/mL	Immune modulation; translocation across the membrane	[276,277,278,279]
**AMPs with rare modified amino acids**				
Nisin	*Lactococcus lactis*	highly active against G^+^ bacteria: *L. monocytogenes, S. aureus, B. cereus, L. plantarum, M. luteus*, and *M. flavus*	Membrane permeabilization	[222,280,281,282]
Leucocin A	*Leuconostoc pseudomesenteroides*	active against G^+^ bacteria, *L. monocytogenes* MIC 11.7–62.5 μM	Pore formation and ion disruption of the target cell	[283,284]
**Protein (polypeptide)**				
Sericin	wild silkworms (*Antheraea pernyi, Samia cynthia ricini*), domestic silkworms (*Bombyx mori*)	antimicrobial activity against G+ and G- bacteria, ZOI for *E. coli and S. aureus* 22.6 mm 22.16 mm, highly active against G+ bacteria: *B. subtilis, S. aureus, S. epidermidis*	no clarity as to the mode of action	[227,228]
Fibroin	domestic silkworms (*Bombyx mori*)	antibacterial effect of silk fibroin-based biomaterials	the mechanism has not yet been fully elucidated	[232]

MIC—minimum inhibitory concentration, ZOI—zone of inhibition.

#### 3.4.7. Antimicrobial Peptides in Clinical Trials

Although AMPs often present antibacterial activities, only a few clinical trials have been conducted with natural antimicrobial peptides so far. Most of the AMPs approved for clinical trials are analogs of natural AMPs, but some are fully synthetic. Compared to conventional antibiotics, AMPs are expensive due to their mass-scale synthesis [188]. Most AMPs in clinical trials are limited to topical applications because of systemic toxicity, susceptibility of peptides to protease degradation, and rapid renal clearance of these peptides when ingested. Many strategies have been investigated to improve the effectiveness of AMP in terms of medical applications. These include the chemical modification of AMP and the use of delivery vehicles [186]. Examples of clinically tested peptides are shown in Table 4 [186,224,285,286].

## 4. Synthetic Biodegradable Polymers with Antibacterial Properties

Along with the dynamic development of medicine and biomedical sciences, increasing importance and broadened use of biodegradable polymers is observed [287]. Virtually every application of this material in the form of biodegradable, or rather bioresorbable [288] surgical implant, dressing to heal difficult wounds, cell scaffold used in tissue engineering techniques [289], whether the implanted drug carrier requires full sterility. In these applications, the antibacterial activity of this type of medical device is very important.

Medical applications of synthetic polymers are perspective because in this field the high price of production does not play a primary role. The composition and properties of such materials are reproducible and can be adapted in a fairly wide range at the synthesis stage, which is usually problematic in the case of materials of natural origin. In the case of synthetic polymers, it is possible to tailor the final properties of the polymer to the intended application. An ideal polymer obtained synthetically must meet many requirements in order to be a more attractive and competitive material compared to modified natural polymers. Apart from the obvious conditions of biocompatibility, the polymer itself should be relatively easy to obtain and stable in terms of composition and structure during storage. The degradation time of such a polymeric material should be controlled to a large extent by the composition and structure of the polymer chain. Of course, this diversity of polymer properties is easier to obtain by using different monomers to build its chain, and hence most commonly this group of materials is copolymers.

The most important thing, however, is that this polymer should have a clear biocidal effect on a broad spectrum of pathogenic microorganisms in brief times of contact while maintaining biocompatibility with both tissues and cells and released degradation products must not be toxic and cause long-term side effects manifested by organ toxicity or pathogenesis. In the case of the materials in question, it is particularly challenging to meet the latter condition.

The choice of polymer structure depends on essential factors such as (1) copolymer degradation mechanism and expected degradation profile; (2) mechanism of antibacterial activity; (3) expected mechanical and thermal properties. With these three essential factors, it is possible to optimize the properties of the designed polymer specifically for the intended purpose of its use.

The ability to control mechanical and thermal properties and the appropriate degradation profile has been studied and explored for many years [287,290]. Bioresorbable polymers used in biomedicine, due to the possibility of ester bond hydrolysis and non-toxic degradation products, are aliphatic polyesters and copolyesters obtained by ROP; such as polylactide, lactide and glycolide, or lactide copolymers with ε-caprolactone or cyclic aliphatic carbonates [291,292]. To improve mechanical and thermal properties and/or obtain smart polymers that react to thermal stimuli or changes in pH, copolymers containing polyamide multiblocks are synthesized [293], or those containing side functional groups [294]. By controlling in this manner, the degree of crystallinity or wettability of the obtained copolymers, it is also possible to shape the rate and course of their hydrolysis.

The antibacterial activity of commonly used bioresorbable polymers is weak [295], therefore, products made of them are often modified with special antibacterial coatings, as is the case with nondegradable products [296,297,298,299,300]. However, because of the course of degradation of such materials, preceded by strong surface erosion changes, special coatings, even chemically naturally bonded to the polymer, cannot be a completely reliable barrier against colonization by microorganisms. Bioresorbable medical devices made of polymers with strong antibacterial and antifungal properties are much more interesting in terms of their practical application. These polymers are obtained in many ways using known mechanisms of antibacterial activity.

### 4.1. Biomimetic Polymers

A large part of the synthesized antibacterial bioresorbable polymers were obtained during attempts to mimic natural substances, primarily antimicrobial polypeptides (AMPs). These peptides are distinguished by a relatively short chain length (10–50 amino acid units) and are amphiphilic polymers with a cationic charge [301]. Understanding the mechanisms of action of these peptides [302,303,304,305] allowed the design of potentially similar counterparts, with a much simpler chemical structure, possible for relatively simple synthesis.

Intensive research has been begun on the synthesis of new synthetic aminopeptides (SAMPs), containing the skeleton of natural peptides (amino acids) and characterized by low toxicity [306,307,308]. Many of the obtained SAMPs show much stronger activity compared to their natural counterparts. For example, the synthetic version of AamAP1, AaP1-Lysine, showed 4 to 20 times higher activity against the same bacteria compared to its natural counterpart (scorpion venom) while showing much lower hemolytic and cytotoxic activities against eukaryotic cells [309].

Still, the optimal way to learn about this type of new peptides is the traditional method based on stepwise action: looking for the right natural source (microbial, animal), extraction and biological screening again of pathogens, selection of more perspective strains with scale up, and completing the research cycle through bioinformatic analysis and determining its chemical structure. The synthesis of the new peptide itself can be carried out by biotechnology with the prepared recombinants, most safely using eukaryotic cells of animal or human origin. An effective synthetic way of obtaining these compounds through solid-phase peptide synthesis has also been known for a long time, the principles of which are presented in Figure 9 [310].

The process proceeds in a stepwise manner by attaching the first amino acid chain to a solid, insoluble polymer (which can be, for example, polystyrene) by a covalent bond, then adding another group of amino acids—one at a time in a gradual manner. Until the desired sequence is obtained and, finally, the resulting peptide from the solid support. However, as can be seen, the main shortcoming of this synthetic approach is the requirement of many iterative reaction steps, leading to a limited final yield of the synthesis, limiting both the scale size and the length of the peptide chain that can be synthesized [311]. The biotechnological and synthetic way to obtain this type of peptide using solid-phase sequential synthesis is still expensive, which strongly limits the possibility of introducing these biomolecules into the market for production and sale.

It is necessary to look for the simplest methods of polypeptide synthesis that could remove the above-mentioned drawbacks by increasing the yield and efficiency of such production. One of the ways is to obtain peptides in ROP α -aminoacid-N-carboxy anhydrides (NCA) (Figure 9B). Chan-Park and his team, using obtained anhydrides of this type containing various substituents in the form of a chain of several amino acids (lysine and hydrophobic leucine, alanine, or phenylalanine residues), initiating the reaction with transition metal compounds, obtained a series of copolymers that varied the hydrophobic amino acid content from 0 to 100% [312]. In contrast to AMPs, the peptides obtained in this way have a chain made of repeating units, the sequence of which was difficult to control, and the material obtained showed a slightly high mass dispersion.

However, in recent years intensive research has allowed enhancing the efficiency of this synthesis to be improved, allowing polymers to present a broad spectrum of antibacterial properties, also suitable as antiseptic drug carriers [312,313,314,315]. Hammond et al. prepared the polymer by the ring-opening polymerization of γ-propargyl-L-glutamate N-carboxyanhydride, and the alkyne-azide cycloaddition click reaction (Figure 10), which mimics the favorable characteristics of naturally occurring antimicrobial peptides (AmPs).

They showed that the antibacterial activity of the obtained biomimetic polypeptides strongly depends on the presence of the pendant primary to quaternary amine groups and the construction of long alkyl spacers connecting these groups with the main chain. The optimal length of the main chain of the polypeptide is also crucial, depending on the type of functional group [316]. On the other hand, polymers containing cationic quaternary amines exhibited high mammalian blood hemolysis, which limited their application in the field of biosafety materials [317].

During NCA polymerization with the use of tetraalkylammonium carboxylate as the initiator of ROP, it became possible to obtain polypeptides with a multiblock chain structure, which allowed shaping the properties of the copolymer obtained, creating polypeptides with amphiphilic properties, similar to natural AMPs [314]. This one-pot synthesis of a polypeptide is insensitive to water and can proceed under ambient conditions to provide polypeptides with high molecular weight and narrow mass dispersity. Similar results were obtained by running the ROP of N-carboxyanhydrides catalyzed by crown ether [318].

Another solution is the use of epoxy compounds as ultrafast HCl scavengers for the moisture-tolerant synthesis of this kind of polypeptide [319]. A particularly excellent initiator of NCA polymerization is also sodium hexamethyldisilazide (NaHMDS) and potassium hexamethyldisilazide (KHMDS). The use of these compounds allowed for the acquisition of a polypeptide with a very long chain length (Mn = 2.5 × 10^5^ Da) and its controlled structure [320]. An interesting example of the use of ROP α -aminoacid-N-carboxyanhydrides in the construction of highly biologically active biomimetic polymers that are modified peptides is the results published by Du et al. [321]. As a result of the copolymerization of a mixture of Z-Lys-NCA and Phe-NCA anhydrides initiated with a macroinitiator obtained in the ROP reaction of ε-caprolactone with Boc-2-aminoethanol (Figure 11), a series of amphiphilic polypeptides were obtained. The produced peptide chain is composed of a hydrophilic block containing amino acids and aliphatic side chains terminated with primary amino groups and hydrophobic blocks built of caproyl units, too.

The copolymers obtained were used to build antibiotic carriers in the form of vesicles with an antibacterial activity designed to combat paradentosis. In vivo studies have confirmed the effectiveness of these systems in removing the biofilm created by bacterial strains. The β-peptides, another group of polypeptides, which have an extra methylene group in the backbone, due to high resistance to lysis by proteases, with consequent improved in vivo stability and usually non-mutagenic properties, are very interesting. They are most often produced by the anion-initiated ROP of β-lactams [322,323]. Liu [324], during the copolymerization of two types of lactams (one of which contained a blocked side amino group) series mimicking antibacterial poly-β-peptides, was obtained containing hydrophilic blocks with side cations +H_3_N and hydrophilic blocks containing alkyl side groups in the chain (Figure 12).

Although the obtained polymers had a low molecular weight (M_n_ about 4000 Da), the polypeptide contained 30 mol% hydrophobic units that display potent activities against all tested bacterial species (including drug-resistant) and low hemolysis and cytotoxicity. Based on the observation of the reasons for the occurrence of the antibacterial mechanism of action of polypeptides, inspired by their structure, the synthesis of polymers containing biologically active functional groups and with a specially designed chain structure that is not peptides was performed as well. The most essential criteria in the design of antimicrobial polymers are; the presence and maintenance of an appropriate balance between the cationic charge and hydrophobic content, the presence of the selected structure of the hydrophobic side chains, the selection of cationic groups and counteranions [325,326]. Palermo and Kuroda clarified the structure of the character of the cationic group in copolymers with varied content of hydrophobic alkyl groups to evaluate antimicrobial and hemolytic activities [13].

### 4.2. Polycarbonates and Carbonates Copolymers

Several polymers that mimic the amphiphilic structure seem to be a better approach than a polypeptide. They can be prepared readily, and their syntheses can be more easily scaled up. The first polymers synthesized that exhibited antibacterial properties, containing hydrophilic and hydrophobic chain segments similar to peptides, and cationic side groups were amphiphilic polymethacrylates [327]. They were obtained by radically initiated copolymerization of N-(tert-butoxycarbonyl) aminoethyl methacrylate and butyl methacrylate (Figure 13). These polymers are practically non-biodegradable, which considerably limits their use in medicine.

Kuroda [328] with the team showed that it is possible to obtain biodegradable acrylic polymers. He synthesized polyacrylates containing ester bonds in the main chain and side amino groups. These random copolymers are capable of biodegradation and are also assisted by spontaneous intramolecular amidation. During the synthesis, this team used simultaneous chain and step-growth radical polymerization of t-butyl acrylate and 3-butenyl 2-chloropropionate, followed by the transformation of t-butyl groups into primary ammonium salts. The authors also confirmed the antibacterial effect of the polyacrylates obtained. However, it seems that chain structures ensuring amphiphilicity and the presence of cationic side groups can be accomplished much more easily using the ROP mechanism of cyclic carbonates or lactides.

The aliphatic polycarbonates are bioresorbable and are variants of previously synthesized amphiphilic polymethacrylates containing pendant groups. They are produced in the ROP reaction of aliphatic cyclic carbonates or their copolymerization with lactides or lactones. Yang and Hedric [329,330], with the team, were the first to publish the properties of bioresorbable triblock polycarbonates obtained by sequential ROP of the MTC–(CH_2_)_3_Cl monomer (3-chloropropyl-5-methyl-2-oxo-1,3-dioxane-5-carboxylate) followed by trimethylene carbonate (TMC) polymerization initiated from a diol in the presence of a mixture of the Lewis acid (Figure 14).

Using the synthesized copolymers, the authors formed micelles, which were used in research on antibacterial activity. The obtained results show that the nanoparticles disrupt microbial walls/membranes selectively and efficiently, thus inhibiting the growth of Gram-positive bacteria, methicillin-resistant *Staphylococcus aureus* (MRSA), and tested fungi. Importantly, no significant hemolysis was observed in contact with the micelles, even at relatively high concentrations of this material.

Another intriguing illustration of the possibilities of this method is an attempt to obtain polycarbonate copolymers containing long side chains by the reaction of forming graft copolymers [330,331]. The copolymers were produced in a two-step process. The first stage was the copolymerization of cyclic carbonates with side groups (MTC-OEt and MTC-OBn holding a blocked hydroxyl group). After the polycarbonate, the side hydroxyl groups were deprotected by hydrogenation. In the next step, the manufactured polyol was used as an ROP macroinitiator of L-lactide and trimethylene carbonate polymerization. As a result, long-side esters were formed and then terminated in reaction with MTC-OBnCl cyclic carbonate. The end chloride groups were subjected to the diamine quaternization procedure (TMEDA) were subjected (Figure 15).

Many groups have researched the synthesis of antibacterial polymers using cyclic carbonate monomers. Various side cationic groups, not only quaternized amine groups, were introduced in the obtained polymers. Jin and Swift [332,333] synthesized polycarbonates containing guanidynyl groups. The variation in the amphiphilicity of guanidinylated polycarbonates was obtained by manufacturing a series of copolymers of cyclic 5-methyl-5-prop-argyloxycarbonyl-1,3-dioxan-2-one (MPC) and 5-methyl-5-benzy-loxycarbonyl-1,3-dioxane-2-one (MBC) using methylbenzyl alcohol as an initiator. Then, to introduce guanidinyl groups, a post-synthesis modification of alkyne-containing polycarbonates by azide-alkyne cycloaddition click chemistry (CuAAC) using Cu(I)-catalyzed was performed. The course of the synthesis is shown in Figure 16.

The authors observed that these guanidine polycarbonates exhibited broad-spectrum biocidal activity with low toxicity to red blood cells (RBC). Polycarbonate of the lowest molecular weight (approximately 8000 Da) showed the best antimicrobial activity and the least RBC toxicity (0.6% hemolysis at MIC). It also turned out that optimizing the length of the arms of the spacer between the guanidine group and the polycarbonate main chain is an efficient design strategy to alter the hydrophobic/hydrophilic balance without changing the cationic charge density. A six-methylene unit spacer arm shows the best antimicrobial activity with low hemolytic activity. Similar functionalized polycarbonates have also been obtained [333], as well as polyethylene glycol comb with polyquaternium cation arms [334].

Another way to obtain amphiphilic polycarbonates with pendant cationic groups is through the polymerization of cyclic carbonates containing pendant bromoalkyl groups [14]. The manufactured polycarbonate with bromide-terminated side groups is then subjected to the quaternization procedure. The authors conducted the quaternization of manufactured polycarbonates with various nitrogen-containing heterocycles, such as imidazoles and pyridines. The final polymers showed a wide spectrum of activity against Gram-positive as well as Gram-negative bacteria. The hemolysis induced by the polycarbonates also presents a high selectivity toward the tested microbes over the mammalian red blood cells.

Similarly, polycarbonates containing pendant primary amino groups can be obtained by copolymerizing cyclic carbonates with protected pendant amino groups. After the final polycarbonate is obtained, the amino groups are deprotected by acid hydrolysis [335] (Figure 17).

Chen et al. [336] obtained a series of copolymers with a pendant amino group by ROP initiated with a tin compound during the copolymerization of L-lactide and specially synthesized cyclic carbonate ((2-nitrobenzyl) amino)-1,-3-dioxan-2-one (NBAC). The use of carbonate with a nitrobenzyl group allowed the troublesome process of deprotection of functional groups. The 2-nitrobenzyl group has been used as a photoremovable protecting group in the final stage of synthesis. The UV deprotection method used was very efficient and resulted in virtually no final decrease in average molecular weight. The copolymer contained hydrophobic blocks of lactide units and hydrophilic blocks with amino pendant carbonate groups. Compared to poly(L-lactide), a copolymer with a content of 10 mol.% carbonate units with pendant amino groups showed a pronounced bactericidal effect.

Polycarbonates containing amino cations were also successfully obtained directly in the ROP of aliphatic N-substituted eight-membered cyclic carbonates initiated with benzyl alcohol and in the presence of DBU (Figure 18) [337].

The presented examples show that using the ROP, it is relatively easy to obtain cationic linear or branched copolymers with a variety of planned chain structures, which have hydrophilic and hydrophobic segments in their structure. The main difficulty in the synthesis of these compounds, limiting to a large extent the possibilities of this synthesis, is related to the stage of obtaining the appropriate structure of cyclic monomers active in polymerization.

### 4.3. Antibacterial Biodegradable Polyesters

Aliphatic biodegradable polyesters with antibacterial properties are obtained basically in two ways. The first of them is the ROP of lactones or lactides, and the second is a gradual polymerization, most often polycondensation or polytransesterification of dialcohols and aliphatic diacids or their esters.

Saha et al. obtained an infection-resistant coating on the surface of biodegradable polyester [338]. This polyester was obtained by the polycondensation reaction of a hydroxyl-protected tartaric acid derivative with 1,6-hexanediol (Figure 19). The resulting polyester was then subjected to the procedure of deprotection of hydroxyl groups, obtaining a linear aliphatic polyester containing pendant hydroxyl groups with the chain in about 80% yield. These reactive groups were used to create cationic polymer brushes on its surface by polymerization of RAFT of 2-(Methacryloyloxy) ethyl tri-methylammonium chloride (META). The authors showed a strong antibacterial effect of the surface produced in this way (Figure 19).

A similar cationic polyester was synthesized by Yan et al. [339]. Poly(butylene succinate) obtained in the first stage of the synthesis by traditional polycondensation was then subjected to a bromination reaction by the addition of bromine to carbon-carbon double bonds. The product of this reaction, brominated poly(butylene succinate) (BPBS), was used in the quaternization reaction with a series of N,N-dimethyl alkylamines afforded poly(butylene succinate) containing quaternary ammonia cations with various ion contents and alkyl chain lengths. Finally, a series of biodegradable polyesters with alkyl side chains of various lengths containing quaternary amino groups were obtained. The authors showed that polyesters showed strong antibacterial properties when the alkyl chain length was greater than 8.

In turn, Tiller and the team [340] used 1,4-dibromobutene as a monomer in a polycondensation reaction with a series of previously synthesized tertiary diaminodiesters (Figure 20). The resulting biodegradable polyionenes esters are antibacterially active against a wide range of bacterial strains. They do not induce hemolysis, and they are toxic to human mesenchymal stem cells.

Another interesting example of the synthesis of antibacterial polyesters using the ROP mechanism is the method described by Kalelkar [341], which presents the way to obtain polylactide copolymers substituted with pendant azide groups. In the first stage, during the copolymerization by opening the ring of L-lactide and chloromethylglycolide, a random copolymer with a chain made of lactidyl and methylchloroglycolidyl units is obtained. Then, this copolymer was subjected to one-pot dehydrochlorination and thiol addition. The copolymer formed containing azide groups was finally subjected to azide-alkyne click chemistry (Figure 21).

The resulting polylactide copolymer with azide groups was selected to formulate polycationic antibacterial materials. The incorporation of trioctylammonium substituents presents the creation of prospective biodegradable materials with antimicrobial activity against two Gram-negative bacterial strains with just only 5 mol.% of the copolymer chain units with the cationic group. Other biodegradable copolyesters-poly(ester-*co*-phosphoester)s were also obtained by ring-opening copolymerization of ε-caprolactone and cyclic (2-chloroethoxy)-2-oxo-1,3,2-dioxaphospholane [342]. The obtained copolymer was then subjected to the process of quaternization with tertiary amines, finally obtaining cationic polyester phosphoesters (Figure 22). Again, the copolymers with an elongated alkyl group with quaternary ammonium cation displayed good antibacterial activity against both Gram-negative and Gram-positive bacteria.

### 4.4. Polyamines and Polyesteramines

Polyamines are polycationic molecules that are involved in numerous cellular functions in both eukaryotes and bacteria [343]. Studies suggest a close link between polyamines and cell proliferation. Much effort has been put into developing drugs capable of acting on polyamine-mediated biochemical pathways. These compounds are generally tested as gene carriers in gene therapy [344]. These compounds usually present a strong antibacterial effect [345,346,347]. Laabei and a team based on natural polyamines such as spermine and spermidine synthesized several linear polyamines [345]. The authors have shown that manufactured polyamines can be used as scaffolds for the design of novel antimicrobial compounds that are effective against *S. aureus*. It is particularly interesting some of the obtained materials show a clear tendency to enhance antibiotic action and restore the sensitivity of MRSA and VRSA isolates to daptomycin, oxacillin, and vancomycin. A large part of the synthesized polyamines was intended for forming gene carriers; the authors did not verify the antibacterial properties of the obtained polymers. The disadvantage of high-molecular-acid synthetic polyamines obtained in this way is their slow degradation.

Saltzman et al. obtained a large series of polyesteramines differing in chain structure in the lactone ROP reaction of lactones (ε-caprolactone, 12-dodecanolide, 15-pentadecalanolide, 16 hexadecanolide), carried out in the presence of amino-substituted diol (methyl diethanolamine) as initiator and oligodiesters (diethyl sebacate). The reaction was enzyme-catalyzed (immobilized *Candida antarctica* lipase B) [348,349]. These types of polymers, which contain ester bonds in the main chain, are biodegradable materials.

The antibacterial activity of polyamines is also evidenced by the results published by Meng et al. This team prepared a positive charge polyamine cyclophosphazene membrane by interfacial polymerization of polyethyleneimine and hexa-chlorocyclotriphosphazene [350]. The tests that were passed indicated that the membrane had a bacterial inhibiting rate against Gram-negative *E. coli* and Gram-positive *S. aureus.* In a concentrated suspension, all bacteria were eliminated very quickly, in about 20 min, upon contact with this material.

## 5. Biodegradable Polymer Composites with Antibacterial Properties

Biodegradable polymer matrix composites are a wide group of materials that have been used, among others, in medicine, pharmacy, or materials in contact with food. Loading polymers with metal particles, metal oxides, drugs, or other compounds can improve their antibacterial properties. These composites can be in the form of films, fibers, particles, membranes, and coatings [351]. Some of the polymer composites are described below and shown in Table 4 with their antibacterial activity. The antibacterial activity of metals and metal oxides has been confirmed to proceed through several mechanisms, such as metal binding to the bacterial reaction with intracellular proteins [352] and interference with nutrient assimilation [353]. A simplified operation of the antibacterial mechanisms of metal particles is shown in Figure 23.

Metal binding to the bacterial cell wall is the result of a positive charge of the metal ions and a negative charge of the cell components. Metals bind to the cellular wall electrostatically and disrupt its potential. This leads to disruption of the integrity of the internal structures of the bacterial cell, resulting in the release of water from the cytosol. The imbalance of ions and instability of the cell membrane causes significant difficulty in respiration and, ultimately, cell death [352,354,355]. This mechanism occurs in the case of Au [354] and Ag_2_O [356]. The antibacterial activity of metal particles and metal oxides also depends on the type of bacteria. Gram-negative bacteria are generally more sensitive to mechanical damage than Gram-positive bacteria due to significantly thinner cell walls [355].

Metal particles indirectly induce the production of ROS, which increases the oxidative stress response in cells. This disturbs the balance of ROS in the bacterial cell, and the concentration of ROS is no longer controlled by the cell. This leads to damage to the proteins, lipids, and DNA of the bacteria, resulting in their death [355,356]. The antibacterial activity of TiO_2_ [354], MgO [355] Ag_2_O [356], and Al_2_O_3_ [357,358], was confirmed to follow this mechanism.

The interaction with intracellular proteins involves the penetration of metallic nanoparticles into the cell, penetrating the bacterial cell wall. Inside the cell, the nanoparticle reacts with proteins, nucleic acids, and enzymes, which can lead to the inhibition of replication [352,355]. The reaction with phosphoric acid residues in DNA is associated with the genotoxicity of the molecules. This mechanism occurs, for example, in the case of Ag_2_O nanoparticles [356].

Metal and metal oxide nanoparticles can inhibit bacterial cell growth by reacting with bacterial nutrients. This leads to blocking the uptake of nutrients by bacteria. An example of this mechanism is the uptake of Fe by Ga in *P. aeruginosa*. Thus, Ga indirectly affects the deregulation of gene expression responsible for iron uptake in *P. aeruginosa* [353].

A large group of metallic particles with antibacterial properties simultaneously uses several mechanisms of antibacterial action. Examples of such particles are Al_2_O_3_ [359] and Ag_2_O [356] NPs. The size of the particles also affects antibacterial activity. Nanoparticles have a higher surface area-to-volume ratio than microparticles, which makes them more reactive and easier to penetrate microbial membranes. Adjusting the pH by knowing the point of zero charges can also enhance the antibacterial activity of metal oxides. At pH below zero charges, particles are positively charged and show higher antibacterial activity than neutral or negatively charged particles [360].

Poly(lactic acid) (PLA) is a biodegradable and bioresorbable polymer that is used in medicine for implants. Due to its biodegradability, it is used as a composite matrix with the addition of antibacterial oxides and metals, such as TiO_2_ and ZnO. PLA is also easily processable. PLA degrades to form nontoxic substances such as lactic acid, water, and carbon dioxide through hydrolytic mechanisms and cellular oxidation.

PLA/ZnO composites can be used in wound dressings, regenerative tissue applications, and functional coatings. Increasing the concentration of ZnO increases not only the antibacterial properties of the composite but also its cytotoxicity [361].

PLA fiber mats containing Al_2_O_3_-Ag nanopowder are a composite proposed for antibacterial applications. Fibrous mats prepared by electrospinning showed antibacterial activity against *E. coli* and *S. lutea*. The tests were carried out on Al_2_O_3_/Ag powder concentrations of 5%, 25%, and 50%. For *E. coli*, 8.87, 10.55, and 11.08 mm diameter of zone growth inhibition was obtained, and for *S. lutea*, 13.69, 13.88, and 15.08 mm. Additionally, tests were carried out for PLA mats without the addition of powder, and a 6.35 mm diameter of zone growth inhibition was obtained. This means that antibacterial activity increases with the concentration of the Al_2_O_3_/Ag powder [362].

PLA/Ag nanofibers were fabricated by an electrospinning technique in which PLA with 2% Ag was found to have optimal properties such as strength and wettability for dressing applications. Furthermore, the antibacterial properties of the composite against *E. coli* and *S. aureus* have been proven [363].

PLA/TiO_2_ film has been used as a material used for wound dressings. Its antibacterial properties have been demonstrated with *S. aureus* (ZOI = 28 mm). However, PLA itself did not show antibacterial activity [364].

The antibacterial effect of PLA is also achieved by applying an appropriate initiator of the ROP of lactide already in the polymerization stage [365]. Low-toxic zinc complexes such as Zn[(acac)(L)H_2_O], where L represents N-(pyridin-4-ylmethylene) tryptophan or N-(2-pyridin-4-ylethylidene) phenylalanine were used as initiator during bulk polymerization of L-lactide. Using an initiator/monomer molar ratio of 1:400, polylactides with a number average molecular weight of about 40,000 g/mol were obtained. The polymers did not show cytotoxicity but displayed a strong antibacterial effect, especially on strains of *S. epidermidis*, *S. aureus*, and *P. aeruginosa*. For these bacteria, the determined value of MIC was approximately 100 µg/mL.

Polyvinyl alcohol (PVA) has been used for drug delivery applications such as wound dressing and scaffolds. PVA is biodegradable, has adhesive properties, is biocompatible, and is easily processable. The PVA characteristics also include good chemical resistance and thermal stability. PVA/ZnO nanocomposite fibers are used in wound dressings due to their nontoxicity and antibacterial properties. The antibacterial activity of the composite was compared to a PVA polymer filled with the antibiotic ampicillin. In the case of the PVA/ZnO composite, the MIC antibacterial activity against *E. coli* and *S. aureus* was 62.5 μg/mL and 250 μg/mL, respectively, while in the case of the PVA/ampicillin composite: The MIC was 4 μg/mL and 250 μg/mL for *E.coli* and *S. aureus*, respectively [366]. CuS/polyvinyl alcohol-chitosan nanocomposites are also characterized by antibacterial activity, which has been proven in the tests against *E. coli*, *S. aureus*, *P. syringae*, and *S. pneumoniae* bacteria [367].

The antibacterial activity of selenium has been used in PVA composites. An example of such a composite is PVA/Chitosan/SeNPs nanocomposite films prepared via a one-pot laser ablation route. The addition of selenium has been shown to increase the antibacterial properties of the PVA/chitosan blend against *S. aureus, B. subtilis, E. coli*, and *P. aeruginosa*. The composite has been shown to have greater antibacterial properties against gram-negative bacteria (activity index against *E. coli* totals 69%, activity index against *P. aeruginosa* totals 64%) than against gram-positive bacteria (activity index against *S. aureus* totals 54%, and against *B. subtilis*—44%). It is caused by better penetration of selenium ions through the thin cell wall of Gram-negative bacteria. In addition, antibacterial properties have been shown to increase with laser ablation time (up to 30 min) [368]. Another example of a composite with selenium particles is PVA/carboxymethyl cellulose loaded with selenium nanoparticles. An increase in antibacterial activity was also shown with increasing concentration of selenium nanoparticles. The antibacterial activity of the composite was tested on *E. coli*, *S. aureus*, *P. aeruginosa*, and *B. cereus* bacteria. The blend without the addition of selenium did not show antibacterial properties, while the composite with 0.8% SeNP showed the highest antibacterial activity (at the level of 98%) against *S. aureus* and *B. cereus* bacteria (78%) in the case of *E. coli* bacteria at the level of 66% and *P. aeruginosa*—16% [369].

Polycaprolactone (PCL) is a biodegradable polyester used in long-term implants and drug delivery systems. PCL/Ag membranes increase the strength of the material and also show excellent antibacterial activity against *E. coli* (inhibitory zone diameter 7.9 ± 0.6 mm) and *S. aureus* (inhibitory zone diameter of 11.6 ± 0.5 mm), while neat PCL shows antibacterial activity with an inhibitory zone diameter of 6.0 ± 0.0 mm against *E. coli* and *S. aureus* [370]. PCL/ZnO scaffolds can be used for bone tissue engineering. The higher the ZnO content, the higher the antibacterial properties of the composite, but also the addition of ZnO particles to the polymer can also affect the formation of mechanical stresses [371].

Poly(lactic-co-glycolic acid) (PLGA) is a biocompatible and biodegradable polymer. PLGA is synthesized by the copolymerization of two monomers, the cyclic dimers of glycolic acid and lactic acid. PLGA is used in drug delivery devices or dental membranes for guided tissue regeneration. PLGA/Ag/ZnO nanorods composite coating shows moderate antibacterial properties against *E. coli* and *S. aureus* [372]. PLGA nanoparticles loaded with azithromycin show excellent antibacterial properties against *Salmonella typhi* (MIC = 3.12 µg/mL) regardless of drug ratio, while the minimum inhibitory concentration of pure azithromycin is 25.00 µg/mL [373]. The PLGA/Al_2_O_3_ composite is proposed as a potential material for food packaging due to its controllable mechanical properties and antibacterial properties proven on *E. coli* bacteria. Increasing the concentration of Al_2_O_3_ in the composite increased the antibacterial activity. A concentration of 0.1% Al_2_O_3_ NPs reduced the growth of *E. coli* bacteria to 74%. It has also been proven that the presence of the composite in the soil does not affect the growth of plants such as cucumbers (*Cucumis sativus*) and tomatoes (*Solanum lycopersicum*) [358]. The PLGA/Fe_2_O_3_ composite has been proposed as a potential packaging material in agriculture. Its antibacterial activity has been demonstrated in the example of *E. coli* bacteria. An increase in the concentration of Fe_2_O_3_ nanoparticles increased the reduction of bacterial cells (the concentration of 0.01% Fe_2_O_3_ NPs resulted in a more than 5-fold reduction in the number of *E. coli* cells), while PLGA without the addition of Fe_2_O_3_ NPs did not show antibacterial activity. It was also proved that the presence of the PLGA/Fe_2_O_3_ composite did not affect the growth of peppers (*Capsicum annuum*) [374].

Chitosan is a non-toxic, biodegradable, and biocompatible polymer that can inhibit microbial growth. Chitosan/ZnO nanocomposite films can be used in food packaging. The study showed that the addition of ZnO nanoparticles improved the mechanical and antibacterial properties of chitosan [140]. Furthermore, ZnO nanoparticles embedded in TPP-cross-linked chitosan appeared very efficient in bacteria removal (5 log reduction for *E. coli* and complete reduction for *S. aureus*). At the same time, the chitosan matrix inhibited the photogeneration of ROS that can be produced by ZnO. The material has been proposed to be used as an effective and safe sunscreen agent [375].

The chitosan-pectin-TiO_2_ dressing material shows excellent antibacterial properties against *E. coli*, *S. aureus*, *P. aeruginosa*, and *B. subtilis* bacteria; the inhibition zone exceeds 45 mm against any bacteria. This material can absorb moisture from the wound without dehydrating it; it also shows faster wound healing compared to chitosan alone [376].

AgNPs suspended in the porous chitosan matrix can be considered as potential wound dressing material. Nanocomposite sponges show antibacterial properties against *E. coli* and *S. aureus*. After 2 h of exposure, sponges with 15 mM showed 6.00-log reductions of viable bacteria cell numbers, while the chitosan sponge alone showed antibacterial activity against *E. coli*, 0.32-log reductions of viable cell numbers, and 0.96-log reduction against *S. aureus*. An increase in the concentration of AgNPs increases the antibacterial properties of the composite [377].

The Ag_2_O/chitosan nanocomposite film has been proposed as a material for food packaging applications. The antibacterial activity of the composite has been proven in tests against Gram-positive (*B. subtilis, S. aureus*) and Gram-negative (*E. coli* and *P. aeruginosa*) bacteria. The study showed that the film solutions exhibited higher antibacterial activity than the film itself. The inhibition zones for the composite solution against *B. subtilis, S. aureus, E. coli*, and *P. aeruginosa* were 20 mm, 23 mm, 16 mm, and 24 mm, respectively [378].

Chitosan-coated FeO nanocomposite has been presented as a potential bioflavonoid nanomaterial in the biomedical and pharmaceutical industries. The study examined antioxidant activity using the DPPH method. The higher antioxidant activity of the CS/FeO nanocomposite compared to that of FeO nanoparticles has been proven. Antibacterial activity was demonstrated based on studies of *B. subtilis, S. aureus*, and *E. coli*. The zone of inhibition increased with the increase in FeO concentration in the composite, and for the concentration of 40 µg/mL, it was 13 ± 0.5 mm, 12 ± 0.5 mm, 15 ± 0.5 mm for *B. subtilis, S. aureus*, and *E. coli.* bacteria, respectively [379].

Chitosan-capped gold nanoparticles can be considered potential antibacterial materials. The antibacterial properties of the composite have been proven against *E. coli*. *S. aureus*, *S. enterica*, and *L. monocytogenes* bacteria [380].

Tobramycin-chitosan nanoparticles (TOB-CS NPs) coated with zinc oxide nanoparticles (ZnO NPs) are composites that can be used in drug delivery systems. It shows excellent antibacterial activity against *E. coli* and *S. aureus* (MIC = 8.40 ± 0.11 mg/mL against *E. coli* and MIC = 10.70 ± 0.08 mg/mL against *S. aureus*). Tobramycin–chitosan nanoparticles without ZnO coating have worse antibacterial properties than composite with ZnO (MIC = 11.30 ± 0.12 mg/mL against *E. coli* and MIC = 15.60 ± 0.09 mg/mL against *S. aureus*) [381].

Cellulose is a natural polysaccharide built linearly from D-glucose molecules. It is a biodegradable, nontoxic polymer, insoluble in water, with the ability to absorb water. It has found wide applications as a food packaging material. Cellulose fibers are also used for water filters due to their viscosity and reactivity with many elements. The addition of metal particles or metal oxides to cellulose increases its antibacterial properties, so metal-cellulose composites can be used as antibacterial materials [351].

Cellulose/chitosan composite films loaded with ZnO nanoplates show antibacterial activity against *E. coli* and *S. aureus* bacteria. The antibacterial activity of the composite increases with increasing ZnO concentration. It is the highest for 3% ZnO and reaches the value of zone of diameter 13.6 ± 0.2 mm against *E. coli* and 20.6 ± 0.2 mm against *S. aureus*. Composite also shows excellent photocatalytic properties [382].

Spherical-nano cellulose (SNC)/silver-nanoparticle (AgNP) composite also shows antibacterial (against *E. coli* and *S. aureus* bacteria) and catalytic properties. The material can be considered a potential biomass-based antibacterial agent [383].

Cellulose nanofibers aerogels functionalized with AgO (CPS-AgO) are proposed for applications requiring high antibacterial activity. The antibacterial effect of the composite has been proven on *E. coli* and *S. aureus* bacteria. An increase in antibacterial activity was observed with increasing concentration of AgO NPs, to a concentration of 0.2%, at which the maximum inhibition zone diameter was reached. After this concentration, no further increase in antibacterial activity was observed. The maximum inhibition zone diameter against *E. coli* was 23 mm and 20 mm against *S. aureus* [384].

The article presents research on cellulose fiber composites covered with Al_2_O_3_ and ZnO. Cotton and viscose were used as cellulose fibers. The study confirmed the antibacterial activity of both Al_2_O_3_ and ZnO composites against *E. coli* and *S. aureus* bacteria. However, ZnO cellulose fibers showed about 5–10% higher inhibition rate than Al_2_O_3_ composites. Composites with both types of oxides showed higher antibacterial activity when viscose fibers were used compared to cotton fibers. The inhibition rate of the viscose/Al_2_O_3_ composites was 71% (*E. coli*) and 65% (*S. aureus*), and for the viscose/ZnO composites it was 78% (*E. coli*) and 75% *(S.aureus*) [385].

Dextran is a polymer obtained by the polycondensation of glucose. The Dextran/AgNP composite in the form of a thin film can be used in food packaging applications. Its antibacterial properties have been proven against *E. coli*. Additionally, dextran/AgNP film is characterized by improved mechanical properties and is biodegradable [386].

Alginate (alginic acid) is a biodegradable polysaccharide, the copolymer of mannuronic acid and guluronic acid, found in brown algae. Alginate has high water absorption and the ability to form a hydrogel, so it is used as a stabilizer. Alginic acid is also used for wound dressing applications [387].

ZnO nanoparticles dispersed in alginate (~11 wt.% ZnO) show high antibacterial activity against E. coli and *S. aureus* bacteria, with a 100% reduction in *E. coli* and a 99.9% reduction in *S. aureus* after 2 h of exposure [388].

Starch is a biodegradable polysaccharide. Starch is used in the pharmaceutical industry as a filler for drugs. Starch-Albumin-magnesium oxide (S-A-MgO) composite film also has antibacterial properties. Antibacterial activity has been proven against *E. coli* and *S. aureus* bacteria. The study compared the properties of the composite A-MgO, S-MgO, S-A-MgO, S-A, pure albumin (A), and pure starch (S). It has been shown that the best antibacterial properties have S-A-MgO composite (DI = 5 ± 0.13 mm against *E. coli* and DI = 6 ± 0.22 mm against *S. aureus*). Moreover, materials without MgO particles have no antibacterial effects [389].

Starch-capped Ag_2_O nanoparticles are a promising antibacterial material. During the study, a decrease in the size of the nanoparticles was observed with increasing starch concentration. The antibacterial activity of the obtained composite was proven on *E. coli* and *S. aureus* bacteria. With the increase in the concentration of Ag_2_O NPs, the antibacterial activity of starch-capped Ag_2_O increased. For the concentration of 6 mg/mL, the zone of inhibition was 14 ± 0.11 mm against *E. coli* and 15 ± 0.19 mm against *S. aureus* [390].

Polyaniline/starch/hematite composite (PANI/starch/Fe_2_O_3_) is a potential material for industrial water purification applications. The ability of the composite to adsorb heavy metals (As^3+^, Zn^2+^, and Co^2+^) was tested. At low concentrations of heavy metals (5 mg/L), the efficiency was 100% for As^3+^ and Zn^2+^ and 91% for Co^2+^. As the heavy metal concentration increased, the adsorption efficiency decreased. The antibacterial properties of the composite have been proven in *S. typhimurium* (IZ = 18.02 mm), *S. aureus* (IZ = 13.98 mm), and *B. subtilis* (IZ = 14.12 mm) bacteria [391]. Data on the antibacterial activity of biodegradable polymer composites are summarized in Table 5.

## 6. Application of Biodegradable Polymers and Antibacterial Polymer Composites in the Food Industry and Agricultural Technology

Antimicrobial packaging is designed to incorporate antimicrobial substances into the packaging system to delay or prevent microbial pathogens’ growth during postharvest transportation, storage, and retail display of the product. To this end, several types of biodegradable polymers have been used, often modified with bioactive additives (Figure 24). Antimicrobial packaging provides longer storage possibilities and an extension of the shelf life of the products, making them safer for customers [396]. Unfortunately, poor water, vapor, and gas barrier properties are important critical issues in packaging [397]. For this reason, many approaches have been applied to develop antimicrobial packages using various techniques of packaging modification. Incorporation of antimicrobial agents can be performed by several techniques, including encapsulation, direct addition, coating, immobilization of antimicrobials in polymers by ion or covalent linkages, or grafting in or onto the matrix [398,399].

According to Sedlarik (2013) [400], methods for antimicrobial polymer modification determine advantages and disadvantages. Modification of polymer surface properties without an antimicrobial agent protects against the use of additional chemicals but, unfortunately, is ineffective and technologically demanding. Direct deposition of the antimicrobial agent on the polymer surface is fast, cheap, and simple, but its efficiency is low. Chemical deposition of the antimicrobial agent on the polymer surface determines a low amount of active agent on the surface, and, unfortunately, this method is expensive and technologically demanding. Direct incorporation of the antimicrobial agent in the polymer matrix is simple and relatively easy to process using classic techniques; however, the disadvantage of this method is low efficiency, application of a high amount of antimicrobial agent, and limitations coming from the temperature sensitivity of the antimicrobial agent during processing.

Regardless of the method of modification, more research and development are needed to improve the antimicrobial efficiency of current biodegradable packaging. The perspective of this issue is to use more effective natural antimicrobial compounds capable of maintaining their activity during processing technologies. However, the challenge is to combine three characteristics of antimicrobial biodegradable polymer packaging, namely (1) that it has to be safe for the food (cannot release any contaminants), (2) that it should have proper mechanical and use properties, and simultaneously (3) meet the biodegradability criteria after use. In other words, such packaging has to be safe for health and environment.

Metallic nanoparticles [401], oxide nanoparticles [402], clay nanoparticles [403], chelating agents [404], volatile gas form, plant extracts [405], plant essential oil plants [406], bacteriocin [407,408] or inorganic NPs [409], have already been used as antimicrobial agents.

Unfortunately, they are susceptible to migrating from packaging to food depending on their concentration, size, shape, and dispersion. Migration is also determined by environmental factors such as temperature, mechanical stress, food condition including composition and pH, polymer properties, and duration of contact. When they intoxicate the human body, they can potentially cause adverse health and environmental effects, such as intracellular damage, pulmonary inflammation, or vascular disease [410,411].

### 6.1. Polysaccharides Composites

Polysaccharides are increasingly being applied as novel packaging materials due to their wide abundance, and easy availability, and nontoxicity also exhibit excellent CO_2_ and O_2_ barrier properties, which can not only retard the respiration rate of food but also potentially inhibit the growth of bacteria and molds, that cause food spoilage, within the packaging.

#### 6.1.1. Starch

Starch is one of the most abundant and renewable plant polysaccharides, occurring naturally in various sources such as potatoes, corn, wheat, and tapioca. It consists of two types of glucose polymers: amylose is a chain of D-glucose units connected by a-1,-4 bonds, and amylopectin contains short chains of a-1,-4-linked D-glucose units that are branched by a-1,-6 bonds [412].

Despite the many advantages of starch, such as the ease of modification and creation of films with low oxygen permeability and low production costs, the main challenge that limits its use is that its film properties are highly dependent on moisture content, exhibiting relatively low mechanical resistance [413]. Furthermore, because of the high melting point and lower thermal decomposition temperature of starch, it has poor thermal processability. For this reason, the use of starch in the production of plastic bags and food packaging is difficult. In order to make starch suitable for targeted applications, it is necessary to blend it with other polymers or plasticizers. Application as packaging materials, starch blends with polymers which have enhanced water resistance and good mechanical properties, is not only economical but also safe for the environment [414,415].

Starch and starch derivatives have many characteristics compatible with many antimicrobial agents (including essential plant oil, chitosan, and antimicrobial peptides), and the resulting starch films demonstrated the ability to deactivate many species of pathogens [416].

In research, Pyla et al. (2010) studied starch-based films impregnated with tannic acid for the inhibition of *E. coli* O157:H7 and *L. monocytogenes*, popular foodborne pathogens. The films had strong antimicrobial activity in both *L. monocytogenes* and *E. coli* O157:H7 [417].

To develop a bioactive starch-based film, the association of starch with antimicrobial polymers such as aminopolysaccharides could be promising.

Vasconez et al. studied edible antimicrobial edible coatings based on blends of starch-chitosan with or without the addition of potassium sorbate. They showed that the addition of chitosan reduced the solubility of starch films and the water vapor permeability, and the antibacterial action depended on the processing techniques and conditions used [418].

In similar studies, Shen et al. (2010) have also shown that *S. aureus* could be effectively suppressed by incorporating chitosan 10% in a starch-based film [419].

In research, Meira et al. (2017) have incorporated various antimicrobial peptides, nisin, or pediocin, into the starch-halloysite nanocomposite films. Films had improved thermal stability, mechanical strength, and water barrier properties and showed high antimicrobial activity against *L. monocytogenes* and *C. perfringens* [420].

Furthermore, systems based on TPS/layered silicates containing essential oils showed the desired antimicrobial activity with no harmful effect on food [421].

The antimicrobial packaging film was prepared through melt blending of antimicrobial thermoplastic starch (ATPS) with linear low-density polyethylene (LLDPE). Polyhexamethylene guanidine hydrochloride was blended with starch, and the resulting LLDPE/ATPS films demonstrated excellent antimicrobial activity against *E. coli*. Furthermore, the antimicrobial property and biodegradation behavior of poly(butylene adipate-co-terephthalate) (PBAT)/TPS films before and after a three-month soil burial test have been studied. The results demonstrated that starch and PBAT were biodegradable, and TPS showed excellent growth inhibition against *E. coli* [422].

Barzegar et al. (2014) investigated the influence of potassium sorbate (PS) on the physical and antimicrobial properties of starch-clay nanocomposite films. For this purpose, starch-clay nanocomposite active packaging films containing PS were prepared, and their mechanical, physical, and antimicrobial properties were evaluated. The antimicrobial properties of the films were evaluated using *A. niger*. It was found that the addition of PS improved the water vapor permeability of the films and elongation at the break while resulting in a decrease in their tensile strength. Furthermore, antimicrobial activity was improved by increasing the amount in the film formulation [423].

Romaino et al. (2022) evaluated the potential application of starch-citrate film as an antimicrobial and antifungal packaging film. The starch-citrate film could suppress even 98–99% of foodborne bacteria and 87–99% of fungal growth. Additionally, it can be observed that starch-citrate film was able to extend the shelf life of food longer (10 days longer for cake and 40 days longer for bread) than the shelf life of food paced with commercial food packaging [424].

In conclusion, the effectiveness of starch-based antimicrobial packaging materials depends on the number of antimicrobial agents loaded in the polymer matrix and the optimal amount released over time. The performance of the film is usually dependent on the materials used and the concentration of additives. However, the number of additives incorporated into the starch matrix should be properly controlled, either for a particular application or to minimize the negative impact on the environment.

#### 6.1.2. Chitosan

As described in Section 3.1, chitosan [425] is a polymer that, due to its special antimicrobial activity against various microorganisms, has found wide application in many fields, such as pharmaceutical, cosmetic, textile, industrial wastewater treatment, agriculture, and, above all, the food industry.

The solubility of chitosan in weak acids is the key to its film formation ability. It creates durable films and membranes produced by extrusion and pressing, as well as coating and nanoparticles, that can be used as food packaging and edible packaging. When used as ecological food packaging, chitosan films can extend shelf life and protect fresh food [426,427,428]. Chitosan edible coatings and edible food packaging have been a subject of interest for many studies due to their ability to inhibit the growth of bacteria and fungi, except those that include chitosan in their cell wall [18,19].

There are several possible mechanisms by which the chitosan film can protect food: (1) such as any physical barrier, the film blocks the access of microorganisms to food; (2) it inhibits respiratory activity by blocking oxygen transfer; (3) the chitosan chains can chelate nutrients; (4) the outer cell membrane of microorganisms can be altered by electrostatic interactions; (5) chitosan can cause damage to the cell membrane, leading to leakage of intracellular electrolytes; after chitosan penetrates inside through the cell wall, (6) chitosan can chelate internal nutrients or essential metal ions from the cell plasma; (7) it can affect gene expression and (8) it can penetrate the nucleus and bind to DNA, thus inhibiting the replication process [429].

Due to the poor mechanical properties of chitosan that limit its use in some fields, some plasticizers such as glycerin, xylitol, sorbitol, etc., are required. Priyadarshi et al. (2018) obtained a chitosan-based antimicrobial package by forming a polymeric layer directly on the food surface. The binding agent was citric acid, which also improved the stability of the film and affected the antioxidant properties, while glycerol acted as a plasticizer, influencing the flexibility of the chitosan film. The resulting packaging material had excellent water resistance. The films were transparent, which is desirable from the point of view of consumers who want to see the food inside the package, and also showed an extended shelf life [430].

The addition of layers of other polymers is an interesting way to improve the properties of the chitosan film. Sogut et al. (2018) obtained by pressing or coating films of chitosan and polycaprolactone (PCL). Both layers were loaded with nanocellulose (2–5%) and grape seed extract (15% *w*/*w*). Importantly, both types of membranes had antimicrobial activity, and the grape seed extract retained its antioxidant activity when loaded into the chitosan matrix [431].

Jiawei Yan et al. manufactured an edible chitosan-based film for fresh strawberries. The fruits were immersed in solutions of 1% chitosan and 1.5% carboxymethylcellulose. The two-component film showed properties much better than those of the pure chitosan film. The layer of chitosan-carboxymethylcellulose protected the strawberries from spoilage for a longer period, preserving texture and taste [432].

Many researchers have studied various metal oxide nanomaterials/chitosan bio-nanocomposites as potential antimicrobial food packaging materials. Chitosan/titanium dioxide (TiO_2_) nanoparticles (TNP) nanocomposites are in the interest of researchers due to the synergistic effect of the antimicrobial properties of both chitosan and TNP. Zhang et al. (2017) evaluated chitosan-TNP nanocomposites for their antimicrobial activity under visible light in food packaging. Chitosan-TNP films possessed efficient antimicrobial activities against four tested strains (i.e., *E. coli*, *S. aureus*, *C. albicans*, and *A. niger*), with 100% inhibition observed after 12 h [433].

In research by Siripatrawan et al. (2018), attention was drawn to the concentration of antimicrobial agents that play an essential role in their inhibitory efficacy. They developed an active packaging from a combination of chitosan and TNP in a range of concentrations (0, 0.25, 0.5, 1, and 2% *w*/*w*) and tested the composites produced for application as an ethylene scavenging system. The chitosan film containing 1% TNP exhibited antimicrobial activity against Gram-positive (*S. aureus*) and Gram-negative (*E. coli*, *Salmonella Typhimurium*, and *P. aeruginosa*) bacteria and fungi (*Aspergillus* and *Penicillium*). Based on their work, nanocomposite films are believed to have more broad applications as active packaging systems for food [141].

In research, Lin et al. (2015) studied chitosan-TNP hybrids with silver nanoparticles (AgNP) using a photochemical reduction method. They measured antimicrobial activity against a non-toxigenic *E. coli* O157:H7 strain. The TNP-chitosan and AgNP-chitosan films did not show an inhibitory effect; however, the TNP-AgNP-chitosan nanocomposites showed a significant inhibitory effect [434].

Not only different concentrations of TNPs but also mixing them with biobased antimicrobial extracts contribute to options for enhancing the antimicrobial properties of the films. Zhang et al. (2019) investigated multifunctional food packaging films based on chitosan, TNPs, and black plum peel extract (BPPE). They reported that the chitosan/TNPs/BPPE composite film showed the highest antimicrobial activity against four food pathogens, including *E. coli*, *S. aureus*, *Salmonella*, and *L. monocytogenes*, compared to the other samples, which probably was due to the combined antimicrobial effect of chitosan, TiO_2_, and BPPE. The scientists confirmed the higher antimicrobial activity of chitosan/TNPs/BPPE films against Gram-positive than Gram-negative bacteria, which was related to differences in the cell membrane structure [435].

Hanafy et al. (2021) studied a series of different combinations of chitosan-TNP-oleic acid nanocomposite films. Antimicrobial effects were determined as zones of inhibition against *B. cereus*, *S. aureus*, *C. albicans*, *A. niger*, and *E. coli*. The authors reported, as a consequence of increasing TiO2 concentration, an increase in antimicrobial activity against *B. cereus*, *S. aureus*, and *A. niger* [436]. In addition, Lan et al. (2021) studied packaging films based on chitosan, nano-TiO2, and red apple pomace (APE). Inhibition zones against *E. coli* and *S. aureus* were estimated, and, similar to other reports, all of the studied films had more effective antimicrobial activities against Gram-positive (*S. aureus*) compared with Gram-negative bacteria (*E. coli*) [437].

Youssef et al. (2015) investigated ZNP/AgNP-chitosan nanocomposites for food packaging applications. Chitosan-based nanocomposite films with ZNPs and AgNPs showed good antimicrobial activity against both Gram-negative (i.e., *E. coli*, *S. Typhimurium*, and Gram-positive (*S. aureus*, *B. cereus*, and *L. monocytogenes*) bacteria. Unfortunately, scientists have not studied the application of both the ZNPs and AgNPs, which could be an interesting study of the possible synergic antimicrobial effects of nanoparticles [438].

Al-Naamani et al. (2016). studied chitosan-ZNPs as an antimicrobial coating on polyethylene (PE) films. The films were tested against two Gram-negative bacteria, *E. coli*, *S. enterica*, and one Gram-positive bacteria, *S. aureus*. The PE films did not show any antibacterial effect, while both the chitosan-coated PE and the chitosan-ZNP-coated PE significantly inhibited bacterial growth. Based on their study, PE coated with chitosan-ZNP nanocomposites offers a promising solution to improve the antimicrobial properties of PE films [439].

To increase the desired antimicrobial properties of chitosan-based nanocomposites, Sani et al. (2019) developed a film of chitosan-ZNPs with Melissa essential oil. Based on their study, all films showed an inhibitory effect against *E. coli* that was enhanced by the addition of ZNPs and essential oil [440].

Another important ingredient with antimicrobial properties is silver nanoparticles (AgNPs). Pandey et al. (2020) developed AgNP chitosan nanocomposites in polyvinyl alcohol (PVA) blend. The antimicrobial activity of the fibrous layer was analyzed against *E. coli* and *L. monocytogenes*. As expected, the PVA nanolayer did not show any inhibition effect; however, the PVA-chitosan and AgNP nanolayers showed an inhibitory effect against both tested strains [441].

In research Ghasemzadeh et al. (2021) studied a series of chitosan-agarose-AgNP nanocomposites. The antimicrobial effect of the films against *P. aeruginosa*, *E. coli*, and *S. aureus was evaluated.* Chitosan-agarose films did not show antibacterial activity; however, AgNP-containing films showed significant zones of inhibition against the tested strains [442].

Qin et al. (2019) developed chitosan-based films with incorporated AgNP (2%) and purple corn extract. The antibacterial behavior of the films produced was studied against *E. coli*, *Salmonella*, *S. aureus*, and *L. monocytogenes*. Chitosan nanocomposites with AgNP showed five times greater antimicrobial effects against all pathogens tested compared to pure chitosan films [443].

Shankar et al. (2021) examined the effect of chitosan essential oils-AgNP nanocomposite films on the shelf life. This nanocomposite was found to reduce the viability of bacteria (*E. coli*, *L. monocytogenes*, *Salmonella*) and fungal strains (*A. niger*) [444].

In conclusion, chitosan and its derivatives seem promising biodegradable and biocompatible polymers for food packaging. Because raw chitosan is characterized by weak mechanical, hydrophobic, and thermal properties, it was decided to use metal and metal oxide (e.g., TiO_2_, ZnO, and Ag) nanomaterials to provide them with improved functionality, including enhanced antimicrobial activity [445].

#### 6.1.3. Cellulose

Cellulose, due to its biocompatibility, chemical stability, nontoxicity, good barrier properties, and low price, are used as a key component in paper production and also to fabricate edible packages. As with starch-based films, cellulose-based films are hygroscopic, and the properties depend on the relative humidity, temperature, and nature of the substituents. Films made from cellulose derivatives have no inherent inhibitory or antioxidant properties. Most studies are related to the development of biocomposite systems, with a cellulose-based matrix associated with an antioxidant or natural biocide [416,446,447,448].

The use of several edible films is described in the literature, for example, those made from carboxymethylcellulose (CMC), and glycerol mixed with citrus essential oil as an antimicrobial agent. Tests have shown a significant increase in antibacterial activity against *E. coli* or *S. aureus* when added to 2% from essential oils of lime, lemon, and orange essential oils [449].

Ortiz et al. (2018) have studied microfibrillated cellulose mixed with soy proteins and glycerin as a plasticizer and with clove essential oil as the antimicrobial and antioxidant ingredient. The results obtained confirmed the increase in antioxidant properties as well as the antimicrobial activity of the film against various bacteria relevant to foodborne diseases [450]. Many authors had investigated the antimicrobial efficacy of cellulose-based films when they incorporated the most studied bioactive compounds such as bacteriocins [451,452,453], essential oils [454,455], probiotics [456,457], anthocyanins [458], and chitosan [459,460].

In particular, zinc oxide nanorods (ZnOs) and grapefruit seed extract (GSE) are potential nanofillers in cellulose-based films. ZnO is also a well-known nanofiller and has already been applied to fabricate numerous biopolymer-based films because of its exceptional antimicrobial action and noncytotoxicity. Grapefruit seed extract (*Citrus paradise Macf.)* is known to have outstanding antioxidant properties and comprehensive antimicrobial activity. Polyphenols, tocopherol, and flavonoids are the main active compounds present in GSE [461,462]. Roy et al. (2021) prepared CNF-based films incorporating ZnO and GSE as functional nanofillers. The reinforcement of functional fillers in CNF-based nanocomposite films improved their UV light blocking and water vapor barrier properties. Furthermore, the CNF-based nanocomposite film also exhibited good antioxidant activity and intense antimicrobial activity against foodborne pathogens [458].

Silver nanoparticles (AgNP) in cellulose films significantly increase the stability and mechanical strength of such a compound and provide antimicrobial food packaging [463].

Mocanu et al. (2019) tested bacterial cellulose (BC) as a potential antimicrobial polymer for food packaging. The authors investigated the synergistic effect of zinc oxide nanoparticles and propolis extracts deposited on the surface of a bacterial cellulose film. These BC-ZnO composites were further impregnated with various concentrations of ethanolic propolis extracts. The synergistic antimicrobial effect of the BC-ZnO-propolis films against *E. coli*, *B. subtilis*, and *C. albicans* revealed that this modified cellulose did not influence Gram-negative and eukaryotic cells [464].

Tarabiah et al. (2022) prepared a biodegradable blend of polyethylene oxide and carboxymethylcellulose (CMC) with loaded ZnO nanorods as a filler. The authors demonstrated that PEO/CMC/ZnO nanobiocomposites could be used as a UV mask and demonstrate antimicrobial activity against *S. aureus* and *E. coli* [465]. Su et al. (2017) examined biocompatible polymers in detail, such as sodium alginate (SA) and carboxymethylcellulose (CMC) loaded with silver nanoparticles (AgNP). The antibacterial effects of these water-soluble polymer-protected AgNPs were assessed against both Gram-negative (*K. pneumoniae*) and Gram-positive (*S. pyogenes*) bacteria. Both polymers showed excellent antibacterial activity against both types of bacteria. Importantly, an increase in the concentration of CMC/AgNPs and SA/AgNPs caused increased inhibition of the biofilm relative to the negative control [466].

#### 6.1.4. Alginates and Carrageenans

Alginates and carrageenans can react with polyvalent metal cations, especially calcium ions. Increasing the concentration of cations during alginates gelation results in a dense structure with lower water content, lower porosity, and thus lower gel permeability, which makes these materials used in the packaging industry [467].

Motelica et al. (2021) investigated biodegradable alginate films prepared by adding zinc oxide nanoparticles (ZnONP) and citronella essential oil (CEO) for cheese packaging. Antibacterial activity was determined against two strains of Gram-negative bacteria (*E. coli* and *S. typhi*) and two Gram-positive bacteria (*B. cereus* and *S. aureus*). Inhibition was shown to be relatively uniform among the four bacterial strains tested, with ZnO and CEO acting synergistically [468]. Additionally, this research group investigated the addition of spherical AgNPs and lemongrass essential oil (LGO) as antimicrobial agents. The results confirmed the possibility of using alginate/Ag NPs/LGO films as antimicrobial packaging to preserve the color, surface texture, and softness of the cheese for 14 days [469].

Furthermore, sodium alginate is a material that has been successfully used in the food industry as a packaging material that reduces the drying of meat and as a thickening and gelling agent, as well as a colloidal stabilizer in the beverage industry [470,471]. Galus (2019) studied the effect of adding a plasticizer (glycerol) to sodium alginate and used the obtained material as a coating for fruits and vegetables. The results of the investigation showed that the material obtained delays microbiological and degenerative damage enables color retention, and maintains the content of polyphenols and anthocyanins, resulting in a complete improvement in the quality of the fruit after harvest [472].

Carrageenan films are produced by cooling their hot neutral or alkaline aqueous solutions to form a gel, followed by drying. These films have poor mechanical strength, and as a result of the hydrophilic properties of these films, only minimal moisture barrier properties are expected. Despite this, they provide a good oxygen and lipid barrier, which makes them widely used to extend the shelf life of various food products.

Carrageenan films have been used as protective coatings in the packaging of, among others, fruits, vegetables, cheese, frozen desserts, and meat and fish products. As protective agents, they prevent the superficial dehydration of products and are also important carriers of antimicrobials and antioxidants [473]. The results obtained by Bico et al. (2009) confirmed that dipping in a preservative solution in combination with a carrageenan coating and storage in a controlled atmosphere can be a good method to store freshly cut bananas for 5 days at 5 °C [474]. In a similar study, Lee et al. (2003) studied the effect of edible carrageenan coatings in combination with antibrowning agents on apple slices when stored at 3 °C for 2 weeks. The edible coating controlled the initial respiration rate of the apple slices while maintaining the correct color and texture of the apples and reducing microbes at the same time [475,476].

Furthermore, antimicrobial and antioxidant carrageenan films have been proposed for food application as active packaging using a compound with essential oil [477,478]. Shojaee-Aliabadi et al. (2014) characterized films containing different concentrations of essential oil of *Zataria multiflora Boiss* (ZEO) and *Mentha pulegium* (MEO). The addition of essential oils to carrageenan films significantly improved the water vapor barrier properties, and the films were opaque and more flexible [479].

### 6.2. PLA, PGA, PCL, PBS Composites

Bioresorbable and biocompatible polylactide (PLA), due to its excellent useful physicomechanical properties, which are comparable to traditional polyolefins (mechanical strength, high modulus, transparency), is one of the most attractive polymers among biodegradable ones [480]. It found many applications in agriculture, packaging (as film or coating), the automotive industry, medicine, and pharmacy. PLA packaging can be produced through many processes, including film blowing, thermoforming, injection molding, and sheet or blow molding.

Turalija et al. (2016) modified PLA with silver, chitosan, and biobased alcohol (glycerol and polyethylene glycol). Silver, as the main efficacious and useful antibacterial agent, was incorporated into the PLA polymer by surface modifications using plasma technology. Studies of the antimicrobial activity of PLA films against *E. coli*, *L. monocytogenes*, and *S. typhimurium* bacteria have shown a significant inhibitory effect on the growth of bacteria strains. A lower efficiency in the inhibition of *S. aureus* was observed. The results indicate that such an environmentally friendly material could find application both in the food industry (preservative packages of food and juices) and for medical and other purposes [481]. Bayraktar et al. (2019) [482] studied the antibacterial properties of silver nanowire/PLA 3D printed nanocomposites against different standard bacterial strains using conventional microbiological methods. The prepared nanocomposites were found to have a high antibacterial effect against both Gram-positive bacteria *S. aureus* and *E. coli*. Only 4% of the loaded nanowires were sufficient to inhibit the growth of both specimens at 100% in 2 h and maintained the bactericidal effect for 24 h against *E. coli* and 8 h against *S. aureus*.

Very promising antimicrobial properties have PLA blends with chitosan [483]. Although the antimicrobial activities of this blend come from chitosan, PLA gives these blends better mechanical and barrier properties [484]. The antimicrobial activity of PLA/chitosan films in food packaging is based on: (1) the protection of food against microorganisms outside of the packaging (physical barrier); (2) inhibition of oxygen transfer that facilitates access to nutrients to the microbial cell; (3) chelation of nutrients by the chitosan chain; (4) disruption of the functionality of cell membrane functionality by electrostatic disruption and (5) cell apoptosis due to penetration of chitosan within the volume of the cell-damaging nucleus.

Polyglycolide (PGA) and its copolymer with PLA-poly(lactide-co-glycolide) (PLGA) are widely described in the literature regarding antimicrobial packaging applications [485,486,487]. Biswal et al. (2020) examined the antimicrobial properties of PLGA microparticles. Porous particles were found to completely suppress the *E. coli* and *S. aureus* growth for more than 2 months [485]. This excellent bacterial activity was probably due to the synergistic action of antibacterial and degradation products of PLGA, i.e., lactic and glycolic acids, which are antibacterial by nature. Radusin et al. (2019) used electrospun technology to obtain microsized PLA fibers with an immobilized extract of *Allium ursinum*. The natural extract of *A. ursinum* exhibited significant antimicrobial activity against foodborne bacteria (high antibacterial activity against *E. coli* and moderate reduction against S. aureus) [487].

Granata et al. (2018) manufactured nanocapsules of PCL with loaded essential oils of *Thymus capitatus* and *Origanum vulgare.* They found a higher activity against foodborne pathogens than that of the corresponding pure essential oils [488]. Siskova et al. (2022) investigated the nonwoven electrospun PCL fabric loaded with the popular antimicrobial food additive nisin. Active porous PCL loaded with varying concentrations of nisin inhibited the growth of *S. aureus* and *E. coli*. The high porosity of the materials obtained facilitated the permeability of the gases (CO_2,_ O_2_), which is highly desired from an economic and environmental protection point of view because the amount of material used for preparation is much lower than that used in traditional methods. Packages made of PCL and PCL/nisin fibrous mats demonstrated antispoilage properties that result in a prolonged freshness of fruits, improving their shelf life and, consequently, their safety [489].

Some studies have been conducted using biodegradable poly(butylene succinate)-based composites with antimicrobial properties. PBS is also known to be easily biodegradable in blends with other polymers, such as PLA, cellulose, or starch [490,491,492]. Due to its excellent processing properties, PBS can be processed through blow films, fiber spinning, injection molding, thermoforming, or blow molding [493]. Veranitisagul et al. (2019) studied PBS-based composites modified with carbon black or nanosilver-coated carbon black for antimicrobial, conductive, and mechanical properties. Composites demonstrated not only antimicrobial activity against *E. coli* and *C. albicans* but also anti-electrostatic properties and therefore have the potential to produce smart and environmentally friendly keyboards, lowering the same requirements for the cleaning process [494].

De Souza et al. (2022) used ZnO and Ag-ZnO nanoparticles as fillers for the fabrication of poly(butylene adipate-co-terephthalate (PBAT) antibacterial films which are completely biodegradable flexible polyester, synthesized from 1,4-butanediol, adipic acid, and terephthalic acid. The antimicrobial evaluation of the nanobiocomposites against *E. coli* proved their high activity compared to that of neat PBAT. A synergistic effect was observed between ZnO and Ag nanoparticles; therefore, these types of biocomposites can also be beneficial for the production of antibacterial food packaging [495].

### 6.3. PHA Composites

Polyhydroxyalkanoates (PHA) are a family of lineal biopolyesters obtained through bacterial fermentation and are fully biodegradable to carbon dioxide and water under aerobic conditions or to methane in an anaerobic environment [496]. The most widely used are poly(3-hydroxybutyrate-co-3-hydroxyvalerate) (PHBV) and poly(3-hydroxybutyrate) (PHB). PHBV has better mechanical properties and a lower melting point, as well as a wider processing window due to lower glass melting and transition temperatures compared to PHB [497,498].

PHBV has low cytotoxicity, piezoelectric properties, and excellent oxygen barrier properties; however, it is brittle and has poor thermal stability [499,500]. Nevertheless, it is a very interesting material from the point of view of its modification to provide it with antimicrobial properties [497,501]. Castro-Mayorga et al. (2017) demonstrated the antimicrobial properties of PHBV modified with silver nanoparticles (AgNPs) and zinc oxide (ZnO) for food packaging. They found that PHBV/ZnO nanostructures may be very useful for active food packaging and food contact surface applications. Furthermore, ZnO nanoparticles have improved the optical and thermal properties of nanocomposite films in addition to the antimicrobial properties that qualify them for food packaging [502]. The other study conducted by Anzlovar et al. (2018) revealed that PHBV is less susceptible to degradation when fabricated in the presence of ZnO nanoparticles by melting process compared to PLA/nZnO compound (ZnO should not exceed 0.1 wt.%) [503].

Many approaches have been made to modify PHBV for its advanced application [504]. Ibrahim et al. (2022) studied a PHBV bionanocomposite with silver-doped zinc oxide. The obtained PHBV loaded with Ag-ZnO nanoparticles was found to be promising antibacterial materials. The microbial study proved antimicrobial activity against Gram-positive and Gram-negative strains of *S. aureus* and *E. coli*. The higher the concentration of Ag-ZnO NPs loaded into the PHBV matrix, the stronger the inhibition effect of bacteria growth was observed. Among the others, bio-nanocomposite PHBV/Ag–ZnO(10%) was the most effective against both types of the selected bacteria [505]. Diez-Pascual et al. (2014) demonstrated that the best barrier and antimicrobial properties, as well as the maximum Young’s, storage moduli, and tensile strength, were found for the composite of PHBV loaded with 4% ZnO NPs [506].

Diez-Pascual et al. (2021) used poly(3-hydroxybutyrate-co-3-hydroxy-hexanoate) (PHBHHx) for the preparation of biocomposite loaded with zinc oxide (ZnO) nanoparticles. PHBHHx demonstrates lower crystallinity than PHB, hence reduced stiffness and poorer moisture and gas barrier properties, which is a disadvantage for its use as a food packaging material. The antibacterial test against two model pathogen bacteria Gram-negative *E. coli* and Gram-positive *S. aureus*, of neat PHBHHx and biocomposite with ZnO NPs, showed that the survival ratio of both bacteria decreases progressively with the growing concentration of ZnO in the polymer matrix and finally reached almost 98 and 95% growth inhibition for *E. coli* and *S. aureus*, respectively, at the highest ZnO loading (5.0 wt.%) [507].

### 6.4. PVA Composites

Poly(vinyl alcohol) (PVA) is obtained through a hydrolysis reaction of poly(vinyl acetate) [508]. PVA is hydrophilic due to the presence of numerous hydroxyl groups in the main chain of the macromolecule. In addition, it has high crystallinity, high mechanical stability, and gas barrier properties, remarkable chemical resistance, is biodegradable, and is not toxic. Many reports are devoted to the formation of PVA-based blends with other polymers such as chitosan, agar, and poly(ethylene glycol) or incorporated additives [509,510,511]. Hu & Wang (2016) examined PVA-based composite films with incorporated hydroxypropyltrimethylammonium chloride cellulose (CM). Compared with a neat PVA film, the addition of CM improved the surface roughness, hydrophobicity, and water swelling ratio of the obtained films. The authors obtained an environmentally friendly, biocompatible polymeric film with excellent antimicrobial properties. The CM/PVA film demonstrated significantly higher antimicrobial activity (increased from 5 to 20%, probably due to the presence of more ammonium groups compared to plain CM).

A study on films consisting of PVA and chitosan (Ch) with the addition of poly(hexamethylene guanidine) (PHMG) demonstrated their biocidal potential against Gram-positive (*S. aureus*) and Gram-negative (*E. coli*) bacteria when compared to PHMG-free blends [512]. The authors found that PHMG not only plays the role of antimicrobial properties but also acts as a plasticizer and increases the elasticity and thermal stability of the obtained materials, most probably because of the presence of nitrogen atoms in the structure of PHMG that hindered the diffusion of oxygen into polymer matrixes.

Santiago-Morales et al. (2016) manufactured electrospun nanofibers from blends of poly(acrylic acid) (PAA) and PVA and evaluated their antimicrobial activity against *E. coli* and *S. aureus*. Membranes containing more than 35 wt.% PAA demonstrated significant antibacterial activity, which was particularly high for the Gram-positive *S. aureus*. The authors found that the chelation of the divalent cations that stabilizes the outer cell membrane was the driver mechanism of antibacterial activity. The effect on Gram-positive bacteria was attributed to the destabilization of the peptidoglycan layer [513]. Hashmi et al. (2020) tested the antibacterial activity of an electrospun PVA loaded with *Momordica charantia* (bitter gourd), which is a natural remedy for diabetic patients. These materials have potential antibacterial activity against both Gram-positive (*B. subtilis*) and Gram-negative (*E. coli*) bacteria strains [514].

### 6.5. Proteins

Proteins, because of their environmentally friendly properties, such as biodegradability, biocompatibility, and functionalities for the encapsulation of antimicrobial molecules, are relatively often used to produce fibers by electrospinning. Da Silva et al. examined the antimicrobial activity of ginger essential oil (GEO) incorporated into ultrafine fibers of a polymeric blend consisting of soy protein, polyethylene oxide, and zein (in the ratio of 1:1:1 ratio by volume). GEO showed antimicrobial activity against Gram-positive (*L. monocytogenes* and *S. aureus*) and Gram-negative (*E. coli*, *S. typhimurium*, and *P. aeruginosa*) Bacteria. The antimicrobial activity of fibers containing 12% GEO on fresh Minas cheese significantly reduced the proliferation of *L. monocytogenes* during refrigerated storage of 12 days, demonstrating its potential application in active food packaging [515].

Aziz and Halmasi (2018) studied the antimicrobial activities of whey protein isolate-based films with incorporated nanoencapsulated thyme extract (*Thymus vulgaris)* at various concentrations. Strong antimicrobial activity of the films containing nanoliposomes loaded with thyme extract was observed against S. aureus compared to E. *coli*. However, the antimicrobial activity was diminished in nanoactive films compared to free TE-loaded samples. The results indicate the potential application of these modified polymeric materials in the packaging of foods such as meat, products, cheese, nuts, fruits, and vegetables [516]. Altan et al. (2018) reported the antimicrobial activities of composite nanofibrous films consisting of zein and poly(lactic acid) loaded with carvacrol at various concentrations. Highly volatile carvacrol in electrospun zein and PLA fiber-preserved bread samples, indicating that they are good candidates for active food packaging applications to extend the shelf life of this product. The composite fibrous zein and PLA films containing 20% carvacrol inhibited the growth of mold and yeast (99.6 and 91.3%, respectively) [517].

Studies of modified gelatin toward its antimicrobial activities have also been carried out. Figueroa-Lopez et al. (2018) evaluated the antimicrobial effect of gelatin-cast films deposited on electrospun PCL as a barrier coating and black pepper oleoresin as a natural extract [518]. The antimicrobial activity of the developed multilayer was evaluated against S. aureus strains for 10 days. The results demonstrated that active multilayer systems stored in hermetically closed bottles increased their antimicrobial activity after 10 days by inhibiting bacteria growth and promoting its potential use in active food packaging applications.

Kim et al. (2018) examined hagfish skin gelatin films containing cinnamon bark essential oil (CBO) as an active packaging material. The antimicrobial evaluation of the gelatin films demonstrated that films containing 1% CBO were the most efficient against Gram-negative and Gram-positive foodborne pathogens such as *L. monocytogenes* and *S. Typhimurium*, exhibiting simultaneously antioxidant activities, as well as suitable physical and water barrier properties. For this reason, the authors recommend such films (with 1% CBO) as food packaging materials to maintain the quality of food by reducing the risk of microbial spoilage and lipid oxidation [519].

In addition, casein as a biopolymer was the subject of antimicrobial investigations. Abdollahzadeh et al. (2018) determined the activity of encapsulated *L. acidophilus* and *L. casei* in a sodium caseinate matrix to develop a probiotic-based film that can improve food safety. The viability of *L. acidophilus* and *L. casei* in the films was determined over 12 days. The antibacterial activities of the films were also tested against *L. monocytogenes*. The results demonstrated that the lactic acid bacteria remained viable for a storage period of 12 days. Samples covered with sodium caseinate film supplemented with lactic acid bacteria showed greater antibacterial activity than the control group on day 6 of preservation. Based on the results obtained, the authors concluded that the developed material could be used as a new effective packaging method to improve food safety [520].

### 6.6. Lipids and Waxes

Akyuz et al. (2018) studied the use of commercially available oils and fats consumed in the daily human diet (olive, corn, and sunflower oils, butter, and animal fats for the preparation of low-cost biodegradable films. The authors studied the antimicrobial activity of various chitosan-oil blends against a wide spectrum of foodborne and human pathogen microorganisms such as *E. coli*, *S. aureus*, *P. microbilis*, *P. vulgaris*, *P. aeruginosa*, *E. aerogenes*, *B. thuringiensis* and *S. enterica*, *S. typhmurium*, and *St. mutans*. The antioxidant activities of the chitosan-unsaturated oil blend films were higher than those of the chitosan-saturated oil films. Among all blends tested, the chitosan-olive oil film demonstrated the best mechanical properties and thermal stability as well as antimicrobial activity (comparable to commercial antibiotic gentamicin. Based on the result obtained, the authors recommended biodegradable chitosan-oil/fat blend films for food packaging and preservation as an alternative to synthetic materials [521].

Tonyali et al. (2020) encapsulated selected essential oils as a natural antimicrobial compound in liquid and solid lipid nanoparticles (thymol, cinnamaldehyde, and eugenol) to improve their solubility, stability, and control release kinetics. The release kinetics of immobilized compounds from pullulan-based films were studied as a function of the lipid structure (liquid versus solid) and carrier particle concertation. The release rate of essential oils was dependent on the concentration of active compounds and the physicochemical properties of the polymer carrier. The results may be useful for further and more detailed studies devoted to the use of volatile antimicrobial compounds in carrier lipid particles in active packaging applications and their large-scale manufacturing [522].

Zhang et al. (2015) studied the antimicrobial effect of beeswax and carnauba wax latex particles grafted with the antimicrobial agents polyhexamethylene guanidine hydrochloride and polyhexanide. Modified materials were examined as potential bifunctional agents to improve the water-vapor resistance of the paper surface after coating.

The antimicrobial evaluation against *E. coli* showed that the antimicrobial performance of the tested materials was significantly improved when the amount of biowax was greater than 20 mg/g of fiber. Beeswax had better water vapor barrier properties compared to carnauba wax. The bacterial inhibition rate of the modified biowax particles was driven primarily by contact between the bacteria, and the antibacterial agents adsorbed on the fibers rather than by diffusion of antibacterial components from the cellulose fibers [523].

## 7. The Use of Biodegradable Antibacterial Polymers in Biomedical Applications

Contamination of pathogenic microorganisms in medical devices and implants is a serious threat to the lives of patients. This risk is intensified by the observed increase in the resistance of these pathogens to antibiotic treatment. In addition to the introduction of new procedures and sterilization methods, one of the leading ways to combat this phenomenon and reduce the risk of cross-contamination [524] is the use of various medical devices, such as intravascular catheters, heart valves, orthopedic implants, threads, fibers, and surgical fabrics resistant to the development of bacteria and preventing their colonization and growth becomes necessary. It seems that in the case of implantable materials, it may be sufficient to use special antibacterial coatings that protect not only against contamination but also against the formation of a surface biofilm [525]. In the manufacturing of this type of coating, antibacterial polymers, including many of the biodegradable polymers discussed in this paper, are invaluable [526,527]. However, due to relatively poor mechanical properties or processing problems, most of these polymers are not suitable as a material for forming bioresorbable implants. In such applications, polymeric composites with nanoparticles of selected metals, metal oxides, and other compounds with strong antibacterial properties seem more beneficial [528,529].

### 7.1. Biodegradable and Antibacterial Wound-Dressings

Antibacterial polymers, however, can be widely used in the formation of dressings for difficult-to-heal wounds [530]. Biomaterials such as films, hydrogels, sponges and foams, and electrospun scaffolds have been produced that can significantly accelerate the wound healing process, including the four phases of coagulation and hemostasis, inflammation, proliferation, and wound remodeling with scar formation [531].

An interesting antibacterial material successfully and often used as a dressing material is chitosan in conjunction with other ingredients [532,533]. Liu et al. (2019) developed a CS/aloe vera film with poly(lactic and glycolic acid) microspheres encapsulated in curcumin. Thanks to the prolonged release of curcumin, the presence of corticosteroids, and aloe vera, bacterial colonies around the films were completely inhibited. These types of films can promote wound healing and reduce inflammation around the wound [532]. Similarly, Li et al. (2012) successfully developed a film based on CS and methoxy poly(ethylene glycol) loaded with curcumin, increasing collagen synthesis and accelerating wound healing [534]. Liu and Kim (2012) prepared a CS/PEG film cross-linked with genipin loaded with various amounts of ZnO and Ag NPs. Films prepared in this way show significantly increased antibacterial activity [535]. Moreover, the addition of nitric oxide (NO) to the CS film makes it possible to obtain a material with antibacterial properties [536].

Due to properties such as wound exudate absorption, moisture retention, and oxygen permeability, special attention has been paid to hydrogel dressings [537,538]. Hydrogels are typically composed of natural/synthetic polymers by physical or chemical cross-linking. By selecting or modifying monomers and cross-linking agents, the desired characteristics of hydrogels can be obtained, and thus they can be used as a carrier for various bioactive agents [539,540]. The use of hydrogels in the healing of chronic wounds has been extensively described in the literature. One can distinguish, e.g., hydrogels with natural antibacterial ingredients [538]; antibiotics [541,542,543]; metals [544,545]; antibacterial peptides (AMPs)[546] and with photo-enhanced antibacterial agents [547].

In the studies of Liang et al. (2019) developed antibacterial hydrogels based on hyaluronic acid-grafted-dopamine and reduced graphene oxide loaded with doxycycline. The resulting multifunctional hydrogel improved granulation tissue thickness and collagen deposition [542,548]. Filling the hydrogel with inorganic antibacterial materials, especially Ag NP and ZnO NP, can maintain a strong antibacterial effect for a long time, which reduces the production of microorganisms. Zhao et al. (2019) designed an antibacterial hydrogel with Ag nanoparticles enriched with polydopamine (PDA@Ag NPs/CPH) for epidermal sensors and dressings, significantly promoting diabetic foot wound healing [549].

Various ionic liquids and poly(ionic liquids) with bactericidal properties were employed to prepare antibacterial dressings. In the study by Li et al. (2019), hydrogels were obtained by mixing PVA and a copolymer of 1-vinyl-3-butylimidazolium bromide and acrylamide and then by adding a poly(ionic) liquid with antibacterial properties; the hydrogel effectively supports the wound-healing process [550].

In the last few years, scientists have also focused their attention on the use of antibacterial peptides in the treatment of antibiotic-resistant pathogens. Wang et al. (2017) developed a biomimetic hydrogel based on modified dopamine-poly-1-lysine-PEG (PPD). It has been shown that PPD hydrogel can easily and tightly integrate into biological tissue and has excellent in vivo hemostasis and wound healing acceleration ability. In addition, hydrogels exhibit exceptional anti-infective properties due to the natural antibacterial ability of ε-poly-l-lysine [551].

Pang et al. (2020) designed a two-layer and self-controlling wound dressing system, unusually integrated with electronics. The bottom layer of the dressing was a hydrogel loaded with gentamicin through a UV cleavable linker, while the top layer was a PDMS network equipped with a thermal sensor and an ultraviolet light emitting diode (UV-LED). With Bluetooth technology, the temperature value of the wound is transmitted to a portable device. When it exceeds 40 °C, a UV diode is turned on, triggering the release of antibiotics [552].

Electrospinning is another increasingly used technique for producing electrospun mats for wound dressings characterized by a large specific surface area, high aspect ratio, and high porosity. When obtained this way, dressings warrant increased exudate absorption and water and oxygen permeability. Importantly, the small pores enable the fibers to prevent the invasion of exogenous microorganisms [553,554]. Antibacterial and bioactive agents can be readily introduced into the fibers in situ or by modification after electrospinning by selecting the type of polymer and the electrospinning process. Antibacterial agents, incl. antibiotics [555,556], inorganic nanoparticles (Ag, ZnO, TiO_2_) [557,558,559,560], and AMPs [561,562], show promise for the use of this type of scaffolds in wound healing.

For example, Dodero et al. (2020) developed mats based on alginate with ZnO nanoparticles using the electrospinning technique. By using a commercial collagen product as a control, the biological response of the prepared mats was assessed, paying attention to the influence of the cross-linking agent used (Ca^2+^, Sr^2+^, or Ba^2+^ ions) and the presence of nanofillers. The cultures of fibroblasts and keratinocytes successfully proved the safety of the prepared mats based on alginate, while the ZnO nanoparticles showed strong anti-bacteriostatic and antibacterial properties; most notably, samples cross-linked with strontium and barium exhibited cell adhesion and growth properties [559].

Wang et al. (2017) designed a nanocomposite membrane with Cu_2_S micropatterns, consisting of poly(D, L-lactic acid) and PCL, applicable in the treatment of skin cancer and wound healing. Uniformly deposited Cu_2_S nanoparticles of the membrane exhibited controlled near-infrared photo-thermal activity, resulting in high mortality (>90%) of skin cancer cells and effectively inhibiting tumor growth. In addition, the mem-branes promoted adhesion, proliferation, and skin cell migration, thus significantly stimulating angiogenesis and skin defect healing in vivo [563].

Li et al. (2019) developed a composite membrane design consisting of electrospun polylactide: poly(vinylpyrrolidone)/polylactide: polyethylene glycol (PLA: PVP/PLA: PEG) fibers loaded with antimicrobial peptides (AMPs) as a functionally integrated wound dressing for effective treatment of burns [562].

Similar to mats, biodegradable sponges can absorb huge amounts of wound exudate due to their interconnected porous structure. Hydrophilic sponge dressings interacting with cells are commonly used as hemostatic agents and burn wound healing materials [564]. In order to accelerate the wound healing process, various systems have been developed based on sponge dressings produced, most often using the chitosan-based freeze-drying method. Additional strengthening of the antimicrobial properties of sponge dressings, as in the above cases, was obtained by introducing antimicrobial agents [565,566,567,568,569,570,571,572].

Zhao et al. (2021) developed a chitosan sponge cross-linked with polydopamine (PDA) with antioxidant activity and high hemostasis efficiency for lethal noncompressible/coagulopathy hemorrhage. The resulting material showed a significantly better wound closure effect than chitosan sponge and commercial Tegaderm™ dressing, reducing inflammatory infiltration and promoting vascularization and cell recruitment [572].

Liang et al. (2016) developed a novel composite dressing of silver nanoparticles (AgNPs) and chitosan as a material used for wound healing. First, the AgNPs were combined into a chitosan sponge, which was prepared in a freeze-drying process. One side of the sponge was then modified with a thin layer of stearic acid. The inclusion of AgNP in the chitosan dressing increases the antibacterial activity against drug-sensitive and drug-resistant pathogenic bacteria. The hydrophobic surface of the dressing is waterproof and non-adhesive to dirt, while the hydrophilic one retains the ability to absorb water and effectively inhibits the growth of bacteria [567].

### 7.2. Biodegradable Drug Release Systems with Antibacterial Properties

The use of biodegradable antibacterial carriers in the release of antibacterial drugs is also fascinating. Such systems in the form of nanoparticles (micelles, nanogels, or polymersomes) releasing antibiotics show significantly increased activity (Figure 25) and allow the elimination of antibiotic-resistant strains of bacteria due to the expected synergistic effect [573,574,575].

For example, Xiong et al. (2012) showed that methicillin-resistant *S. aureus* (MRSA) could be treated therapeutically with targeted antibiotic delivery to macrophages using a mannosylated nanogel as the drug carrier [576]. Dey et al. (2018) applied micelles built of amphiphilic salicylaldehyde derivatives carrying Rifampicin and showed the increased antibacterial activity of this type of system compared to antibiotic alone. He showed that by using these micelles, it is possible to fight MRSA and prevent the formation of a biofilm of this bacterium [577]. Micelles formed from chitosan and cyclodextrin derivatives, containing rifampicin or moxifloxacin, also show a similar effect [578,579]. These micelles were then filled with Ciprofloxacin. The authors demonstrated a method to enhance the antibacterial targeting of micelles and to trigger the release of both antibiotics at the infection site.

Systems based on antibacterial polymer nanoparticles can fight severe, difficult-to-treat bacterial diseases. For example, Zhang et al. (2018) showed that micelles containing ciprofloxacin and an anti-inflammatory agent formed from an antibacterial block copolymer (poly(ethylene glycol)-b-poly(βamino ester)-b-poly(ethylene glycol) grafted with PEGylated lipid (Biotin- PEG-b-PAE(-g-PEG-b-DSPE)-b-PEG-Biotin), coated with ICAM-1 antibody due to targeted activity can be an effective drug in the treatment of sepsis [580]. Pan et al. (2022) conducted a similar study to find systems against sepsis. The team used nanoparticles formed with other synthetic copolymers grafted with antimicrobial peptides and biotin. Formed micelles were also coated with anti-ICAM-1. Researchers confirmed that such a system exhibited ideal antibacterial properties against drug-resistant bacteria and might provide a specific strategy for targeting sepsis [581].

Casanova et al. (2023) found that by using amphiphilic antimicrobial synthetic copolymers with side groups containing quaternized amine cations, it is possible to combat major bacterial pathogens associated with cystic fibrosis [582,583]. The composite system was prepared by incorporating mono-dispersed polydopamine functionalized bioactive glass nanoparticles into a Pluronic F127-ε-poly-L-lysine hydrogel containing imine bonds. Based on in vitro and in vivo studies, the authors demonstrated a strong antibacterial effect of formed nanoparticles, especially against multidrug-resistant bacteria. In addition, the hydrogel effectively accelerates wound healing. Sebastian et al. (2023) propose poly(2-aminobenzoic acid)-blend-Aloe vera as a means in the fight against breast cancer [584]. Biswas et al. (2022) report a strategy to design an antibacterial drug delivery scaffold effective against *E. coli* able to deliver doxorubicin. Diblock amphiphilic copolymers of poly(ethylene glycol methyl ether methacrylate) (PEGMA) and tyrosine were used to form the antibacterially active micelles loaded with the anticancer drug. The system looks at perspective in colorectal cancer and inflammatory bowel disease therapy [585].

## 8. Conclusions and Perspectives

In this Review, in reference to the mechanisms of selective neutralization of pathogenic bacteria by natural polymers (e.g., chitosan and antimicrobial peptides), we show different approaches to the development of biodegradable polymers and polymer matrix composites with antibacterial properties. We show that through the appropriate control of the chain structure, the lengths of hydrophilic and hydrophobic blocks, the share of amines or amine salts, and other biologically active groups it is possible to adjust the biological properties of the polymers for the selective setting of its antimicrobial activity. Such innovative materials can be applied in medicine, the healthcare industry, food packaging and storage, and other domains of everyday life.

Although today the total antimicrobial polymer market is $37.41 billion, and the revenue forecast in 2028 is projected to be $81.95 billion [586]. Devices made of engineering polymers with antibacterial coatings or modified with antibacterial additives predominate. Of course, only a small part of the antimicrobial polymer market is devoted to medical devices and food packaging based on biodegradable polymers. The global market for bioresorbable polymers is projected to reach much less, around $6.437 million in 2027 [587].

It seems that because of both the amount of research conducted and their degree of advancement, these are primarily composites containing polymers of natural origin or obtained by biotechnology, polysaccharides, and polypeptides. This is particularly visible in increasing attempts to use such as food packaging, presenting additional properties that increase the expiration date, being an alternative to polymers based on petroleum derivatives [588]. Unfortunately, polymers of natural and synthetic origin usually do not have intrinsic antibacterial properties (except chitosan); therefore, they are intensively studied as a matrix with loaded active additives such as nanoparticles. Antibacterial bionanocomposites with an improved barrier and mechanical and thermal properties also play an important role in food packaging [501,589]. Bionanocomposites consisting of a biopolymer matrix and active antimicrobial agents (particles in size between 1–100 nm) are a new class of materials that demonstrate much-improved properties compared to the base biopolymers due to the high aspect ratio and high surface area of the nanoparticles.

When it comes to biomedical applications, the path of strict imitation of natural polypeptides does not seem to be entirely correct. Through the appropriate selection of the chain structure, the length of hydrophilic and hydrophobic blocks, and the share of amines or amine salts and other biologically active groups, it is possible to control the biological properties of the polymers obtained in this way, which allows for the selective setting of its antibacterial or antifungal activity. The way of synthesis of new, highly active, antibacterial, biodegradable polymers that simultaneously fulfill other biomaterial functions is certainly still wide open and probably the most promising for the future.

We can assume that bioresorbable medical devices (implants, dressings, controlled drug delivery systems) and biodegradable packaging materials with bactericidal or bacteriostatic properties will become more and more popular by the end of this decade. The way of synthesis of new, highly active, and antibacterial polymers that simultaneously fulfill other functions such as mechanical properties, transparency, and biodegradability is certainly still wide open and probably the most promising for the future.

## Figures and Tables

**Figure 1 ijms-24-07473-f001:**
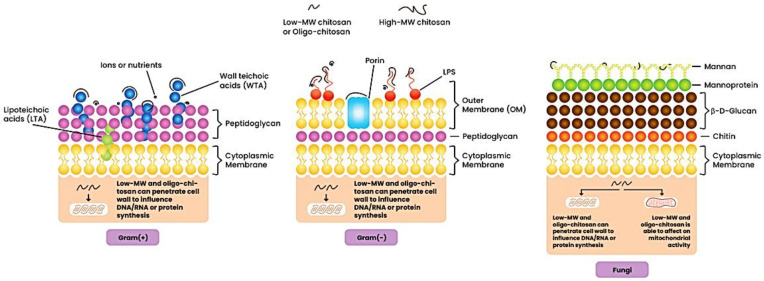
Schematic of action modes of chitosan on Gram-positive bacteria, Gram-negative bacteria, and fungi. Inspired by [18].

**Figure 2 ijms-24-07473-f002:**
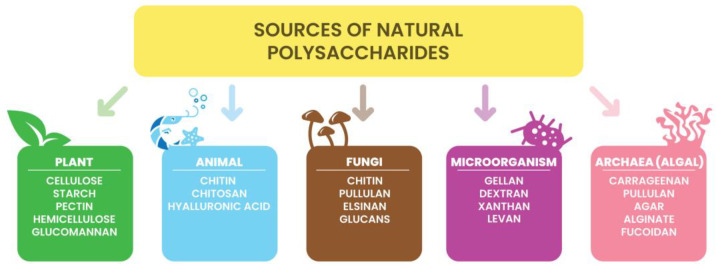
Resources of natural polysaccharides with antibacterial and bacteriostatic properties.

**Figure 3 ijms-24-07473-f003:**
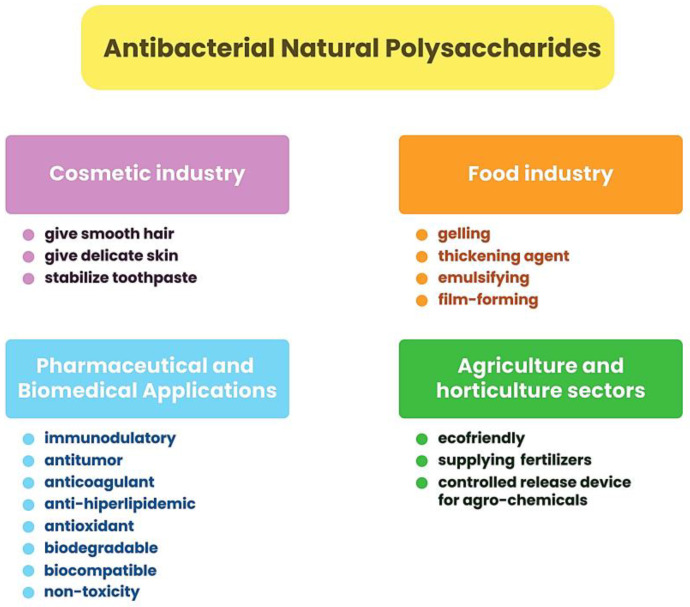
Application of natural antibacterial polysaccharides.

**Figure 4 ijms-24-07473-f004:**
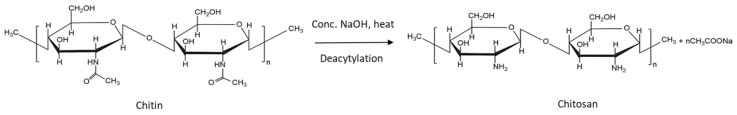
Deacetylation of chitin to chitosan.

**Figure 5 ijms-24-07473-f005:**
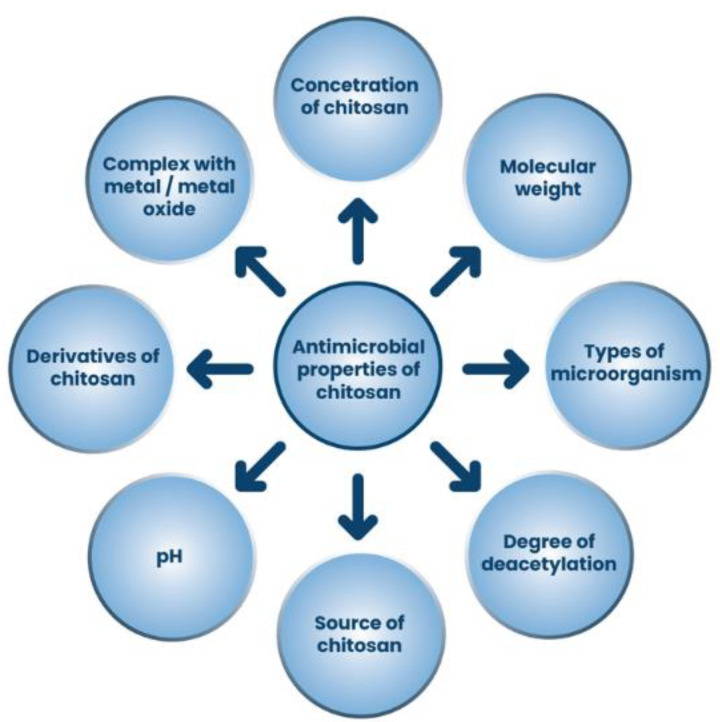
Factors influencing antimicrobial properties of chitosan.

**Figure 6 ijms-24-07473-f006:**
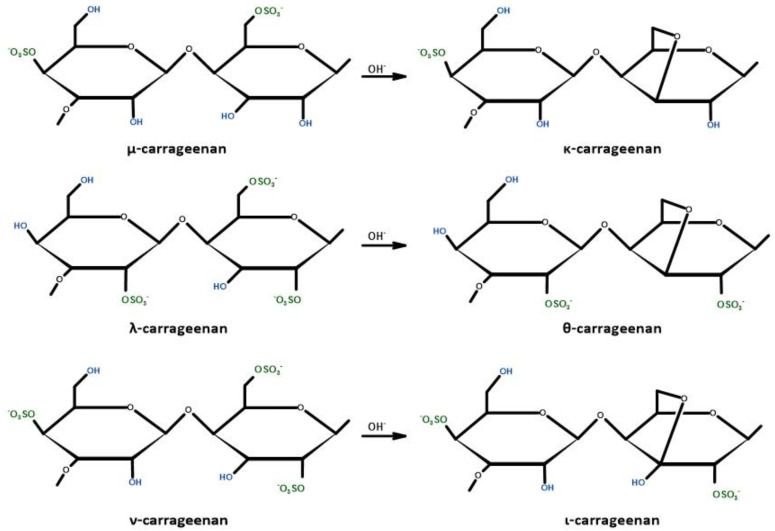
Chemical structure of commercial carrageenans.

**Figure 7 ijms-24-07473-f007:**
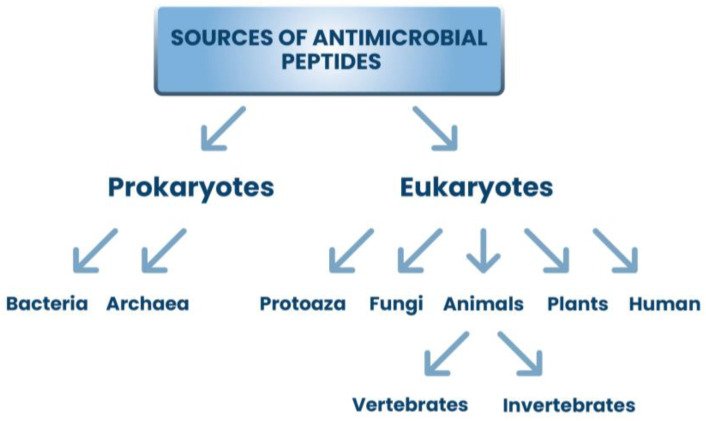
Sources of antimicrobial peptides.

**Figure 8 ijms-24-07473-f008:**
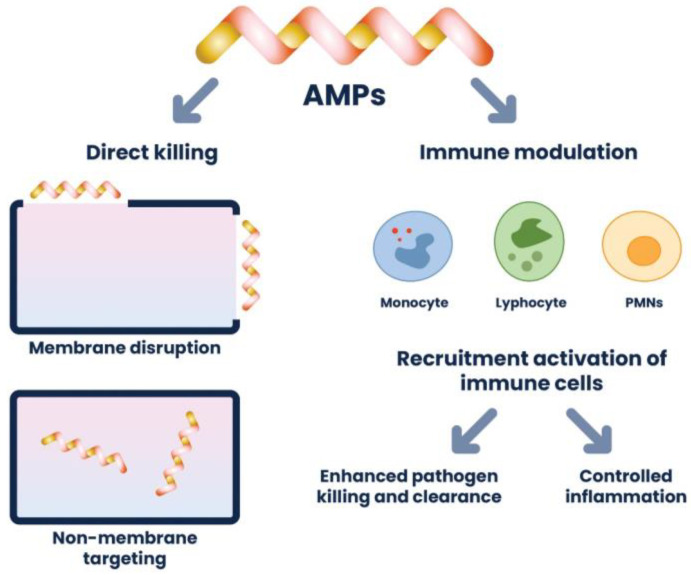
Mechanisms of action of antimicrobial peptides.

**Figure 9 ijms-24-07473-f009:**
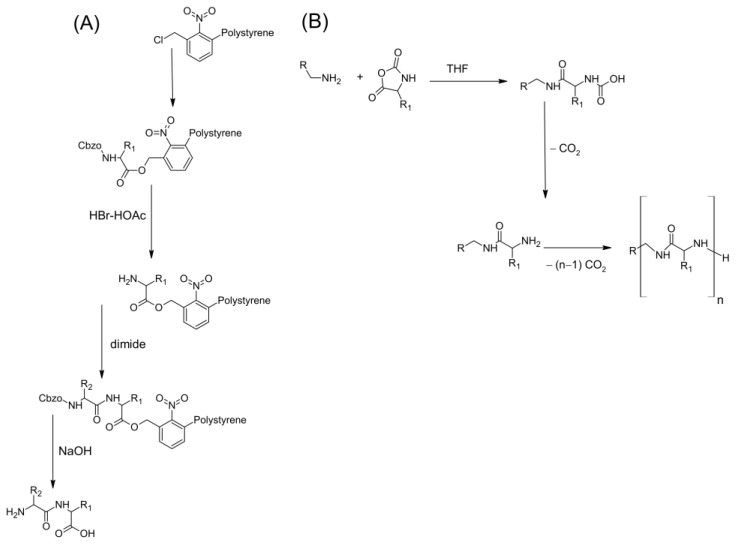
Peptide synthesis (**A**) Concept of the Solid Phase Peptide Synthesis; (**B**) by ROP of α -aminoacid-N-carboxyanhydrides with the use of amine initiator.

**Figure 10 ijms-24-07473-f010:**
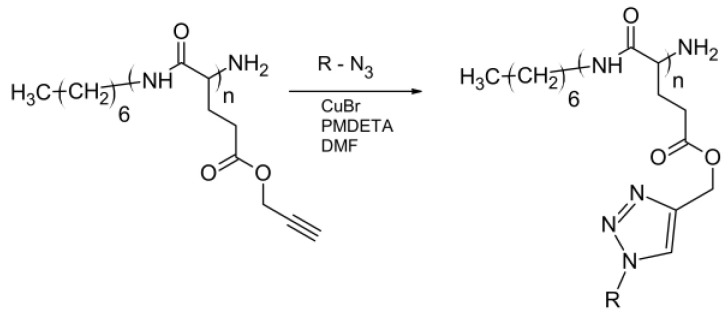
Synthesis of biomimetic polypeptides by ROP γ-propargyl-L-glutamate N-carboxyanhydride and click reactions—based on [316].

**Figure 11 ijms-24-07473-f011:**
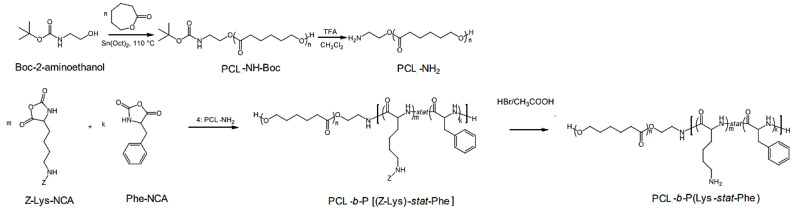
Synthetic route to obtaining block copolymers poly(ε-caprolactone)-block-poly(Lys-co-Phe) based on [321].

**Figure 12 ijms-24-07473-f012:**
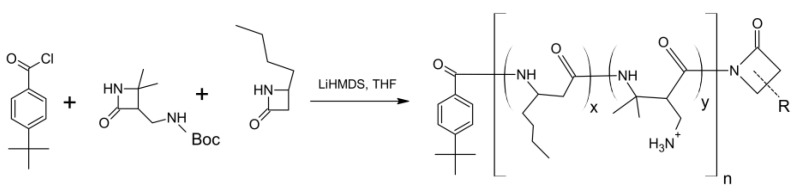
Synthesis of β-peptides by copolymerization of two different β lactams with the use of lithium bis(trimethylsilyl)amide as initiator and 4-*tert*-butylbenzyl chloride as co-initiator—based on [324].

**Figure 13 ijms-24-07473-f013:**
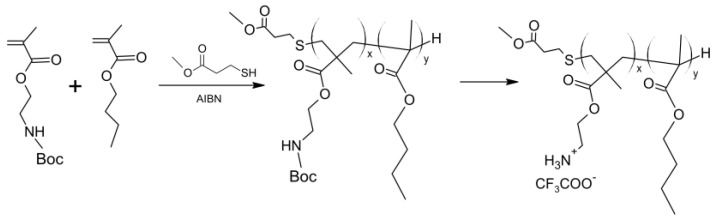
Synthesis of amphiphilic polymethacrylates with antibacterial properties—based on [327].

**Figure 14 ijms-24-07473-f014:**
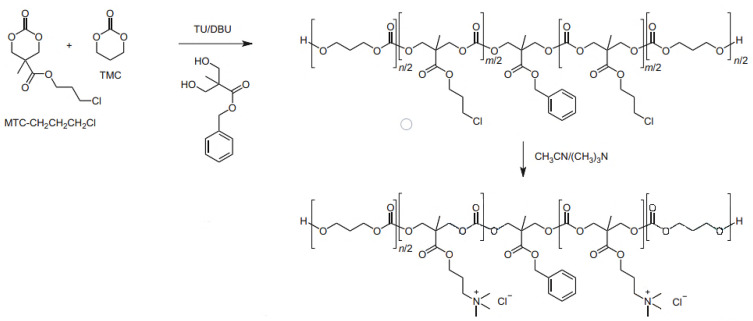
Synthesis of amphiphilic polycarbonates—based on [329].

**Figure 15 ijms-24-07473-f015:**
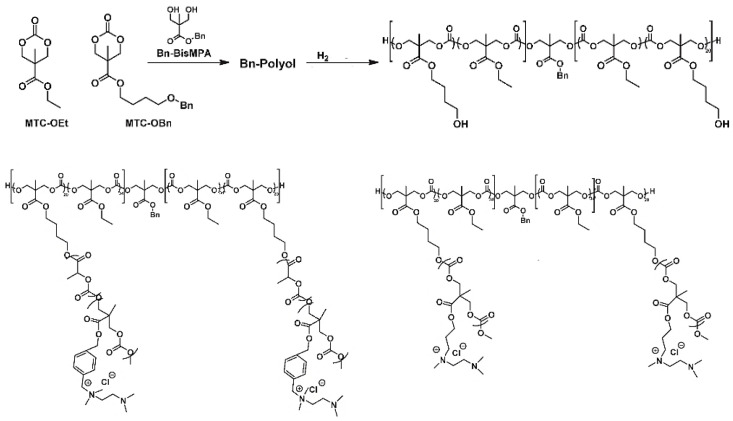
Scheme of synthesis of polycarbonate polyols and final grafted polycarbonates with different long side chains ended by cationic centers—based on [330].

**Figure 16 ijms-24-07473-f016:**
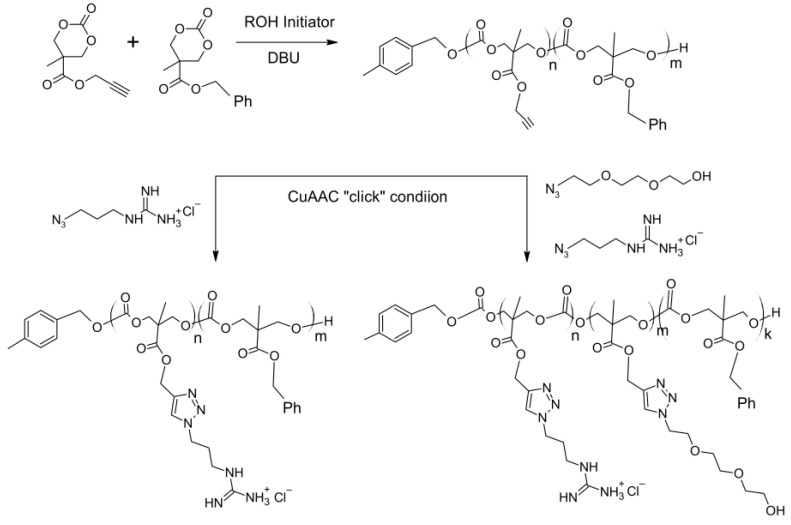
Scheme of synthesis of guanidine functionalized polycarbonate copolymers based on [332].

**Figure 17 ijms-24-07473-f017:**
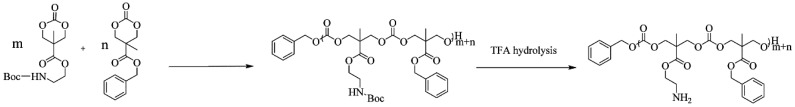
Synthesis of polycarbonates with pending amine groups.

**Figure 18 ijms-24-07473-f018:**
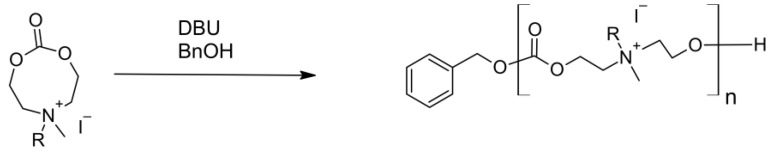
Direct ROP of quaternized cyclic carbonate.

**Figure 19 ijms-24-07473-f019:**
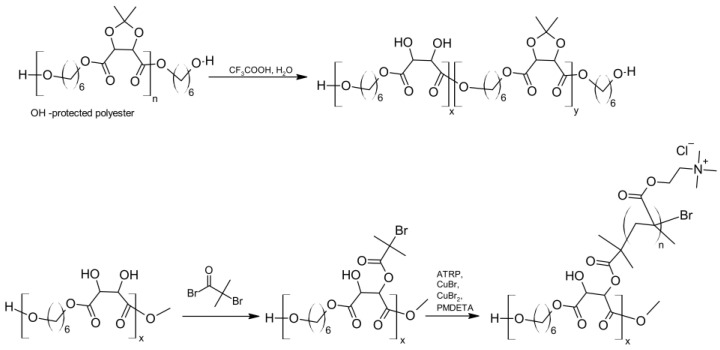
Scheme of biodegradable polyester with hydroxyl groups synthesis and process immobilizations of the cationic brushes [338].

**Figure 20 ijms-24-07473-f020:**
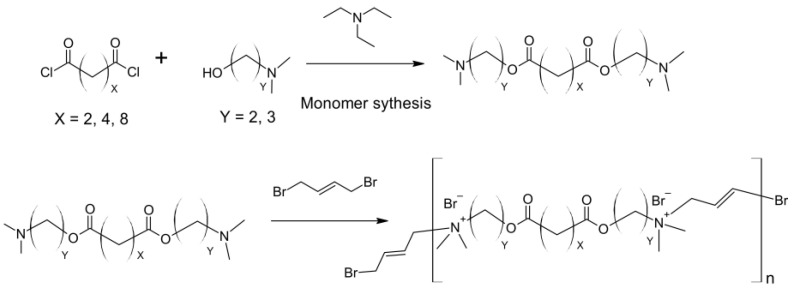
Scheme of the polyionenes esters synthesis based on [340].

**Figure 21 ijms-24-07473-f021:**
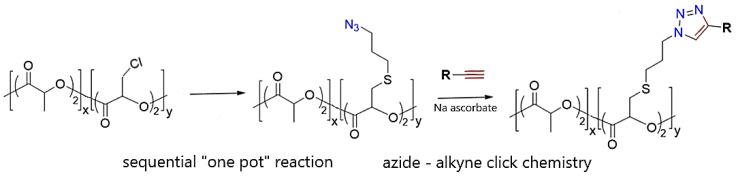
Synthesis of azide-substituted polylactide—based on [341].

**Figure 22 ijms-24-07473-f022:**
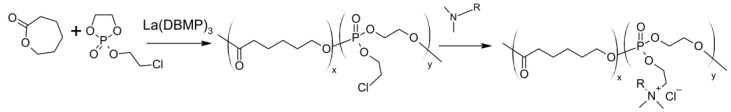
Synthesis of cationic poly(ester-*co*-phosphoester) with quaternary ammonium cation.

**Figure 23 ijms-24-07473-f023:**
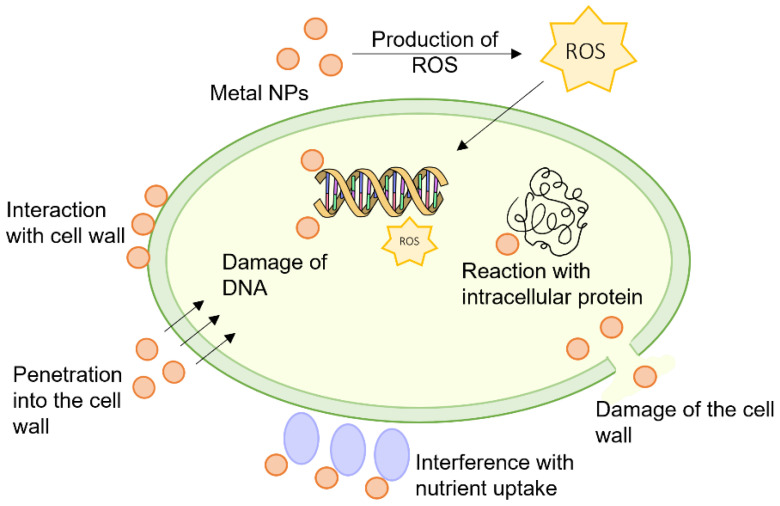
Mechanisms of antibacterial activity of metal nanoparticles. Based on [354].

**Figure 24 ijms-24-07473-f024:**
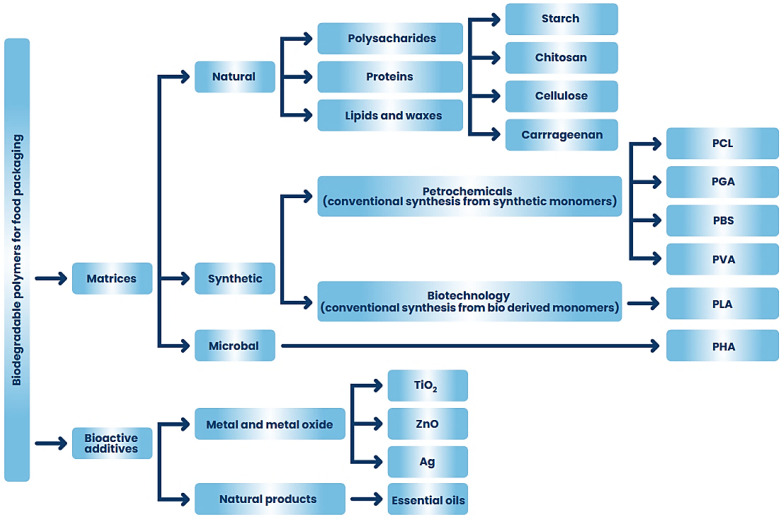
Classification and application of antibacterial polymers in the production of food packaging.

**Figure 25 ijms-24-07473-f025:**
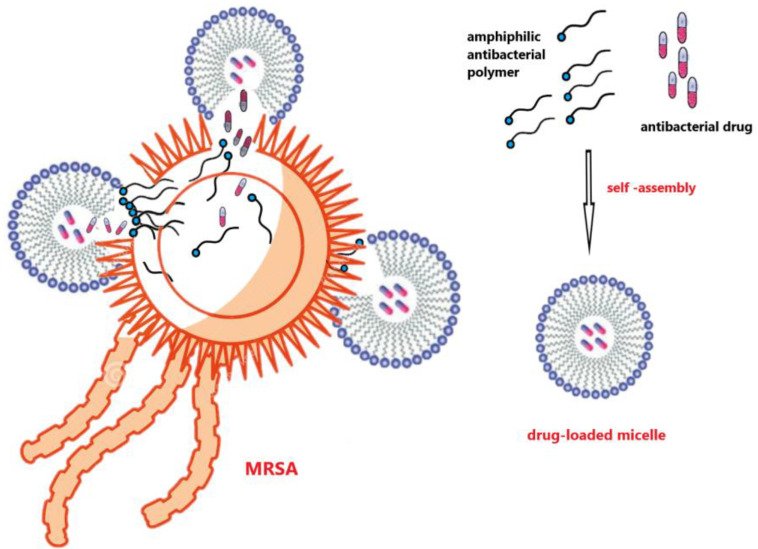
Action mechanism of the antibiotic-loaded micellar particles formed with antibacterial amphiphilic polymer for the elimination of MRSA.

**Table 1 ijms-24-07473-t001:** The antimicrobial activity of chitosan and chitosan derivatives.

Materials	Typ of Microorganism	Antimicrobial Activity	Ref.
CS (chitosan) (0.2% *w*/*v*)	*B. cereus*	ZOI = 4 mm	[81]
	*P. aeruginosa*	ZOI = 2 mm	
CS (1.5% *w*/*v*)	*B. cereus*	ZOI = 12 mm	
	*P. aeruginosa*	ZOI = 10 mm	
CS (Mw = 322.04 kDa)	*S. aureus*	MIC = 80 μg/mL	[82]
	*E. coli*	MIC = 80 μg/mL	
CS (Mw = 41.1 kDa)	*S. aureus*	MIC = 32 μg/mL	
	*E. coli*	MIC = 64 μg/mL	
CS (Mw = 14.3 kDa)	*S. aureus*	MIC = 32 μg/mL	
	*E. coli*	MIC = 32 μg/mL	
CS (Mw = 5.06 kDa)	*S. aureus*	MIC = 32 μg/mL	
	*E. coli*	MIC = 16 μg/mL	
CS (Mw = 110 kDa; DA = 2)	*E. coli*	ZOI = 12.5 mm; MIC = 0.01%	[74]
	*K. pneumoniae*	ZOI = 12.0 mm; MIC = 0.002%	
	*S. typhi*	ZOI = 10.5 mm; MIC = 0.005%	
	*S. aureus*	ZOI = 9.5 mm; MIC = 0.025%	
	*B. cereus*	ZOI = 10.0 mm; MIC = 0.025%	
CS (Mw = 42.5 kDa; DA = 2)	*E. coli*	ZOI = 12.5 mm; MIC = 0.005%	[74]
	*K. pneumoniae*	ZOI = 11.5 mm; MIC = 0.001%	
	*S. typhi*	ZOI = 10.5 mm; MIC = 0.002%	
	*S. aureus*	ZOI = 8.5 mm; MIC = 0.025%	
	*B. cereus*	ZOI = 9.0 mm; MIC = 0.025%	
CS (67 kDa; 5% *w*/*v*)	*B. cereus*	ZOI = 51.5 ± 0.42 mm; MIC = 0.625%	[83]
	*S. aureus*	ZOI = 33.4 ± 0.53 mm; MIC = 1.25%	
	*E. coli*	ZOI = 49.7 ± 0.31 mm; MIC = 0.156%	
	*S. typhi*	ZOI = 41.7 ± 0.60 mm; MIC = 0.312%	
	*P. aeruginosa*	ZOI = 50.4 ± 0.71 mm; MIC = 0.312%	
	*A. niger*	ZOI = 53 ± 0.94 mm; MIC = 0.07%	
	*C. albicans*	ZOI = 30.8 ± 0.88 mm; MIC = 1.25%	
CS (0.5% *w*/*v*)	*E. coli*	MIC = 256 μg/mL	[84]
	*S. enteritidis*	MIC = 128 μg/mL	
	*L. monocytogenes*	MIC = 128 μg/mL	
	*S. aureus*	MIC = 256 μg/mL	
	*B. cereus*	MIC = 256 μg/mL	
	*C. albicans*	MIC = 64 μg/mL	
QC (quaternized chitosan)	*E. coli*	ZOI = 12.3 ± 0.1 mm	[85]
	*S. aureus*	ZOI = 13.7 ± 0.1 mm	
	*P. aeruginosa*	ZOI = 14.3 ± 0.1 mm	
	*C. albicans*	ZOI = 14.6 ± 0.1 mm	
QC + Ag NPs (0.125% wt%)	*E. coli*	ZOI = 15.6 ± 0.1 mm	[85]
	*S. aureus*	ZOI = 16.3 ± 0.1 mm	
	*P. aeruginosa*	ZOI = 17.7 ± 0.1 mm	
	*C. albicans*	ZOI = 19.6 ± 0.1 mm	
2,6 DAC (2,6-diamino chitosan)	*E. coli*	MIC = 16–32 μg/mL	[86]
	*S. aureus*	MIC = 16 μg/mL	
	*P. aeruginosa*	MIC = 8 μg/mL	
CMCs (Carboxymethyl chitosan)	*B. subtilis*	ZOI = 5 mm	[87]
	*S. aureus*	ZOI = 7 mm	
	*S. faecalis*	ZOI = 6 mm	
	*E. coli*	ZOI = 8 mm	
	*P. aeruginosa*	ZOI = 7 mm	
TMC(N,N,N-trimethyl chitosan)	*E. coli*	MIC = 0.125 μg/mL	[88]
	*S. aureus*	MIC = 0.0625 μg/mL	
	*E. facialis*	MIC = 128 μg/mL	
	*P. aeruginosa*	MIC = 256 μg/mL	
SCS (Sulfonated chitosan)	*E. coli*	MIC = 0.13 μg/mL	[89]
	*S. aureus*	MIC = 2.00 μg/mL	
	*A. sacchari*	MIC = 64 μg/mL	
	*B. cinerea*	MIC = 0.25 μg/mL	
Phosphorylated chitosan (concentration 100%)	*V. cholerae*	good activity (11–15 mm dia)	[90]
	*K. pneumoniae*	weak activity (7–10 mm dia)	
	*Salmonella* sp.	weak activity (7–10 mm dia)	
	*S. aureus*	very good activity (above 16 mm dia)	
	*V. alginolyticus*	weak activity (7–10 mm dia)	
	*V. parahemolyticus*	good activity (11–15 mm dia)	
	*P. vugaris*	weak activity (7–10 mm dia)	
CS	*E. coli*	ZOI = 24 ± 0.63 mm	[91]
	*K. pneumoniae*	ZOI = 26 ± 0.73 mm	
	*S. aureus*	means not detected	
	*S. mutans*	means not detected	
	*C. albicans*	ZOI = 26 ± 0.79 mm	
	*A. fumigatus*	ZOI = 16 ± 0.83 mm	
CSSBs (Chitosan Schiff bases) with 2-chloroquinoline-3-carbaldehyde	*E. coli*	ZOI = 22 ± 0.73 mm	[91]
	*K. pneumoniae*	ZOI = 22 ± 0.73 mm	
	*S. aureus*	ZOI = 22 ± 0.30 mm	
	*S. mutans*	ZOI = 15 ± 0.89 mm	
	*C. albicans*	ZOI = 34 ± 0.99 mm	
	*A. fumigatus*	ZOI = 26 ± 0.91 mm	
CSSBs with quinazoline-6-carbaldehyde	*E. coli*	ZOI = 27 ± 0.83 mm	[91]
	*K. pneumoniae*	ZOI = 27 ± 0.72 mm	
	*S. aureus*	ZOI = 20 ± 1.20 mm	
	*S. mutans*	ZOI = 17 ± 0.50 mm	
	*C. albicans*	ZOI = 31 ± 1.29 mm	
	*A. fumigatus*	ZOI = 25 ± 0.72 mm	
CSSBs with oxazole-4-carbaldehyde	*E. coli*	ZOI = 22 ± 0.98 mm	[91]
	*K. pneumoniae*	ZOI = 26 ± 0.65 mm	
	*S. aureus*	ZOI = 19 ± 0.62 mm	
	*S. mutans*	ZOI = 18 ± 1.20 mm	
	*C. albicans*	ZOI = 26 ± 0.49 mm	
	*A. fumigatus*	ZOI = 21 ± 0.65 mm	
CS-NPs	*E. coli*	MIC = 117 μg/mL	[92]
	*S. choleraesuis*	MIC = 117 μg/mL	
	*S. aureus*	MIC = 234 μg/mL	
CS-Ag^+^ NPs	*E. coli*	MIC = 3 μg/mL	[92]
	*S. choleraesuis*	MIC = 3 μg/mL	
	*S. aureus*	MIC = 6 μg/mL	
CS-Cu^2+^ NPs	*E. coli*	MIC = 9 μg/mL	[92]
	*S. choleraesuis*	MIC = 9 μg/mL	
	*S. aureus*	MIC = 21 μg/mL	
CS-Zn^2+^ NPs	*E. coli*	MIC = 18 μg/mL	[92]
	*S. choleraesuis*	MIC = 18 μg/mL	
	*S. aureus*	MIC = 36 μg/mL	
CS-Mn^2+^ NPs	*E.coli*	MIC = 73 μg/mL	[92]
	*S.choleraesuis*	MIC = 73 μg/mL	
	*S. aureus*	MIC = 85 μg/mL	
CS-Fe^2+^ NPs	*E. coli*	MIC = 121 μg/mL	[92]
	*S. choleraesuis*	MIC = 121 μg/mL	
	*S. aureus*	MIC = 146 μg/mL	

MIC—minimum inhibitory concentration, ZOI—zone of inhibition.

**Table 2 ijms-24-07473-t002:** Antibacterial composites and polymer blends of carrageenans.

Materials	Bacteria	Antimicrobial Activity	Ref.
Carr/AgNP/Clay	*Listeria monocytogenes*	DI = 6.34 mm	[156]
	*Escherichia coli*	DI = 7.43 mm	
Carr/AgNP (melanin-mediated synthesis)	*Listeria monocytogenes*	MIC = 64 μM	[157]
	*Escherichia coli*	MIC = 16 μM	
Carr/HNT-AgNP(SDS)	*Listeria monocytogenes*	reduced the bacterial count within 12 h	[158]
	*Escherichia coli*	reduced the bacterial count within 12 h	
Carr/AgNP (from pine needle)	*Staphylococcus aureus*	3.5 Log CFU/mL reduction	[159]
	*Escherichia coli*	2 Log CFU/mL reduction	
SSPS/AgNPs/Carr	*Staphylococcus aureus*	ZOI = 1.83 ± 0.13 mm	[154]
	*Escherichia coli*	ZOI = 1.92 ± 0.25 mm	
CA–AgNPs (10 mM)	*Staphylococcus aureus*	ZOI = 17.67 + 1.15 mm	[160]
	*Escherichia coli*	ZOI = 12.67 + 0.41 mm	
Carr/AgNP (AgNO_3_ 0.6%)	*Staphylococcus aureus*	ZOI = 7.59	[161]
Carr/AgNP (AgNO_3_ 1.2%)	*Staphylococcus aureus*	ZOI = 9.12	
Ag/Fe/g-C_3_N_4_-Carr	*Klebsiella pneumoniae*	excellent antimicrobial	[162]
	*Enterococcus faecalis*	excellent antimicrobial	
Carr/Fe_3_O_4_@NH_2_-Ag	*Listeria monocytogenes*	10.09 vs. 3.93 log reduction	[163]
	*Escherichia coli*	8.82 vs. 5.02 log reduction	
Carr/Fe_3_O_4_@SNP	*Listeria monocytogenes*	decrease growth by about 4 log cycles after 12 h of incubation compared to neat Carr	[164]
	*Escherichia coli*	decrease growth by about 3.5 log cycles after 12 h of incubation compared to neat Carr	
Pul/Carr/DL/CuS	*Listeria monocytogenes*	decrease growth by about 2 log cycles after 12 h of culturing compared to the control	[165]
	*Escherichia coli*	decrease growth by about 7 log cycles after 12 h of culturing compared to the control	
Carr/CuS	*Staphylococcus aureus*	reduced the bacterial counts 69.8 ± 1.8%	[166]
	*Escherichia coli*	reduced the bacterial counts52.6 ± 5.4%	
Carr/ZnONPs	*Staphylococcus aureus*	reduced the bacterial count (3 to 4 log reductions) within 24 h	[167]
	*Escherichia coli*	reduced the bacterial count (4 log reductions) within 24 h	
Carr/CuONPs	*Staphylococcus aureus*	reduced the bacterial count (3 to 4 log reductions) within 24 h	
	*Escherichia coli*	reduced the bacterial count (4 log reductions) within 24 h	
Carr/SiO_2_NPs	*Staphylococcus aureus*	no antibacterial activity	
	*Escherichia coli*	reduced the bacterial count (3 log reductions) within 24 h	
Kappaphycus alvarezii/ZnONPs	*Staphylococcus aureus*	reduced the bacterial count (2 to 4 log reductions) within 24 h	
	*Escherichia coli*		
Kappaphycus alvarezii/CuONPs	*Staphylococcus aureus*		
	*Escherichia coli*		
Kappaphycus alvarezii/SiO_2_NPs	*Staphylococcus aureus*		
	*Escherichia coli*		
Carr/Lig/AgNPs/CaCl_2_ hydrogel	*Staphylococcus aureus*	completely killed within 6 h of incubation	[168]
	*Escherichia coli*	completely killed within 3 h of incubation	
Carr/Lig/AgNPs/CuCl_2_ hydrogel	*Staphylococcus aureus*	completely killed within 3 h of incubation	
	*Escherichia coli*	completely killed within 3 h of incubation	
Carr/Lig/AgNPs/MgCl_2_ hydrogel	*Staphylococcus aureus*	completely killed within 6 h of incubation	
	*Escherichia coli*	completely killed within 6 h of incubation	
Carr/CNC/AgNPs cryogel	*Staphylococcus aureus*	R = 100%	[169]
	*Escherichia coli*	R = 100%	
Ag/Carr/Gelatin hydrogel	*Streptococcus agalactiae*	zone of clearance = 21 mm	[170]
	*Streptococcus pyogenes*	zone of clearance = 18 mm	
	*Escherichia coli*	zone of clearance = 19 mm	
Carr/KCl/ZnO/CuO	*Listeria monocytogenes*	increased slightly till 3 h of incubation then decreased linearly as the time increased	[171]
	*Escherichia coli*	completely killed within 6–9 h of incubation	
gelatin/carr/bacterial cellulose hydrogel scaffolds	*Staphylococcus aureus*	5.4 ± 0.43 mm	[172]
	*Escherichia coli*	3.1 ± 0.88 mm	
	*Klebsiella pneumonia*	resistance against the bacteria	
KaMA-ZnO/PD hydrogel	*Staphylococcus aureus*	ZOI = ~2.05 mm	[173]
	*Escherichia coli*	ZOI = 2.8 ± 0.4 mm	
H-OCA-Dop-Zn hydrogel	*Staphylococcus aureus*	96.3% bacterial reduction	[174]
	*Escherichia coli*	99.6% bacterial reduction	
CO-CNF 400 mg SHκ-carrageenan oligosaccharides linked cellulose nanofibers hydrogel loaded 400 mg surfactin and Herbmedotcin	*Streptococcus mutans*	ZOI = 26.33 ± 1.52 mmMIC = 60%	[175]
	*Porphyromonas gingivalis*	ZOI = 18.33 ± 0.57 mmMIC = 50%	
	*Fusobacterium nucleatum*	ZOI = 20.33 ± 0.63 mmMIC = 70%	
	*Pseudomonas aeruginosa*	ZOI = 20.66 ± 1.25 mmMIC = 40%	
ampicillin sodium salt-loaded PVA/HA-ĸ-Carr hydrogel	*Staphylococcus aureus*	ZOI = 12 mm	[176]
	*Escherichia coli*	ZOI = 13 mm	
ciprofloxacin-loaded PVA/Carr/HA hydrogel	*Staphylococcus aureus*	MIC and/or ZOI not given. Only photos are show	[178]
	*Escherichia coli*		
Berberine-loaded /Carr/KGM hydrogel	*Staphylococcus aureus*	ZOI = 16.1 ± 0.2 mm	[179]
	*Candida albicans*	ZOI = 12.4 ± 0.1 mm	
Carr/GSE/SNP hydrogel	*Staphylococcus epidermis*	destroy the bacteria within 3 h	[180]
	*Escherichia coli*	destroy the bacteria within 3 h	[181]
AR/rGO/Carr hydrogel	*Staphylococcus aureus*	ZOI ≈ 33 mm	[182]
	*Pseudomonas aeruginosa*	ZOI ≈ 31 mm	
	*Escherichia coli*	ZOI ≈ 29 mm	
Carr/Agar/MMT/CLPCarr/Agar/MMT/LDC/CLP	*Staphylococcus aureus*	ZOI = 25.7 ± 1.2 mm	[183]
	*Escherichia coli*	ZOI = 31.0 ± 1.0 mm	
	*Staphylococcus aureus*	ZOI = 29.3 ± 1.2 mm	
	*Escherichia coli*	ZOI = 29.7 ± 0.6 mm	

DI—diameter of inhibition zone, MIC—minimum inhibitory concentration, ZOI—zone of inhibition.

**Table 4 ijms-24-07473-t004:** AMPs in the clinical phase of development.

Peptide Name/Analog	Application	Clinical Phase
**Medicine application**		
Human lactoferrin	Bacterial infections and mycoses	Approved
Vancomycin	Staphylococcal infections	Approved
Gramicidin/Cationic polycyclic peptide	Purulent skin disease	Approved
Colistin	Multidrug-resistance Gram-negative infections	Phase IV
SGX942(Dusquetide)/IDR-1	Oral mucositis	Phase III
Pexiganan/Magainin	Topical application for diabetic foot ulcers	Phase III
PXL01/Lactoferrin	Postsurgical adhesions	Phase II
Omiganan/Indolicidin	Acne, atopic dermatitis	Phase II
PAC 113/Histatin	Oral candidiasis (mouth wash) Treatment of inflammation and ulceration	Phase II
PMX-30063 Brilacidin/Defensin mimetic	Acute bacterial skin infection	Phase II
OP145/LL-37	Bacterial ear infection Topical cream for prevention of catheter	Phase I/II
LL-37/Leucine	Melanoma	Phase I/II
**Food industry**		
Nisin	Dairy (*L. monocytogenes* and *S. aureus*)	Approved
Polylysine/Natural cationic antibacterial agent	Sushi, boiled rice, noodles, meat, and drinks	Approved

**Table 5 ijms-24-07473-t005:** Biodegradable polymer composites and their antibacterial activity.

Polymer	Nanoparticle	Form of Composite	Application	Bacteria	Antibacterial Activity	References
PLA-polylactide	ZnO	Nanofibers, nanofibrous mats,	wound dressings, tissue regenerative applications	*E. coli*, *S. aureus*	MIC = 6.5 ± 10^−4^ mg/L	[361]
	Carbon doped TiO_2_	Nanocomposite films (H-PLA/3GST)	Wound dressing	*S. aureus*	ZOI = 28 mm	[364]
	Al_2_O_3_ + Ag	PLA-Al_2_O_3_/Ag fiber mats (25% of Al_2_O_3_/Ag powder concentration)	Antibacterial applications	*E. coli*	ZOI = 10.55 mm	[362]
				*Sarcina lutea*	ZOI = 13.88 mm	
	Ag	Nanofiber membrane (PLA—2Ag)	wound dressing	*E. coli*	ZOI = 7.57 mm	[363]
				*S. aureus*	ZOI = 6.75 mm	
	Zn[(acac)(L)H_2_O], where L represents N-(2-pyridin-4-ylethylidene) phenylalanine	Initiator of polymerization, M/I ratio as 1:400	drug delivery, wound healing	*S. aureus* *S. epidermidis*, *P. aeruginosa*	MIC = 100 µg/mL	[365]
PVA-poly vinyl alcohol	ZnO	nanocomposite fibers	wound healing and tissue reconstruction	*E. coli*	MIC = 62.5 μg/ mL	[366]
				*S. aureus*	MIC = 250 μg/ mL	
	Se	PVA/Chitosan/SeNPs nanocomposite films (at 30 min laser ablation)	antimicrobial applications	*E. coli*	Activ. Index = 69%	[368]
				*S. aureus*	Activ. Index = 54%	
				*P. aeruginosa*	Activ. Index = 64%	
				*B. subtilis*	Activ. Index = 44%	
		PVA/carboxymethyl cellulose blend with selenium nanoparticles	Candidate for food packaging	*E. coli*	D(inhibition zone diameter) = 33 mm^2^	[369]
				*S. aureus*	D = 49 mm^2^	
				*P. aeruginosa*	D = 6 mm^2^	
				*B. cereus*	D = 39 mm^2^	
	CuS + chitosan	CuS/PVACS nanocomposite	Antibacterial aplications	*E. coli*	DI = 13.51 ± 0.33 mm	[367]
				*S. aureus*	DI = 23.81 ± 0.09 mm	
				*P. syringae*	DI = 18.23 ± 0.41 mm	
				*S. pneumoniae*	DI = 27.11 ± 0.31 mm	
PCL-polycaprolactone	ZnO	scaffolds of PCL compounded with hydroxyapatite and ZnO (6% ZnO)	bone tissue engineering	*S. aureus*	R = 96%	[371]
	Ag	nanocomposite membranes PCL/Ag (1%)	Wound dressing	*E. coli*	DI = 7.9 ± 0.6 mm	[370]
				*S. aureus*	DI = 11.6 ± 0.5 mm	
PLGA-poly(lactide-*co*-glycolide)	Ag + ZnO	PLGA/Ag/ZnO nanorods composite coating	-	*E. coli*	R = 60.8%	[372]
				*S. aureus*	R = 70.3%	
	azithromycin	poly(lactide-co-glycolide) nanoparticles loaded with azithromycin	Drug delivery	*Salmonella typhi*	MIC = 3.12 µg/mL	[373]
	Al_2_O_3_	PLGA/Al_2_O_3_ (0.1%) nanocomposite	candidate for packaging materials, biomedicine	*E. coli*	74% reduction in bacteria growth	[358]
	Fe_2_O_3_	PLGA/Fe_2_O_3_ (0.01%) nanocomposite	Candidate for packaging material in agriculture	*E. coli*	over 5-fold reduction in the number of cells	[374]
Chitosan	ZnO	Nanocomposite films	Food packaging	*E. coli*	Plate count values: 2.5 ± 0.421× 10^7^ cfu/g	[141]
				*S. aureus*	9 ± 0.367× 10^7^ cfu/g	
	ZnO	ZnO nanoparticles embedded in TPP—crosslinked chitosan	Sunscreen agent	*E. coli*	5 log reduction	[375]
				*S. aureus*	Total reduction	
	TiO_2_ + pectin	TiO_2_ nanotubes loaded with a drug on chitosan coating	Drug carrier	*E. coli*	ZOI = 45 mm	[376]
				*S. aureus*	ZOI = 45 mm	
				*P. aeruginosa*	ZOI = 47 mm	
				*B. subtilis*	ZOI = 49 mm	
	Ag	Ag nanoparticles in the porous chitosan matrix—sponges (15 mM Ag)	wound dressing	*E. coli*	6 Log reductions of viable cell numbers after 2 h of exposure	[377]
				*S. aureus*		
	Ag_2_O	The solution of the chitosan– Ag_2_O encapsulated nanocomposite film	Food packaging	*E. coli*	ZOI = 16 mm	[378]
				*S. aureus*	ZOI = 23 mm	
				*B. subtilis*	ZOI = 20 mm	
				*P. aeruginosa*	ZOI = 24 mm	
	Fe_3_O_4_ + gelatin	Fe_3_O_4_/CS/GE nanofiber membrane (1% Fe_3_O_4_)	Wound dressing	*E. coli*	ZOI = 20 mm	[392]
				*S. aureus*	ZOI = 19 mm	
	FeO	CS/FeO nanocomposite (40 μg/mL FeO)	Biological applications	*B. subtilis*	ZOI = 13.0 ± 0.5 mm	[379]
				*S. aureus*	ZOI = 12.0 ± 0.5 mm	
				*E. coli*	ZOI = 15.0 ± 0.5 mm	
	Au	chitosan-capped gold nanoparticles	Antibacterial applications	*E. coli*	MIC = 64 μg/mL	[380]
				*S. aureus*	MIC = 128 μg/mL	
				*Salmonella enterica*	MIC = 32 μg/mL	
				*L. monocytogenes*	MIC = 4 μg/mL	
	MnS + CaAlg	MnS_2_/CS-CaAlg	Antibacterial applications	*E. coli*, *S. aureus*	reduced the bacterial count (6 log to 7 log reductions in 60 min)	[393]
	Tobramycin	Tobramycin-chitosan nanoparticles (TOB-CS NPs)	Drug delivery	*E. coli*	MIC = 11.30 ± 0.12 mg/mL	[381]
				*S. aureus*	MIC = 15.60 ± 0.09 mg/mL	
	Tobramycin + ZnO	Tobramycin-chitosan nanoparticles (TOB-CS NPs) coated with zinc oxide nanoparticles (ZnO NPs)	Drug delivery	*E. coli*	MIC = 8.40 ± 0.11 mg/mL	[381]
				*S. aureus*	MIC = 10.70 ± 0.08 mg/mL	
	tetracycline	tetracycline hydrochloride loaded into 1% fungal chitosan	Drug delivery	*E. coli*	ZOI = 26.17 ± 1.53 mm	[394]
				*B. sbtilis*	ZOI = 20.0 ± 1.4 mm	
				*S. aureus*	ZOI = 22.0 ± 1.4 mm	
cellulose	ZnO + chitosan	cellulose/chitosan composite films loaded with ZnO nanoplates (3% ZnO)	wastewater treatment, antibacterial materials	*E. coli*	ZOI = 13.6 ± 0.2 mm	[382]
				*S. aureus*	ZOI = 20.6 ± 0.2 mm	
	Ag	spherical-nano cellulose (SNC)/silver-nanoparticle (AgNP)	Antibacterial and catalytic applications	*E. coli*	DI = 12.1 ± 0.91 mm	[383]
				*S. aureus*	DI = 11.3 ± 0.01 mm	
	AgO	Cellulose nanofibers /PVA/SA-AgO (CPS-AgO) aerogel	Antibacterial aerogel	*E. coli*	ZOI = 23 mm	[384]
				*S. aureus*	ZOI = 20 mm	
	Al_2_O_3_	Conformal coatings with Al2O3 on viscose fabrics	Antibacterial applications	*E. coli*	Inhibition rate of bacterial growth = 71%	[385]
				*S. aureus*	Inhibition rate of bacterial growth = 65%	
dextran	Ag	Dextran-coated silver nanoparticles	Food packaging	*E. coli*	R = 99.9% (after 24 h of exposure)	[386]
alginate	ZnO	ZnO nanoparticles on alginate (~11 wt.% ZnO)	Antibacterial applications	*E. coli*	R = 100%	[388]
				*S. aureus*	R = 99.91% (after 2 h of exposure)	
	Ag	copper alginate hydrogel doped with Ag nanoparticles (5% Ag)	Antibacterial applications	*E. coli*	ZOI ≈ 23 mm	[395]
				*S. aureus*	ZOI ≈ 35 mm	
starch	MgO + albumin	starch-Albumin-magnesium oxide (S-A-MgO) film	Antibacterial applications	*E. coli*	DI = 5 ± 0.13 mm	[389]
				*S. aureus*	DI = 6 ± 0.22 mm	
	Ag_2_O	Starch-capped Ag_2_O nanoparticles (2:1 Ag_2_O ratio)	Antibacterial applications	*E. coli*	ZOI = 14 ± 0.11 mm	[390]
				*S. aureus*	ZOI = 15 ± 0.19 mm	
	Fe_2_O_3_	polyaniline/starch/hematite biocomposite	water purifier	*S. typhimurium*	IZ = 18.02 mm	[391]
				*S. aureus*	IZ = 13.98 mm	
				*B. subtilis*	IZ = 14.12 mm	

DI—diameter of inhibition zone, MIC—minimum inhibitory concentration, ZOI—zone of inhibition.

## Data Availability

Not applicable.

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
