# Peer review of "Biodegradable Polymers and Polymer Composites with Antibacterial Properties"

_ijms, 2023, doi:10.3390/ijms24087473_

Round 1

Reviewer 1 Report

This is an excellent review paper, with a detailed review of the literature, very well written and organized. I just have two very small considerations:

1.       It is necessary to review the references formatting: The incorporation of the references is the text is not always the same, see for example, page 3 line 127.

2.       There is a legend in the end of Table 2, but on Table 1 is missing.

Author Response

This  is  an  excellent  review  paper,  with  a  detailed  review  of  the  literature,  very  well  writt en  and organized. I just have two very small considerations:

Thank you for your favourable opinion.

  1. It is necessary to review the references formatt ing: The incorporation of the references i s the text is not always the same, see for example, page 3 line 127.

We have corrected it according to the reviewer's comment 

  1. There is a legend in the end of Table 2, but on Table 1 is missing.

We added the legends to all tables (Tables 1, 3 and 5).

Reviewer 2 Report

This review considers the main classes of biological polymers for the creation of antibacterial composite materials. The text of the review can be presented in the form of 3 main blocks: 1. classification of biodegradable polymers and mechanisms of their antibacterial activity; 2 synthesis of antibacterial composite materials and methods for their improvement; 3. application of composite materials based on biodegradable polymers in the food industry (as packaging). Despite the more complex structure of the sections of the text of this review, it contains useful and comprehensive information on the synthesis of antibacterial composite materials based on biodegradable polymers, including those of natural origin, considers a large number of relevant studies in this area, as well as the areas of practical application of these materials. There are only minor comments, which is still recommended to be corrected:
- The manuscript is too big. I don’t know what to do with this, I just inform the authors that I have been reading the manuscript for a very long time. In general, I liked it!
- The manuscript often contains repetitions, for example, the description of chitosan (and not only), the mechanisms of its antibacterial activity are present in the text simultaneously in several sections (line 334-786, 1723, 1733, 1909).
- Some illustrations are mechanistic. For example, Figure 5. The essence of the presented transitions is in the change in the conformation of residues, and not in a simple modification of the chemical structure. In figure 1, there are algae that are not a taxonomic unit due to polyphyly. It was possible to add, for example, archaea.
- There are several types of nanoparticles in the review manuscript, but there are absolutely no articles about a number of common nanoparticles. For example, nanoparticles of aluminum oxide (10.1016/j.reactfunctpolym.2021.105143 or 10.3390/jcs6100298) or iron oxide (10.3390/inventions7030061 or 10.3390/polym14224880) or Ag2O or Se.
- There is no structured explanation in the subchapter on nanoparticles why metal and metal oxide nanoparticles have an antibacterial effect. See how many times this has been presented in previous reviews (10.3390/nano12152635 or 10.3390/ph15080968). Maybe add a picture?

Author Response

This review considers the main classes of biological polymers for the creation of antibacterial composite materials. The text of the review can be presented in the form of 3 main blocks: 1. classification of biodegradable polymers and mechanisms of their antibacterial activity; 2 synthesis of antibacterial composite materials and methods for their improvement; 3. application of composite materials based on biodegradable polymers in the food industry (as packaging). Despite the more complex structure of the sections of the text of this review, it contains useful and comprehensive information on the synthesis of antibacterial composite materials based on biodegradable polymers, including those of natural origin, considers a large number of relevant studies in this area, as well as the areas of practical application of these materials. There are only minor comments, which is still recommended to be corrected:

Thank you very much for your opinion and valuable comments that make our manuscript more scientifically attractive.

- The manuscript is too big. I don’t know what to do with this, I just inform the authors that I have been reading the manuscript for a very long time. In general, I liked it!

We are pleased with the interest in the presented content of the manuscript. We are aware that the work is extensive, perhaps a little too long. On the other hand, we wanted to include the most important threads of the subject in one paper. Moreover, the other reviewers suggested adding new chapters, which significantly increased the volume of the manuscript. However, we had to remove a large part of the collected data, and selected subjects we had to shorten or omit. We think that not every reader has to read everything, but we hope that he or she will find something interesting and valuable.

- The manuscript often contains repetitions, for example, the description of chitosan (and not only), the mechanisms of its antibacterial activity are present in the text simultaneously in several sections (line 334-786, 1723, 1733, 1909). –

As suggested by the reviewer, we shortened or even removed the repeated descriptions. We have also moved the part concerning the mechanism of antibacterial action of chitosan and its derivatives to the section of the manuscript describing the mechanism of antibacterial action of polymers containing amino groups (pages 3-4).

- Some illustrations are mechanistic. For example, Figure 5. The essence of the presented transitions is in the change in the conformation of residues, and not in a simple modification of the chemical structure. In Figure 1, there are algae that are not a taxonomic unit due to polyphyly. It was possible to add, for example, archaea.

The reviewer is absolutely right because the antimicrobial activity of chitosan is a complex issue. However, Figure 5 (now Figure 1) focuses only on a schematic representation of the differences in the interaction of low and high-molecular chitosan with various types of microorganisms (Gram-positive, Gram-negative and fungi) depending on the structure of the cell membrane/cell wall and its ability to penetrate it or interact with its surface. Figure 1 (now Figure 2) has been revised as suggested by the reviewer.

- There are several types of nanoparticles in the review manuscript, but there are absolutely no articles about a number of common nanoparticles. For example, nanoparticles of aluminum oxide (10.1016/j.reactfunctpolym.2021.105143 or 10.3390/jcs6100298) or iron oxide (10.3390/inventions7030061 or 10.3390/polym14224880) or Ag2O or Se.
- There is no structured explanation in the subchapter on nanoparticles why metal and metal oxide nanoparticles have an antibacterial effect. See how many times this has been presented in previous reviews (10.3390/nano12152635 or 10.3390/ph15080968). Maybe add a picture?

Following the suggestions, we have extended the section on polymer antibacterial composites (Chapter 5). We have added these data to Table 5 presented there and introduced a drawing illustrating the mechanism of antibacterial action of the discussed composites.

Reviewer 3 Report

This review provides extensive information about the biodegradable and composite polymers with intrinsic antibacterial features and suitable to consider for publication in the IJMS; however, some comments appended below should be addressed as follows:

1.  The authors should scrutinize the entire review and correct some grammatical mistakes and typos, which is commonly happened during writing a long manuscript.

2. The figures should be enlarged and improved.

3. Section of antimicrobial peptides; the author should discuss one to two examples, like sericin and fibroin; for instance, please check these articles (doi.org/10.1016/j.ijpharm.2022.122328; doi.org/10.3390/ma13245706).

4. The authors should discuss the intensive applications of antibacterial polymers in wound healing and drug delivery systems in brief. You could also illustrate two figures in this context.

5. Some recent publications, particularly those published in 2023 should be added; for instance, (doi.org/10.1016/j.ijpharm.2023.122649; doi.org/10.1016/j.ijbiomac.2023.123944; doi.org/10.1016/j.mtbio.2022.100499; doi.org/10.1016/j.ijbiomac.2023.123687).

6. Future perspectives should be discussed.  

Author Response

This review provides extensive information about the biodegradable and composite polymers with intrinsic antibacterial features and suitable to consider for publication in the IJMS; however, some comments appended below should be addressed as follows:

We thank you for your suggestions and comments, we have tried to fully implement them in the revised version of our manuscript.

  1.  The authors should scrutinize the entire review and correct some grammatical mistakes and typos, which is commonly happened during writing a long manuscript.

We have carried out a thorough proofreading of the entire manuscript, in the hope of removing all errors and mistakes.

  1. The figures should be enlarged and improved.

Following the remark, we enlarged the selected drawings and increased their sharpness and contrast.

  1. Section of antimicrobial peptides; the author should discuss one to two examples, like sericin and fibroin; for instance, please check these articles (doi.org/10.1016/j.ijpharm.2022.122328; doi.org/10.3390/ma13245706).

As noted by the reviewer, we have supplemented this section with a description of these suggested proteins. We also entered the relevant data on these proteins into Table 3.

  1. The authors should discuss the intensive applications of antibacterial polymers in wound healing and drug delivery systems in brief. You could also illustrate two figures in this context.

We have introduced an additional chapter 7, which describes the suggested topics. We focused on a brief characterization of recent achievements in the use of antibacterial polymers in the formation of drug-release systems and multifunctional wound dressings.

  1. Some recent publications, particularly those published in 2023 should be added; for instance, (doi.org/10.1016/j.ijpharm.2023.122649; doi.org/10.1016/j.ijbiomac.2023.123944; doi.org/10.1016/j.mtbio.2022.100499; doi.org/10.1016/j.ijbiomac.2023.123687).

The suggested articles were analyzed and mentioned on the 17 and 18 pages in the body text.

  1. Future perspectives should be discussed.  

We have developed the final chapter 8 , introducing a discussion of the prospects for research development and future applications of the discussed polymers.

Round 2

Reviewer 2 Report

The manuscript has been significantly revised. I have no suggestions for authors to improve the manuscript. Recommended for publication.